# Duality and form factors in the thermally deformed two-dimensional tricritical Ising model

Axel Cortés Cubero[1], Robert M. Konik[2], Máté Lencsés[3,4],
Giuseppe Mussardo[5] and Gabor Takács[3,6]

**1** Department of Mathematical Sciences,
University of Puerto Rico, Mayaguez, Puerto Rico
**2** Condensed Matter and Materials Physics Division,
Brookhaven National Laboratory, Upton, NY 11973-5000, USA
**3** Department of Theoretical Physics, Institute of Physics,
Budapest University of Technology and Economics,
1111 Budapest, Műegyetem rkp. 3, Hungary
**4** BME-MTA Statistical Field Theory 'Lendület' Research Group,
Budapest University of Technology and Economics,
1111 Budapest, Műegyetem rkp. 3, Hungary
**5** SISSA and INFN, Sezione di Trieste, via Bonomea 265, I-34136, Trieste, Italy
**6** MTA-BME Quantum Dynamics and Correlations Research Group,
Budapest University of Technology and Economics,
1111 Budapest, Műegyetem rkp. 3, Hungary

## Abstract

The thermal deformation of the critical point action of the 2D tricritical Ising model gives rise to an exact scattering theory with seven massive excitations based on the exceptional $E_7$ Lie algebra. The high and low temperature phases of this model are related by duality. This duality guarantees that the leading and sub-leading magnetisation operators, $\sigma(x)$ and $\sigma'(x)$, in either phase are accompanied by associated disorder operators, $\mu(x)$ and $\mu'(x)$. Working specifically in the high temperature phase, we write down the sets of bootstrap equations for these four operators. For $\sigma(x)$ and $\sigma'(x)$, the equations are identical in form and are parameterised by the values of the one-particle form factors of the two lightest $\mathbb{Z}_2$ odd particles. Similarly, the equations for $\mu(x)$ and $\mu'(x)$ have identical form and are parameterised by two elementary form factors. Using the clustering property, we show that these four sets of solutions are eventually not independent; instead, the parameters of the solutions for $\sigma(x)/\sigma'(x)$ are fixed in terms of those for $\mu(x)/\mu'(x)$. We use the truncated conformal space approach to confirm numerically the derived expressions of the matrix elements as well as the validity of the $\Delta$-sum rule as applied to the off-critical correlators. We employ the derived form factors of the order and disorder operators to compute the exact dynamical structure factors of the theory, a set of quantities with a rich spectroscopy which may be directly tested in future inelastic neutron or Raman scattering experiments.

# 1 Introduction

One of the most intriguing aspects of theoretical physics is the self-duality of some statistical models, i.e. the existence of an exact mapping between their high and low temperature phases. As well known, such a mapping was originally discovered by Kramers and Wannier in the two-dimensional Ising model [1, 2] and led to the notion of *disorder operators* as the dual companions of the *order operators*. As further clarified by Kadanoff and Ceva [3], the disorder operators have small fluctuations at high temperature when the original order operators have instead large fluctuations, in particular, disorder operators have a vanishing expectation value in the broken phase of the model but a non-vanishing expectation value in the disordered phase. For the two-dimensional Ising model, the duality can be easily stated by considering the microscopical realisation as spins $s_i = \pm 1$ on a lattice with sites $i = 1, \dots, N$ with the Hamiltonian

$$\mathcal{H} = -J \sum_{\langle i,j \rangle}^{N} s_i s_j \,, \tag{1.1}$$

with $\langle i, j \rangle$ indicating that the sum runs over nearest neighbor sites. The self-duality of the model asserts that an $n$-point correlation function $\langle \sigma(x_1) \dots \sigma(x_n) \rangle$ of the order operator $\sigma(x)$ taken at the inverse temperature parameter $\beta = J/k_B T$ is equal to the $n$-point correlation function $\langle \mu(x_1) \dots \mu(x_n) \rangle$ of the disorder operator $\mu(x)$ at the dual inverse temperature parameter $\widetilde{\beta} = -1/2 \log \tanh \beta$.

Dualities of the Kramers–Wannier type are extremely useful in understanding universality classes and phase diagrams of a model. In the special case of two-dimensional critical phenomena, invariant under general symmetries (such as translation, rotation and dilation transformations) and with universal properties described by conformal field theories [4, 5], it has been shown that the information about duality is encoded in the fusion algebra of special fields associated with conformal defects [6–9].

In this paper we address the explicit computation of the form factors of the order and disorder operators in the perturbed tricritical Ising model (TIM). The TIM is the second conformal unitary minimal model, the first being the ordinary Ising model [4]. The class of universality of the TIM admits several microscopic realisations such as, for instance, metamagnets or multi-component fluid mixtures (see, for instance, [10] for a more detailed discussion). To illustrate the main properties of this class of universality, it is useful to consider a classical two-dimensional lattice model, known as the Blume–Capel model [11–15]. This theoretical model involves two statistical variables at each lattice site: $s_k$ – the spin variable – which assumes values $\pm 1$ and $t_k$ – the vacancy variable – with values 0 or 1, which therefore specifies whether the site is empty or occupied. With these variables, the most general Hamiltonian with nearest neighbor pair interaction is given by

$$\mathcal{H} = -J \sum_{\langle i,j \rangle}^{N} s_i s_j t_i t_j - \Omega \sum_{i=1}^{N} t_i - H \sum_{i=1}^{N} s_i t_i - H_3 \sum_{\langle i,j \rangle}^{N} (s_i t_i t_j + s_j t_j t_i) - K \sum_{\langle i,j \rangle}^{N} t_i t_j \,, \tag{1.2}$$

where the parameter $H$ represents an external magnetic field, $H_3$, an additional magnetic source (called the *subleading magnetic field*), $J$, the coupling between two nearest occupied sites, $\Omega$, the chemical potential coupled to the vacancies and $K$, a subleading energy term between them. The tricritical point is reached by switching $K$ to zero (together with the magnetic $H$ and the subleading magnetic field $H_3$) and properly tuning the remaining couplings $J$ and $\Omega$ to some critical values, $J_c$ and $\Omega_c$. As is well known, at the tricritical point a first order phase transition line meets a second order phase transition line: in the two parameter space spanned by $J$ and $\Omega$, the phase diagram is qualitatively the one given in Figure 1.1, with the end point of the second order phase transition line given by the critical Ising model, since at $\Omega \to \infty$

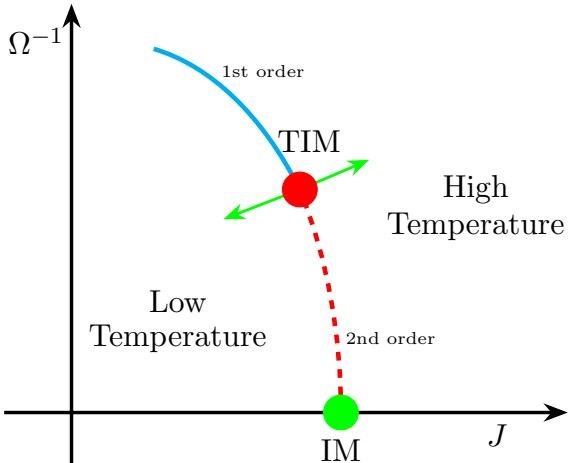

Figure 1.1: Qualitative phase diagram of the TIM in the plane of the two variables $J$ and $\Omega$.

the vacancies are suppressed, all sites are populated with spins $s_i = \pm 1$ but the system can be nevertheless fine-tuned to remain critical[1].

Right at the tricritical point, the model is conformally invariant. This property allows us to establish a correspondence between the operator content present in the Blume–Capel model and the operator content inferred from the CFT. In two dimensions, both the Ising and the tricritical Ising model are self-dual and, as shown below, this common property of these two models can be easily understood in terms of the existence of fermionic fields in their operator content but with an important difference: while in the IM such a field is the familiar free Majorana fermion $\Psi(x)$ of the model [17, 18], in the TIM, on the contrary, the fermionic field is given by the operator $G(x)$ which gives rise to the supersymmetry of the model [19–21]. For both models, the zero mode of these fermionic fields implies the existence of a pair of dual order and disorder operators with the same conformal dimension at the critical point.

Although closely related to the IM, the TIM has however significant new features which make the computation of the form factors (FF) of its order/disorder operators an interesting theoretical problem. First of all, while the thermal deformation of the critical point of the IM is described by a free Majorana field theory with only one massive excitation in the spectrum (which may be considered as a topological kink excitation in the low-temperature phase or as an ordinary particle excitation in the high-temperature phase) [17, 18], for the TIM such a deformation is instead associated to a scattering theory based on the exceptional algebra $E_7$ [22,23]. Hence, the spectrum of the TIM is much richer than the spectrum of the IM and, out of the seven massive excitations $A_1, A_2, \ldots, A_7$, some of which correspond to topological kink excitations in the low-temperature phase, while the others correspond to ordinary particles. Secondly, the operator content of the TIM is larger than the operator content of the IM, in particular in the TIM there are *two* $\mathbb{Z}_2$ odd order operators, i.e. the magnetic operator $\sigma(x)$ and the sub-leading magnetic operator $\sigma'(x)$, accompanied by their dual disorder operators, $\mu(x)$ and $\mu'(x)$. The operator content of a theory is usually pinned down in terms of its ultraviolet behaviour, i.e. in terms of the different highest weight modules of the conformal field theory [4] present at its critical point, but it can also be equivalently identified analysing its infrared behaviour, i.e. in terms of the different solutions of the form factor equations [24, 25]. This implies that there must be a complete equivalence between these ultraviolet/infrared approaches [24] and therefore, in the case of the TIM the form factor equations are expected

---

[1]For a more elegant argument related to the spontaneously symmetry breaking of supersymmetry – present at the critical point of the TIM but broken along this massless renormalisation group flow – see [16].

to have *two* different solutions for the $\mathbb{Z}_2$ odd operators of the theory, and we demonstrate that this is indeed the case. In addition, while in a generic theory the parameters which label the different solutions of the form factor equations are expected to remain undetermined within the framework of form factor bootstrap theory, in the peculiar case of the TIM the self-duality of the model and the cluster property of the form factors can be exploited to determine these matrix elements exactly. Therefore, the conclusion of our analysis is that in the magnetisation sector relative to the thermal deformation of the TIM the only freedom consists of the vacuum expectation values (VEV) of the disorder operators, quantities which are essentially related to the overall normalisation of these fields and, in the conformal normalisation the exact value of the VEV are known [26]. The validity of the form factor solutions constructed for the order and disorder operators can be subjected to a stringent (and successful) test by comparing them to results coming from the numerical diagonalisation of the Hamiltonian of the model on a cylinder geometry by means of the truncated conformal space approach (TCSA) originally introduced in [27].

One of the most important applications of form factors is the construction of time and space dependent correlation functions, whose Fourier transform yields the so-called dynamical structure factors (DSFs), which can be accessed experimentally by inelastic neutron scattering or Raman scattering. These are computed through a Lehmann expansion and because of the presence of energy thresholds in a gapped theory, are exact at low energies [28–30]. Such computations have already been carried out for the critical Ising field theory perturbed by a magnetic field. This theory is described by Zamolodchikov's $E_8$ scattering theory [31] and the exact form factors were computed, for the spin operator, by Delfino and Mussardo in [32] and, for the energy operator, by Delfino and Simonetti in [33]. It has realizations in terms of the quasi-one-dimensional (1D) ferromagnetic material $CoNb_2O_6$ [34,35] and the Ising chain antiferromagnet $BaCo_2V_2O_8$ [36,37]. The latter realisation allows experimental access to the dynamical structure factor, which can also be computed from the exact form factor solution of the $E_8$ scattering theory [38]. The form factors obtained here for the $E_7$ model allow us to perform a similar calculation of the dynamical structure factors, which promises an even richer experimental signature due to the existence of two independent order/disorder fields.

The paper is organised as follows. Section 2 reviews the duality of the Ising model, related to the presence of a Majorana free field in the operator content of this model: the zero mode of this field gives rise to a pair of order and disorder operators with the same conformal dimension. In this section we also briefly review the exact expressions of the form factors of these operators and use them to compute the DSF of the order/disorder operators. In Section 3 we discuss the various formulations of the universality class of the tricritical Ising model, in particular, those relative to a $\varphi^6$ Landau-Ginzburg Lagrangian and to a supersymmetric field theory. The different Ramond representations of the superconformal algebra and the zero mode of the analytic fermionic field $G(z)$ may be regarded as the origin of the two different pairs of fields dual to each other, $(\sigma, \mu)$ and $(\sigma', \mu')$. Section 4 addresses a few basic features of the integrable thermal deformation of the TIM, in particular the associated exact $S$ matrix and the rich spectrum of massive excitations in close correspondence with the exceptional algebra $E_7$. Here we also discuss the generic renormalization group flows which link the classes of universality of the TIM and the IM, arguing that the genuine physics related to the $E_7$ structure requires a proper fine tuning of the coupling constants of the off-critical action. In Section 5, we present in detail the various steps which lead us to the exact determination of the lowest form factors of order and disorder operators in the TIM, together with the symmetries behind these matrix elements and the family of solutions which originate from the recursive equations which they satisfy. In this section we also show how, employing the duality and the cluster property of the form factors, it is possible to pin down the full expressions of the form factors in terms only of the vacuum expectation values (VEV) of the disorder operators. Section 6 presents a thorough

Table 2.1: Kac table of the first minimal unitary model of CFT, corresponding to the Ising model. Irreducible representation of the Virasoro algebra are labeled by indices $r = 1, 2$ and $s = 1, 2, 3$.

| 2 | $\frac{1}{2}$ | $\frac{1}{16}$ | 0 |
|---|---|---|---|
| 1 | 0 | $\frac{1}{16}$ | $\frac{1}{2}$ |
| $r \diagdown s$ | 1 | 2 | 3 |

check of the analytic expressions obtained for the form factors of the magnetisation operators in the TIM by means the truncated conformal space approach, a method which permits not only to numerically diagonalise the Hamiltonian of the model on a cylinder geometry but also to extract the finite-volume expressions of various matrix elements. In Section 7, we discuss the dynamical structure factors of the order/disorder operators in the TIM and we comment on their main differences with respect to the analogous quantities in the IM. Our conclusions are finally gathered in Section 8. The paper also contains several important appendices where we collect the main equations satisfied by the form factors of an integrable two-dimensional field theory, the exact amplitudes of the $S$-matrix of the thermal deformation of the TIM, the exact values of the on-shell three-particle vertices, needed details of the implementation of the TCSA, and necessary details of the spectral representation of the correlation functions together with the dynamical structure factors.

## 2 Duality in the Ising Model

Given the close similarity between the tricritical Ising model and the Ising model, it is useful first to discuss the emergence of the duality of the Ising model, in particular its origin from its underlying conformal field theory and the corresponding operator product expansion (OPE). This is particularly useful in light of the fermionic formulation of the model. After showing the existence of a pair of dual order/disorder operators, we also briefly present their exact form factors and the corresponding correlation functions, in particular the dynamical structure factors of the Ising model. The markedly different behavior of these functions with respect to the analogous functions of the tricritical Ising model (compare Figs (2.1a - 2.1b) of the Ising model with Figs (7.1 - 7.2 - 7.3) of the tricritical Ising model) is expected to be very useful in discriminating between these two classes of universality from an experimental point of view.

### 2.1 Conformal Field Theory of the Ising Model

At the critical point, the Ising model is described by the first minimal unitary conformal model [4] with central charge $c = 1/2$ and the Kac table of the conformal dimensions reported in the table below. In order to discuss the operator content of this theory, let us introduce the

notation[2]

$$\mathbf{1} = (0,0), \qquad \psi = \left(\frac{1}{2}, 0\right), \qquad \overline{\psi} = \left(0, \frac{1}{2}\right),$$
$$\epsilon = \left(\frac{1}{2}, \frac{1}{2}\right), \qquad \sigma = \left(\frac{1}{16}, \frac{1}{16}\right), \qquad \mu = \left(\frac{1}{16}, \frac{1}{16}\right). \tag{2.1}$$

For our purposes, it is important to notice the presence of the fermionic fields $\psi$ and $\overline{\psi}$, which are the analytic and the anti-analytic components of the two-dimensional Majorana fermion field $\Psi = (\psi, \overline{\psi})$. As is well known, the critical action of the Ising model can be written[3]

$$\mathcal{A} = \frac{1}{2\pi} \int d^2 x \left[ \psi \, \partial_{\overline{z}} \psi + \overline{\psi} \, \partial_z \overline{\psi} \right]. \tag{2.2}$$

This action is quadratic and therefore free. The corresponding equations of motion,

$$\partial_z \overline{\psi} = 0, $$
$$\partial_{\overline{z}} \psi = 0, \tag{2.3}$$

show that the two spinor components are decoupled and, moreover, that $\psi$ depends only on $z$ and $\overline{\psi}$ only on $\overline{z}$. The analytic and anti-analytic components of the stress-energy tensor which accompany the action (2.2) are

$$T = -\frac{1}{2} : \psi \partial_z \psi :, \qquad \overline{T} = -\frac{1}{2} : \overline{\psi} \partial_{\overline{z}} \overline{\psi} :, \tag{2.4}$$

with the associated Virasoro algebra given by

$$[L_n, L_m] = (n-m)L_{n+m} + \frac{c}{12} n(n^2 - 1)\delta_{n+m,0}, \qquad c = \frac{1}{2}. \tag{2.5}$$

Focusing on the analytic sector alone, the OPE of $\psi$ with itself is given by

$$\psi(z_1)\psi(z_2) = \frac{1}{z_1 - z_2} + \cdots. \tag{2.6}$$

The mode expansion of the Taylor–Laurent series reads

$$\psi(z) = \sum_{n=-\infty}^{\infty} \frac{\psi_n}{z^{n+1/2}}, \tag{2.7}$$

where

$$\psi_n = \oint_C \frac{dz}{2\pi i} z^{n-1/2} \psi(z), \tag{2.8}$$

with a closed contour $C$ around the origin. Using eqn. (2.6), the anti-commutation relations of the modes can be derived using the operator product expansion and exchanging the order of the contours around the origin:

$$\begin{aligned}
\{\psi_n, \psi_m\} &= \left[ \oint \frac{dz}{2\pi i}, \oint \frac{dw}{2\pi i} \right] z^{n-1/2} w^{m-1/2} \psi(z)\psi(w) \\
&= \oint \frac{dw}{2\pi i} w^{m-1/2} \oint \frac{dz}{2\pi i} z^{n-1/2} \frac{1}{z-w} \\
&= \oint \frac{dw}{2\pi i} w^{m+n-1} = \delta_{n+m,0}.
\end{aligned} \tag{2.9}$$

---

[2]$(\Delta, \overline{\Delta})$ are the conformal weights provided by the Kac table and each pair identifies the physical operator obtained by combining together the analytic and anti-analytic parts.

[3]In the following $z = x_1 + i x_2$ and $\overline{z} = x_1 - i x_2$.

## 2.2 Neveu-Schwarz and Ramond Sectors

The fermion field $\psi(z)$ can satisfy two different monodromy properties since it is naturally defined on the double covering of the complex plane with a branch cut starting from a point, here assumed to be the origin:

$$\psi(e^{2\pi i} z) = \pm \psi(z) \,. \tag{2.10}$$

The first case defines the so-called Neveu-Schwarz (NS) sector, while the second defines the so-called Ramond (R) sector. In the Neveu-Schwarz sector, the mode expansion of the field is given in terms of half-integer indices, while in the Ramond sector the indices $n$ of the (2.7) are instead integers

$$\begin{aligned}
\psi(e^{2\pi i} z) &= \psi(z), & n \in \mathbb{Z} + \tfrac{1}{2}\,, & \text{(NS)} \\
\psi(e^{2\pi i} z) &= -\psi(z), & n \in \mathbb{Z}\,, & \text{(R)}\,.
\end{aligned} \tag{2.11}$$

It is also convenient to introduce the operator $(-1)^F$, where $F$ is the fermionic number, defined in terms of its anti-commutation with the field $\psi$

$$(-1)^F \psi(z) = -\psi(z)(-1)^F \,.$$

This operator satisfies $\left((-1)^F\right)^2 = 1$ and

$$\left\{(-1)^F, \psi_n\right\} = 0 \quad , \forall n \,. \tag{2.12}$$

Let's focus the attention on the Ramond sector, i.e. when the field satisfies the anti-periodic boundary conditions. In such a case, it is necessary to take into account the presence of the zero mode of the field that satisfies

$$\{\psi_0, \psi_0\} = 1\,, \qquad \{(-1)^F, \psi_0\} = 0\,. \tag{2.13}$$

Applying $\psi_0$ to an eigenstate of $L_0$ does not change its eigenvalue, which means that the ground state of the Ramond sector must realise a representation of the two-dimensional algebra given by $\psi_0$ and $(-1)^F$. The smallest irreducible representation consists of a doublet of degenerate operators $\sigma$ and $\mu$, the so-called *order and disorder operators*, with the same conformal weight. In this space, a $2 \times 2$ matrix representation of $\psi_0$ and $(-1)^F$ is given by

$$\psi_0 = \frac{1}{\sqrt{2}} \begin{pmatrix} 0 & 1 \\ 1 & 0 \end{pmatrix}, \qquad (-1)^F = \begin{pmatrix} 1 & 0 \\ 0 & -1 \end{pmatrix}. \tag{2.14}$$

In this representation the fields $\sigma$ and $\mu$ are eigenvectors of $(-1)^F$ with eigenvalue $+1$ and $-1$ respectively.

The OPE of the fermionic field with the order/disorder fields is given by

$$\psi(z)\sigma(w) = \frac{1}{\sqrt{2}} (z-w)^{-1/2} \mu(w) + \cdots ; \quad \psi(z)\mu(w) = \frac{1}{\sqrt{2}} (z-w)^{-1/2} \sigma(w) + \cdots \tag{2.15}$$

Note that there is a square root branch cut which changes sign if the fermion field is transported around the location of $\sigma$ or $\mu$; the latter can be interpreted as defect creating operators changing the fermion boundary condition. Therefore the two-point correlation function of the field $\psi(z)$ with anti-periodic boundary conditions can be interpreted as the two-point correlation of $\psi(z)$ but in the presence of two $\sigma$ or $\mu$ fields, placed at the origin and at infinity respectively

$$\langle \psi(z)\psi(w) \rangle_A \equiv \langle \sigma | \psi(z)\psi(w)\sigma(0) | 0 \rangle = \langle \mu | \psi(z)\psi(w)\mu(0) | 0 \rangle \,, \tag{2.16}$$

where

$$\langle\sigma| = \lim_{z\to\infty}\langle 0|\sigma(z)|z|^{1/4}\,, \tag{2.17}$$

$$\langle\mu| = \lim_{z\to\infty}\langle 0|\mu(z)|z|^{1/4}\,. \tag{2.18}$$

These correlators can be computed using the expansion in the integer modes of $\psi$: separating the zero mode and using $\psi_0^2 = 1/2$ yields

$$
\begin{aligned}
\langle\psi(z)\psi(w)\rangle_A &= \langle\sum_{n=0}\psi_n z^{-n-1/2}\sum_{m=0}^{-\infty}\psi_m w^{-m-1/2}\rangle_A\\
&= \sum_{n=1}^{\infty}z^{-n-1/2}w^{n-1/2} + \frac{1}{2}\frac{1}{\sqrt{z\,w}}\\
&= \frac{1}{\sqrt{zw}}\left(\frac{w}{z-w}+\frac{1}{2}\right) = \frac{1}{2}\frac{\sqrt{\frac{z}{w}}+\sqrt{wz}}{z-w}\,.
\end{aligned}
\tag{2.19}
$$

In the limit $z \to w$, this result reproduces the Operator Product Expansion (2.6), which is expected since the OPE expresses a *local* property of the field and so it is insensitive to the boundary conditions.

In order to compute the conformal dimensions of the fields $\sigma$ and $\mu$, one can use the Ward identity

$$T(z)\sigma(0)|0\rangle = \frac{\Delta_\sigma}{z^2}\sigma(0)|0\rangle + \cdots,$$

that leads to

$$\langle T(z)\rangle_A \equiv \langle\sigma|T(z)\sigma(0)|0\rangle = \frac{\Delta_\sigma}{z^2}\,. \tag{2.20}$$

The left-hand side of this equation can be evaluated using both the definition of the normal order

$$T(z) = \lim_{\eta\to 0}\frac{1}{2}\left(-\psi(z+\eta)\partial_z\psi(z)+\frac{1}{\eta^2}\right) \tag{2.21}$$

and the correlation function (2.19). Hence,

$$\langle T(z)\rangle_A = \lim_{w\to z}\left[-\frac{1}{4}\partial_w\left(\frac{\sqrt{z/w}+\sqrt{w/z}}{z-w}\right)+\frac{1}{2(z-w)^2}\right] = \frac{1}{16z^2}\,, \tag{2.22}$$

and so

$$\Delta_\sigma = \Delta_\mu = \frac{1}{16}\,. \tag{2.23}$$

The full OPE algebra of the local scalar fields of magnetisation and energy operators (hereafter written by omitting the structure constants and the dependence on the coordinates) is

$$
\begin{aligned}
\sigma\,\sigma &= \mathbf{1}+\epsilon\,,\\
\epsilon\,\epsilon &= \mathbf{1}\,,\\
\epsilon\,\sigma &= \sigma\,.
\end{aligned}
\tag{2.24}
$$

An equivalently set of mutually local scalar fields is given by the disorder and the energy operators, with the OPE algebra

$$
\begin{aligned}
\mu\,\mu &= \mathbf{1}+\epsilon\,,\\
\epsilon\,\mu &= \mu\,,\\
\epsilon\,\epsilon &= \mathbf{1}\,.
\end{aligned}
\tag{2.25}
$$

These algebras highlight that the Ising model has two independent $\mathbb{Z}_2$ spin symmetries:

- one of them flips the sign of the order operator

$$\sigma \to -\sigma\,, \qquad \mu \to \mu\,, \tag{2.26}$$

- while the other flips the sign of the disorder field

$$\sigma \to \sigma\,, \qquad \mu \to -\mu\,, \tag{2.27}$$

$\epsilon$ is even under both i.e., $\epsilon \to \epsilon$.

Moreover, at its critical point the Ising model is also invariant under the Kramers–Wannier (KW) duality transformation, under which $\epsilon \leftrightarrow -\epsilon$ and $\sigma \leftrightarrow \mu$. The odd parity of $\epsilon$ under the duality transformation naturally explains the absence of $\epsilon$ in the operator product expansion of this field with itself. The KW duality also enlightens an important property of the aforementioned $\mathbb{Z}_2$ spin symmetries of the Ising model which can be however spontaneously broken:

(i) for the high-temperature phase $T > T_c$, the $\mathbb{Z}_2$ spin symmetry $\sigma \to -\sigma$ on the order operator is exact while the $\mathbb{Z}_2$ spin symmetry $\mu \to -\mu$ on the disorder operator is spontaneously broken, therefore for $T > T_c$, we have a non-zero vacuum expectation value $\langle \mu \rangle$;

(ii for the low-temperature phase $T < T_c$, the $\mathbb{Z}_2$ symmetry of the order parameter $\sigma$ is spontaneously broken while the $Z_2$ symmetry of the disorder operator is exact, therefore for $T < T_c$ we have a non-zero vacuum expectation value $\langle \sigma \rangle$.

Notice, however, that at the critical point $T = T_c$ both vacuum expectation values vanish, $\langle \sigma \rangle = \langle \mu \rangle = 0$.

As a final remark, we observe that the OPE algebra (2.24) of the scalar fields can be also interpreted as the algebra of the composite operators of a $\varphi^4$ Landau-Ginzburg theory – a theory notoriously associated to the class of universality of the Ising model [39]. Choosing the field identification $\sigma \equiv \varphi$, the operator product expansion yields $: \varphi^2 := \epsilon$ and $: \varphi^3 := \partial_z \partial_{\bar{z}} \varphi$. Hence, the conformal model can be seen as the exact fixed-point solution of the field theory associated to the Lagrangian

$$\mathcal{L} = \frac{1}{2}(\partial_\mu \varphi)^2 + g \varphi^4\,. \tag{2.28}$$

## 2.3 Form Factors of the Order and Disorder Operators at $T \neq T_c$

Adding a mass term to the fermionic action (2.2) gives the Ising model away from its critical temperature: this is equivalent to coupling the system to the energy operator $\epsilon(x)$ which is $\mathbb{Z}_2$ odd under duality. The resulting quantum field theory turns out to be integrable and its corresponding form factors can be computed exactly [18, 40]. Due to the self-duality of the model, we can discuss equivalently the case $T > T_c$ or $T < T_c$. We choose to focus our attention on the high-temperature phase where there is no spontaneous symmetry breaking of the $\mathbb{Z}_2$ spin symmetry: in this case there is a unique ground state and only one massive particle excitation $A$ in the spectrum, with a two-body $S$-matrix given by $S = -1$. The particle $A$ can be considered as created by the magnetisation operator $\sigma(x)$, therefore it is odd under the $\mathbb{Z}_2$ symmetry of the Ising model, with its mass given by $m = |T - T_c|$.

In the following we consider (semi)local operator fields $\Phi(x)$, which can be fully characterised by their form factors

$$F^{\Phi}_{a_1,\dots,a_n}(\theta_1,\dots,\theta_n) = \langle 0|\Phi(0)|A_{a_1}(\theta_1),\dots,A_{a_n}(\theta_n)\rangle\,, \tag{2.29}$$

(a short overview of their properties is given in Appendix A).

In the high-temperature phase, the order parameter $\sigma(x)$ is odd under the $\mathbb{Z}_2$ symmetry while the disorder operator $\mu(x)$ is even. Hence, $\sigma(x)$ has matrix elements on states with an odd number of particles, $F^\sigma_{2n+1}$, whereas $\mu(x)$ on those with an even number of particles, $F^\mu_{2n}$. Their form factors can be obtained by solving the form factor equations (A.2) and (A.3). In the case of the the residue equations (A.3) relative to the kinematical poles, it is necessary to take into account that the operator $\mu$ has a semi-local index equal to $1/2$ with respect to the operator $\sigma(x)$ that creates the asymptotic states. Denoting by $F_n$ the form factors of these operators (for $n$ even they refer to $\mu(x)$ while for $n$ odd to $\sigma(x)$), we obtain the recursive equation

$$-i \lim_{\widetilde{\theta} \to \theta} (\widetilde{\theta} - \theta) F_{n+2}(\widetilde{\theta} + i\pi, \theta, \theta_1, \theta_2, \ldots, \theta_{2n}) = 2 F_n(\theta_1, \ldots, \theta_{2n}), \qquad (2.30)$$

for both $n$ odd/even.

These equations admit an infinite number of solutions, that can be obtained by including all possible kernel solutions at each level [25]. The minimal solution corresponds to the form factors of the order and disorder operators

$$F_n(\theta_1, \ldots, \theta_n) = H_n \prod_{i<j}^n \tanh \frac{\theta_i - \theta_j}{2}. \qquad (2.31)$$

The normalisation coefficients satisfy the recursive equation

$$H_{n+2} = i H_n.$$

The solutions with $n$ even are therefore fixed by choosing $F_0 = H_0$, namely by the non-zero value of the vacuum expectation of the disorder operator

$$F_0 = \langle 0|\mu(0)|0\rangle = \langle \mu\rangle, \qquad (2.32)$$

while those with $n$ odd are determined by the real constant $F_1$ relative to the one-particle matrix element of $\sigma(x)$

$$F_1 = \langle 0|\sigma(0)|A(0)\rangle. \qquad (2.33)$$

Imposing the cluster property (A.23) gives the relation

$$F_1^2 = i F_0^2, \qquad (2.34)$$

therefore *all* form factors of the order/disorder operators of the Ising model are fixed up to the overall normalisation of the operators given by $F_0$. Adopting the conformal normalisation of both operators

$$\langle \sigma(x)\sigma(0)\rangle = \langle \mu(x)\mu(0)\rangle \simeq \frac{1}{|x|^{1/4}}, \qquad |x| \to 0 \qquad (2.35)$$

the vacuum expectation value $F_0$ is known exactly and is given by [26]

$$F_0 = 2^{1/12} e^{-1/8} A^{3/2} m^{1/8}, \qquad (2.36)$$

where $A = 1.282427..$ is Glaisher's constant.

The identification of the FF of the order/disorder operators can be confirmed by the $\Delta$-sum rule (A.26). To this aim we need the FF of the stress-energy tensor, given by

$$F^\Theta_{2n}(\theta_1, \ldots, \theta_{2n}) = \begin{cases} -2\pi i\, m^2 \sinh \dfrac{\theta_1 - \theta_2}{2} & , \quad n = 2 \\ 0 & , \quad \text{otherwise} \end{cases} \qquad (2.37)$$

where we used the normalisation of the trace operator $F_2^\Theta(i\pi) = 2\pi m^2$, Using now the matrix elements of $\mu(x)$ and $\Theta(x)$, their correlator can be computed in a closed form

$$
\begin{aligned}
\langle\Theta(r)\mu(0)\rangle &= \frac{1}{2}\int\frac{d\theta_1}{2\pi}\frac{d\theta_2}{2\pi}F^\Theta(\theta_{12})\overline{F}^\mu(\theta_{12})e^{-mr(\cosh\theta_1+\cosh\theta_2)} \\
&= -m^2\langle\mu\rangle\left[\frac{e^{-2mr}}{2mr}+Ei(-2mr)\right],
\end{aligned}
\tag{2.38}
$$

where

$$
Ei(-x) = -\int_x^\infty\frac{dt}{t}e^{-t}.
$$

Substituting this correlation function in the $\Delta$-theorem (A.26) yields the correct value for the conformal dimension of the disorder operator

$$
\Delta_\mu = -\frac{1}{2\langle\mu\rangle}\int_0^\infty dr\, r\langle\Theta(r)\mu(0)\rangle = \frac{1}{4\pi}\int_0^\infty d\theta\,\frac{\sinh^2\theta}{\cosh^3\theta} = \frac{1}{16}.
\tag{2.39}
$$

While all the considerations above referred to the high temperature phase, the form factors in the low temperature phase can be obtained by a simple application of Kramers–Wannier duality. In the low-temperature phase the identification of the order and disorder fields must be swapped, and a calculation identical to the above evaluation of the $\Delta$-theorem sum rule gives the conformal dimension of the order operator $\sigma$.

### 2.4 Dynamical Structure Factors of the Ising Model

Here we compute the dynamical structure factors corresponding to order/disorder correlation functions:

$$
\begin{aligned}
\mathcal{S}^{\sigma\sigma}(\omega,q) &= \int dxdt\, e^{i\omega t-iqx}\langle\sigma(x,t)\sigma(0,0)\rangle, \\
\mathcal{S}^{\mu\mu}(\omega,q) &= \int dxdt\, e^{i\omega t-iqx}\langle\mu(x,t)\mu(0,0)\rangle.
\end{aligned}
\tag{2.40}
$$

We consider the frequency dependence at zero-momentum transfer ($q=0$), which determines the full structure factor due to Lorentz invariance, and compute it using a spectral expansion in terms of form factors as described in Appendix D.

Let us concentrate on the high-temperature phase. Here $\mu$ has only even-particle matrix elements and $\sigma$ has only odd-particle ones. Therefore truncating the expansion states with up to three particles gives

$$
\mathcal{S}^{\mu\mu}(\omega,q=0) = \mathcal{S}_2^{\mu\mu}(\omega,q=0)+\dots
\tag{2.41}
$$
$$
\mathcal{S}^{\sigma\sigma}(\omega,q=0) = \mathcal{S}_1^{\sigma\sigma}(\omega,q=0)+\mathcal{S}_3^{\sigma\sigma}(\omega,q=0)+\dots
\tag{2.42}
$$

For the disorder operator substituting the form factors from Subsection 2.3 into (D.6) gives the two-particle contribution

$$
\mathcal{S}_2^{\mu\mu}(\omega,q=0) = \frac{F_0^2}{\omega^3}\sqrt{\omega^2-4m^2}\,\Theta(\omega-2m),
\tag{2.43}
$$

which has a square root behaviour $\propto\sqrt{\omega-2m}$ at threshold.

For the order operator (D.5) gives the following one-particle contribution:

$$
\mathcal{S}_1^{\sigma\sigma}(\omega,q=0) = \frac{2\pi F_0^2}{m}\delta(\omega-m).
\tag{2.44}
$$

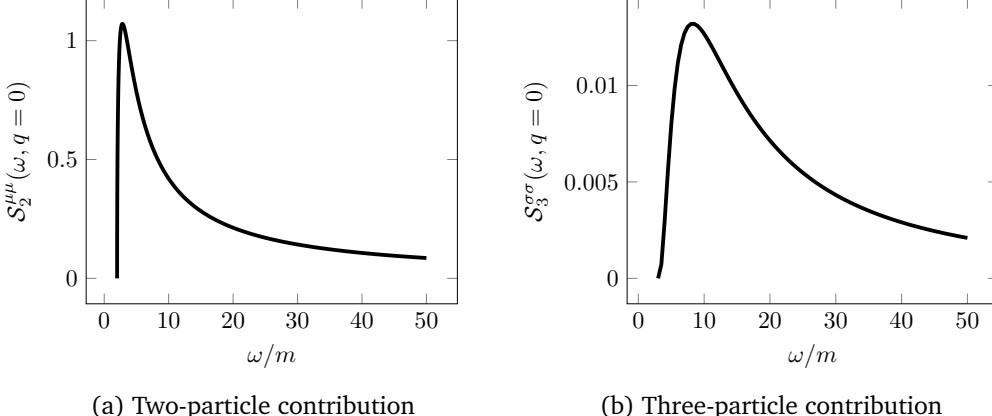

(a) Two-particle contribution
(b) Three-particle contribution

Figure 2.1: Two- and three particle contributions to the dynamical structure factor of disorder (a) and order operators (b) in the Ising model. The values of the dynamical structure factors are shown in units of the mass gap $m$ using the expression (2.36) for $F_0$.

The three-particle contribution (D.8) is a bit more complicated, but after some manipulations it can be evaluated as

$$\mathcal{S}_3^{\sigma\sigma}(\omega, q = 0) = \frac{F_0^2}{6\pi m^2} \int_{\theta_{12} > 0} \frac{d\theta_3}{|\sinh\theta_{12}|} \prod_{i<j}^3 \tanh^2\frac{\theta_{ij}}{2}, \qquad (2.45)$$

where integration is performed over the branch $\theta_{12} > 0$ and the identical contribution from the other branch is taken into account by including a factor of 2. The behaviour of the three-particle contribution at the threshold is given by

$$\mathcal{S}_3^{\sigma\sigma}(\omega, k = 0) = \frac{F_0^2}{192\sqrt{3}m^5}(\omega - 3m)^3 + O((\omega - 3m)^4). \qquad (2.46)$$

The functions $\mathcal{S}_2^{\mu\mu}$ and $\mathcal{S}_3^{\sigma\sigma}$ are shown in Figures 2.1a and 2.1b.

# 3 Duality in the Tricritical Ising Model

Let us now briefly discuss the class of universality and the duality symmetry of the tricritical Ising model (TIM) associated to the second unitary minimal model: the central charge is $c = \frac{7}{10}$, while the conformal dimensions are shown in Table 3.1. We are mainly concerned here with two equivalent quantum field theory formulations of the TIM, namely one based on a Landau-Ginzburg field theory, which explicates the $\mathbb{Z}_2$ spin symmetry, while the other based on supersymmetry, which clarifies the origin of the $\mathbb{Z}_2$ duality symmetry of the model. We discuss these two formulations separately.

## 3.1 Landau–Ginzburg Description

The order-parameters of the TIM, as present in the Blume–Capel Hamiltonian (1.2), can be put in a one-to-one correspondence with the operator content of a $\varphi^6$ Landau-Ginzburg Lagrangian based on a scalar field $\varphi$ and the Euclidean action given by:

$$S = \int d^2x \left[ \frac{1}{2}(\partial_\mu\varphi)^2 + g_1\varphi + g_2\varphi^2 + g_3\varphi^3 + g_4\varphi^4 + \varphi^6 \right], \qquad (3.1)$$

Table 3.1: The Kac table of the conformal dimensions of the TIM, labeled by indices $1 \leq r \leq 3$ and $1 \leq s \leq 4$.

| 3 | $\frac{3}{2}$ | $\frac{3}{5}$ | $\frac{1}{10}$ | 0 |
|---|---|---|---|---|
| 2 | $\frac{7}{16}$ | $\frac{3}{80}$ | $\frac{3}{80}$ | $\frac{7}{16}$ |
| 1 | 0 | $\frac{1}{10}$ | $\frac{3}{5}$ | $\frac{3}{2}$ |
| $r\diagdown s$ | 1 | 2 | 3 | 4 |

where the tricritical point[4] is identified by the condition $g_1 = g_2 = g_3 = g_4 = 0$ [39]. The statistical interpretation of the coupling constants is the following: $g_1$ plays the role of an external magnetic field $h$, $g_2$ measures the displacement of the temperature from its critical value, i.e. $g_2 \sim (T - T_c)$, $g_3$ may be regarded as a sub-leading magnetic field $h'$ and, finally, $g_4$ may be interpreted as a chemical potential for the vacancies. Switching on and tuning the various coupling constants, the model changes its spectrum and its dynamics, as previously studied in a series of papers [16, 19–23, 41–50].

The exact conformal weights of the scaling fields of the model are given by

$$\Delta_{r,s} = \frac{(5r - 4s)^2 - 1}{80} \quad , \quad \begin{array}{l} 1 \leq r \leq 3, \\ 1 \leq s \leq 4. \end{array} \tag{3.2}$$

The six scalar primary fields of the TIM perfectly match the identification provided by the composite fields of the Landau–Ginzburg theory and by the symmetries of the model. In fact, with respect to the $\mathbb{Z}_2$ spin symmetry of the model $\varphi \to -\varphi$, the fields are classified as follows[5] (c.f. Table 3.2):

1. Two odd fields: the magnetisation operator $\sigma \equiv \varphi$ and the sub-leading magnetic operator $\sigma' \equiv \, : \varphi^3 :$;

2. Four even fields: the identity operator $\mathbf{1}$, the energy operator $\epsilon \equiv \, : \varphi^2 :$, and the density operator $t \equiv \, : \varphi^4 :$, associated to the vacancies. Finally, there is also the irrelevant field $\epsilon''$. The operator product expansion of these fields gives rise to a sub-algebra of the fusion rules.

Out of the six scalar primary fields, four of them are relevant operators: the OPE algebra and the relative structure constants, which are useful to implement the truncated conformal space approach, are reported in Table 3.3.

---

[4]A tricritical point occurs when a second order phase transition line meets a first order phase transition line. For the Lagrangian (3.1), at the tree level the curve which describes the second order phase transition is identified by $g_1 = g_2 = g_3 = 0$, but $g_4 > 0$; the curve which describes the first order phase transition is given instead by $g_1 = g_3 = 0$, with $g_2 > 0$ and $g_4 = -2\sqrt{g_2}$.

[5]For simplicity, in the following we use for the fermion fields and the leading order/disorder operators of the TIM the same notations as in the IM, even though it is evident that they are different fields.

Table 3.2: Primary fields of the TIM and their Landau-Ginzburg identifications

| conformal weights | field | physical role | Landau-Ginzburg field |
|---|---|---|---|
| $(0,0)$ | $\mathbb{I}$ | | identity |
| $(\frac{3}{80}, \frac{3}{80})$ | $\sigma$ | magnetisation | $\Phi$ |
| $(\frac{1}{10}, \frac{1}{10})$ | $\epsilon$ | energy | $:\Phi^2:$ |
| $(\frac{7}{16}, \frac{7}{16})$ | $\sigma'$ | submagnetisation | $:\Phi^3:$ |
| $(\frac{3}{5}, \frac{3}{5})$ | $t$ | chemical potential | $:\Phi^4:$ |
| $(\frac{3}{2}, \frac{3}{2})$ | $\epsilon''$ | (irrelevant) | $:\Phi^6:$ |

Table 3.3: OPE and structure constants of primary fields in the TIM.

*even* $*$ *even*

$\epsilon * \epsilon = [1] + c_1[t]$

$t * t = [1] + c_2[t]$

$\epsilon * t = c_1[\epsilon] + c_3[\epsilon'']$

$$c_1 = c_2 = \frac{2}{3}\sqrt{\frac{\Gamma\left(\frac{4}{5}\right)\Gamma^3\left(\frac{2}{5}\right)}{\Gamma\left(\frac{1}{5}\right)\Gamma^3\left(\frac{3}{5}\right)}}$$

*even* $*$ *odd*

$\epsilon * \sigma' = c_4[\sigma]$

$\epsilon * \sigma = c_4[\sigma'] + c_5[\sigma]$

$t * \sigma' = c_6[\sigma]$

$t * \sigma = c_6[\sigma'] + c_7[\sigma]$

$c_3 = \frac{3}{7}$

$c_4 = \frac{1}{2}$

$c_5 = \frac{3}{2}c_1$

$c_6 = \frac{3}{4}$

$c_7 = \frac{1}{4}c_1$

*odd* $*$ *odd*

$\sigma' * \sigma' = [1] + c_8[\epsilon'']$

$\sigma' * \sigma = c_4[\epsilon] + c_6[t]$

$\sigma * \sigma = [1] + c_5[\epsilon] + c_7[t] + c_9[\epsilon'']$

$c_8 = \frac{7}{8}$

$c_9 = \frac{1}{56}$

## 3.2 Supersymmetry and the Kramers-Wannier Duality

We now turn to the second field theoretic representation of the TIM. This representation involves an explicit realisation of a supersymmetry [19–21]. We exploit this supersymmetry to argue for the necessary presence of self-duality in the TIM model.

In two dimensions, super-conformal invariance is associated to two super-currents, $G(z)$ and $\overline{G}(\overline{z})$, where the former is a purely analytic field while the latter is a purely anti-analytic one. They are both fermionic fields, with conformal weights $(\frac{3}{2}, 0)$ and $(0, \frac{3}{2})$ respectively. Notice that $G$ corresponds to the $(r, s) = (1, 4)$ field in the Kac table. The OPE of the field $G(z)$ with itself reads

$$G(z_1)G(z_2) = \frac{2c}{3(z_1 - z_2)^3} + \frac{2}{z_1 - z_2}T(z_2) + \cdots, \tag{3.3}$$

where the parameter $c$ is the same central charge that enters the operator expansion of $T(z)$:

$$T(z_1)T(z_2) = \frac{c}{2(z_1 - z_2)^2} + \frac{2}{(z_1 - z_2)^2}T(z_2) + \frac{1}{z_1 - z_2}\partial T(z_2) + \cdots. \tag{3.4}$$

Since $G(z)$ is also a primary field, it satisfies the operator product expansion:

$$T(z_1)G(z_2) = \frac{3}{2(z_1 - z_2)^2}G(z_2) + \frac{1}{z_1 - z_2}\partial G(z_2) + \cdots. \tag{3.5}$$

Let us define the generators $L_n$ and $G_n$ through the expansions

$$T(z) = \sum_{n=-\infty}^{\infty} \frac{L_n}{z^{2+n}} \, ; \, G(z) = \sum_{m=-\infty}^{\infty} \frac{G_m}{z^{3/2+m}} \,. \tag{3.6}$$

Note that, in the expansion of the field $G(z)$, the indices can assume either integer or half-integer value. In fact, $G(z)$ is a fermionic field and, as discussed in the previous chapter for the free fermionic field $\psi$, it is defined on the double covering of the plane, with a branch cut starting from the origin. Making the analytic continuation, $z \to e^{2\pi i}z$, two possible boundary conditions are permitted:

$$G(e^{2\pi i}z) = \pm G(z). \tag{3.7}$$

In the periodic case (+), termed the Neveu-Schwarz (NS) sector, the indices $m$ are half-integers, $m \in \mathbb{Z} + \frac{1}{2}$. In the anti-periodic case (−), termed the Ramond (R) sector, the indices $m$ are instead integer numbers, $m \in \mathbb{Z}$.

The OPE involving $T(z)$ and $G(z)$ can be equivalently expressed as algebraic relations of their modes, given by the infinite-dimensional supersymmetric algebra:

$$\begin{aligned}
[L_n, L_m] &= (n-m)L_{n+m} + \frac{c}{12}n(n^2-1)\delta_{n+m,0}; \\
[L_n, G_m] &= \frac{1}{2}(n-2m)G_{n+m}; \\
\{G_n, G_m\} &= 2L_{n+m} + \frac{c}{3}\left(n^2 - \frac{1}{4}\right)\delta_{n+m,0}.
\end{aligned} \tag{3.8}$$

The peculiar aspect of this algebra is the simultaneous presence of commutation and anti-commutation relations. The operator content of the TIM can be expressed in terms of the irreducible representations of the above supersymmetric algebra. These representations are either of Neveu-Schwarz and Ramond type as determined by the choice of boundary conditions for the fermionic current $G(z)$.

**Neveu-Schwarz sector**. In the NS sector the only representation besides the vacuum one is given in terms of a superfield,

$$\Phi_l(z, \theta; \overline{z}, \overline{\theta}) = \epsilon(z, \overline{z}) + \theta \, \Psi(z, \overline{z}) + \overline{\theta} \, \overline{\Psi}(z, \overline{z}) + i \, \theta \, \overline{\theta} \, F(z, \overline{z}), \tag{3.9}$$

where $\theta$ and $\overline{\theta}$ are Grassmann variables. The fields $\Psi(z,\overline{z})$ and $\overline{\Psi}(z,\overline{z})$ are interacting fermionic operators with conformal dimensions $\left(\frac{3}{5},\frac{1}{10}\right)$ and $\left(\frac{1}{10},\frac{3}{5}\right)$ respectively, and given by $\Psi = G_{-1/2}\epsilon$, $\overline{\Psi} = \overline{G}_{-1/2}\epsilon$, while $F = G_{-1/2}\overline{G}_{-1/2}\epsilon$. It is easy to see that the Neveu-Schwarz representation employs the $\mathbb{Z}_2$ even fields under spin parity of the TIM[6]. Notice that the equality of the two structure constants $c_1$ and $c_2$ in the OPE of the model (see Table 3.3) comes from $\epsilon(x)$ and $t(x)$ being two different components of the same superfield.

**Ramond sector**. Irreducible representations of the Ramond sectors correspond to the two $\mathbb{Z}_2$ odd fields under the spin symmetry, $\sigma$ and $\sigma'$, the so-called "spin-fields" [21]. However, these fields are both accompanied by their KW dual partners. The origin of these doublets comes from the zero mode $G_0$ of the fermionic field $G(z)$ which satisfies the algebraic relation

$$G_0^2 = L_0 - \frac{c}{24}\,. \tag{3.10}$$

Moreover, as in the Ising model, one can introduce the operator $(-1)^F$, where $F$ is the fermionic number, in terms of its anti-commutation with the field $G$

$$(-1)^F G(z) = -G(z)(-1)^F\,.$$

This operator satisfies $\left((-1)^F\right)^2 = 1$ and

$$\left\{(-1)^F, G_n\right\} = 0 \quad, \forall n. \tag{3.11}$$

In the TIM there is no field whose conformal dimension $\Delta$ satisfies the condition $\Delta = c/24$. Thus the relations (3.9) and (3.10) imply that applying $G_0$ to an eigenstate of $L_0$ does not change its eigenvalue. This means that the ground state of the Ramond sector must realise a representation of the two-dimensional algebra given by $G_0$ and $(-1)^F$. The smallest irreducible representation consists of a doublet of degenerate operators $\sigma$ and its dual companion $\mu$, and the same happens for the sub-leading magnetisation $\sigma'$ which is also accompanied by its dual operator $\mu'$. Hence, in addition to the operators present in the previous Table 3.2, there are also the additional operators shown in Table 3.4.

In the presence of the order/disorder fields, the OPE of the fermionic field $G(z)$ is

$$G(z)\sigma(w) = (z-w)^{-3/2}\mu(w) + \cdots\,; \quad G(z)\mu(w) = \left(\Delta - \frac{c}{24}\right)(z-w)^{-3/2}\sigma(w) + \cdots \tag{3.12}$$

(with $\Delta = 3/80$ and $c = 7/10$) and the same holds replacing $\sigma$ and $\mu$ with $\sigma'$ and $\mu'$ (in the latter case, using in the formula above $\Delta = 7/16$). Similarly to the case of the Ising model, the fields $\sigma$ and $\mu$ are defect operators changing the boundary conditions for fermionic fields, c.f. Eq. (2.15).

We introduce a formal operator $D$ that implements the KW duality, which allows the duality properties of the various fields of the TIM to be summarised as follows:

- the magnetisation order parameters change into the disorder operators:

$$\mu = D^{-1}\sigma D\,, \qquad \mu' = D^{-1}\sigma'D\,. \tag{3.13}$$

- the even fields transform instead in themselves:

$$D^{-1}\epsilon D = -\epsilon\,, \qquad D^{-1}tD = t\,, \qquad D^{-1}\epsilon"D = -\epsilon"\,, \tag{3.14}$$

   i.e., $\epsilon$ and $\epsilon"$ are odd fields, while $t$ is an even field under the duality transformation .

The transformation properties of the various fields of the TIM under the spin-reversal and duality are shown in Table 3.5.

---

[6]The vacancy density operator $t(z,\overline{z})$, with its conformal 2-point function properly normalised to 1, is equal to $F(z,\overline{z}) = 5\,t(z,\overline{z})$, while the fermionic fields, with normalization of their conformal 2-point functions equal to 1, are given by $\Psi(z,\overline{z}) = i\sqrt{5}\,\psi(z,\overline{z})$ and $\overline{\Psi}(z,\overline{z}) = i\sqrt{5}\,\overline{\psi}(z,\overline{z})$.

Table 3.4: Additional operators of the TIM originating from the supersymmetry of the model.

| conformal weights | field | physical role |
|---|---|---|
| $(\frac{3}{80}, \frac{3}{80})$ | $\mu$ | disorder field |
| $(\frac{7}{16}, \frac{7}{16})$ | $\mu'$ | subleading disorder field |
| $(\frac{3}{5}, \frac{1}{10})$ | $\psi$ | fermion |
| $(\frac{1}{10}, \frac{3}{5})$ | $\overline{\psi}$ | anti-fermion |
| $(\frac{3}{2}, 0)$ | $G$ | holomorphic supersymmetry current |
| $(0, \frac{3}{2})$ | $\overline{G}$ | anti-holomorphic supersymmetry current |

Table 3.5: Discrete symmetries of TIM.

| field | spin-reversal | Kramers–Wannier |
|---|---|---|
| $\epsilon$ | $\epsilon$ | $-\epsilon$ |
| $t$ | $t$ | $t$ |
| $\epsilon''$ | $\epsilon''$ | $-\epsilon''$ |
| $\sigma$ | $-\sigma$ | $\mu$ |
| $\sigma'$ | $-\sigma'$ | $\mu'$ |

# 4 Thermal Deformation of the Tricritical Ising Model

## 4.1 $E_7$ Scattering Theory

Now consider perturbing the tricritical Ising model by means of its energy operator $\epsilon(x)$, with the corresponding action given by

$$\mathcal{A} = \mathcal{A}_{TIM} + g \int d^2x\, \epsilon(x)\,, \tag{4.1}$$

where $\mathcal{A}_{TIM}$ is the action of the fixed point. The perturbation with positive/negative coupling constant drives the system into its high/low-temperature phases. While in the latter phase the spin-reversal $\mathbb{Z}_2$ symmetry of the system is spontaneously broken, in the former phase the $\mathbb{Z}_2$ symmetry is unbroken and therefore the corresponding quantum number can be used to label the states. In the low–temperature phase, the massive excitations are given by topologically charged kink states and their neutral bound states, while in the high–temperature phase all excitations are ordinary particle (i.e. topologically trivial) excitations. The two phases are related by a duality transformation and therefore we can restrict our attention to only one of them, which we choose to be the high–temperature phase. The off-critical theory was shown to

be integrable and related to the Toda field theory based on the exceptional algebra $E_7$ [22,23]: the conserved charges have spins

$$s = 1, 5, 7, 9, 11, 13, 17 \pmod{18}, \tag{4.2}$$

(these numbers are the Coxeter exponents of the exceptional algebra $E_7$) and the set of all $S$-matrix amplitudes are reported in Appendix B.

The exact mass spectrum of excitations can be extracted from the pole structure of the $S$ matrices: with respect to the $\mathbb{Z}_2$ spin symmetry of the model, in the high–temperature phase there are three $\mathbb{Z}_2$ odd particle states $A_1, A_3, A_6$ (the ones relative to the masses $m_1$, $m_3$ and $m_6$) and four $\mathbb{Z}_2$ even $A_2, A_4, A_5, A_7$ (those relative to the masses $m_2$, $m_4$, $m_5$ and $m_7$). In the low–temperature phase, the three $\mathbb{Z}_2$ odd particles become kink excitations which interpolate between the two degenerate ground states whereas the four $\mathbb{Z}_2$ even ones correspond to kink-antikink bound states (a.k.a. breathers). This leads, in particular, to an interesting prediction on the universal ratio of the correlation lengths *above* and *below* the critical temperature [48,49]. In fact, identifying the correlation length from the leading exponential asymptotic behaviour of the spin–spin connected correlation function in the long-distance limit

$$\langle 0|\sigma(x)\sigma(0)|0\rangle_c^\pm \sim \exp\left(-\frac{|x|}{\xi^\pm}\right), \tag{4.3}$$

(where the indices $\pm$ refer to the high and low temperature phases respectively), from the $\mathbb{Z}_2$ symmetry property of the $\sigma$ field, the self–duality of the model and the spectral representation of the above correlator it follows that[7]

$$\frac{\xi^+}{\xi^-} = \frac{m_2}{m_1} = 2\cos\frac{5\pi}{18} = 1.28557\ldots \tag{4.4}$$

In fact, in the high-temperature phase the lowest energy state to which the order parameter $\sigma$ couples to is given by the particle $A_1$. To determine the correlation length in the low-temperature phase, using the self-duality of the model, we can consider the correlator of the disorder operator $\mu$ in the high-temperature phase. However, the lowest energy states to which $\mu$ couples is $A_2$, which leads to the non-trivial universal ratio (4.4) of the correlation lengths above and below the critical temperature. It is also worth to remind that in the TIM the relationship between the mass gap and the coupling constant is known exactly exactly and given by [51]:

$$
\begin{aligned}
m_1 &= \left(\frac{2\Gamma\left(\frac{2}{9}\right)}{\Gamma\left(\frac{2}{3}\right)\Gamma\left(\frac{5}{9}\right)}\right)\left(\frac{4\pi^2\Gamma\left(\frac{2}{5}\right)\Gamma^3\left(\frac{4}{5}\right)}{\Gamma^3\left(\frac{1}{5}\right)\Gamma\left(\frac{3}{5}\right)}\right)^{5/18} |g|^{5/9} \\
&= |g|^{5/9} 3.7453728362\ldots.
\end{aligned}
\tag{4.5}
$$

This relation turns out to be useful in our numerical TCSA studies, where it can be used to normalise all units in terms of the lowest mass gap $m_1$ of the theory.

## 4.2 Stability of the $E_7$ Action and the Flow to the Ising Model

In this section we address questions of the stability of the $E_7$ action. Obtaining the $E_7$ symmetry requires fine tuning, which can be seen as follows. First of all, set to 0 any coupling to the $Z_2$ odd sector of the theory, i.e. controlling accurately the presence of any external magnetic

---

[7]Note that for the thermal deformation of the IM the same universal ratio takes instead the value $\xi^+/\xi^- = 2$, since the lowest mass state coupled to the disorder operator is in the IM the two-particle state $AA$.

Table 4.1: Spectrum of the thermal deformation of the tricritical Ising model.

| exact | numerical | parity | excitation |
|---|---|---|---|
| $m_1$ | 1 | odd | kink |
| $m_2 = 2m\cos(5\pi/18)$ | 1.285(6) | even | particle |
| $m_3 = 2m_1\cos(\pi/9)$ | 1.879(4) | odd | kink |
| $m_4 = 2m_1\cos(\pi/18)$ | 1.969(6) | even | particle |
| $m_5 = 4m_1\cos(\pi/18)\cos(5\pi/18)$ | 2.532(1) | even | particle |
| $m_6 = 4m_1\cos(2\pi/9)\cos(\pi/9)$ | 2.879(4) | odd | kink |
| $m_7 = 4m_1\cos(\pi/18)\cos(\pi/9)$ | 3.701(7) | even | particle |

field. Then, at a finite energy cutoff $\Lambda$, the most general action invariant under the spin $Z_2$ symmetry is given by

$$\mathcal{A} = \mathcal{A}_{TIM} + g_\Lambda \int d^2x\, \epsilon(x) + \lambda_\Lambda \int d^2x\, t(x). \tag{4.6}$$

The quantum field theory described by this action (4.6) will generically break the $E_7$ symmetry, which will only be obtained, at any given cutoff $\Lambda$, for a particular choice of $g_\Lambda$ and $\lambda_\Lambda$. When $\Lambda \to \infty$, for the renormalized values of the two couplings this choice consists of a finite value of $g_\infty$ and $\lambda_\infty = 0$ while, at finite cutoff, both couplings need to be non-zero as evidenced by the following OPE:

$$\epsilon(x)\epsilon(y) \simeq \frac{1}{|x-y|^{2/5}} + c_1\,|x-y|^{4/5}t(x) + \cdots. \tag{4.7}$$

This OPE shows that under an RG flow a finite coupling to $\epsilon$ generates a finite coupling to the vacancy density $t$.

At the lowest order, under a rescaling of the cutoff $\Lambda \to (1+1/2dl)\Lambda$, the renormalization group (RG) equations for the two couplings $g_\Lambda$ and $\lambda_\Lambda$ are given by [52]

$$\frac{dg_\Lambda}{dl} \equiv \beta_g(g_\Lambda,\lambda_\Lambda) = 2(1-\Delta_\epsilon)g_\Lambda - 2\pi c_1 g_\Lambda \lambda_\Lambda + \cdots, \tag{4.8}$$
$$\frac{d\lambda_\Lambda}{dl} \equiv \beta_\lambda(g_\Lambda,\lambda_\Lambda) = 2(1-\Delta_t)\lambda_\Lambda - \pi c_1(g_\Lambda^2 + \lambda_\Lambda^2) + \cdots.$$

Of these two couplings, $g_\Lambda$ is the more relevant, therefore it is expected to have the greater influence on the large distance behaviour of the theory. The vector field flows associated to the equations (4.8) in the half-plane $\lambda_\Lambda \geq 0$ are shown in Figure 4.1. These equations have two fixed points:

1. The first, at the origin, $(g_\Lambda, \lambda_\Lambda) = (0,0)$, is identified with the original conformal field theory of central charge $c = 7/10$ of the tricritical Ising model.

2. The second at $(g_\Lambda, \lambda_\Lambda) = (0, 2(1-\Delta_t)/(\pi c_1)) = (0, 0.41...)$ is identified with the $c = 1/2$ Ising model.

In the vicinity of the tricritical point, apart from an overall mass scale, the dynamics of the theory associated to the action (4.6) depends on the dimensionless combination of the

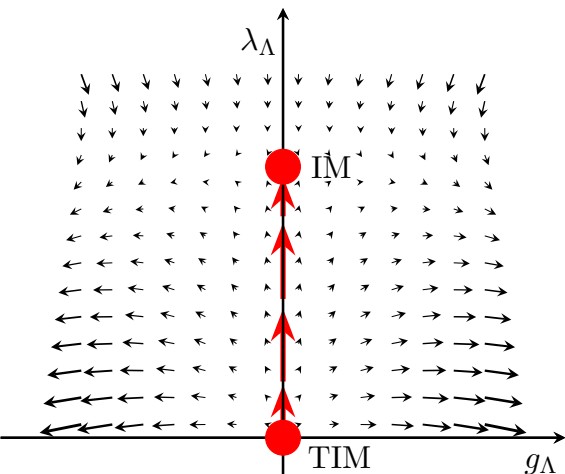

Figure 4.1: Vector fields $(\beta_g, \beta_\lambda)$ in the plane of the two couplings $g_\Lambda$ and $\lambda_\Lambda$.

renormalised coupling constants

$$\eta = \frac{\lambda_\infty}{|g_\infty|^{4/9}}. \tag{4.9}$$

Notice that the red line connecting the two fixed points corresponds to $\eta \to \infty$, i.e. to the renormalised action

$$\mathcal{A} = \mathcal{A}_{TIM} + \lambda_\infty \int d^2x \, t(x), \qquad \lambda_\infty > 0, \tag{4.10}$$

which describes the massless field theory associated to the second order phase transition line previously shown in Figure 1.1. Along such a massless flow to the critical Ising model, the supersymmetry of the original TIM is spontaneously broken[8] [16,44] and therefore the familiar Majorana fermion present in the Ising model (i.e. in the deep infrared of the action (4.10)) can be equivalently interpreted as the Goldstino associated to this spontaneous breaking of supersymmetry.

In conclusion, in regard to the $E_7$ stability, the RG scenario is as follow. In order to observe precisely the $E_7$ dynamics of the TIM (in particular, to observe the singularities induced by *all* excitations of the theory, including those of the higher mass particles, for instance, in the dynamical structure factors discussed in Section 7), it is necessary to fine tune the renormalized value of $\lambda_\infty$ to 0 in the action (4.6).

Any non-zero value of $\lambda_\infty$ induces, on the contrary, a breaking of the integrability of the $E_7$ theory and, correspondingly, a change of the $E_7$ spectrum which may be more or less pronounced depending on the strength of this integrability breaking term (for a detailed analysis of these aspects, see [45]). In the presence of a non-zero value of $\lambda_\infty$, one expects both an adiabatic change of the values of the lowest masses of the $E_7$ particles and decay processes for those having higher masses. Both these effects can be controlled by form factor perturbation theory (FFPT) [53–55]. FFPT involves the form factors of the vacancy density operator $t(x)$ in the $E_7$ scattering theory, namely

$$F^t_{a_1, a_2, \ldots, a_n}(\theta_1, \theta_2, \ldots, \theta_n) = \langle 0 | t(0) | A_{a_1}(\theta_1) A_{a_2}(\theta_2) \ldots A_{a_n}(\theta_n) \rangle. \tag{4.11}$$

We now will estimate the deviations from the $E_7$ values of the masses and decay times of the higher particles, expressing our results in terms of the dimensionless ratio $\eta$ and normalising

---

[8]Setting $\lambda < 0$ gives rise instead to a massive field theory with an exact realization of supersymmetry [42].

the form factors of the operator $t(x)$ accordingly. At the lowest order in $\eta$, the mass $m_i$ of the $i$-th particle gets corrected to

$$m_i(\eta) \simeq m_i + \eta\, \delta m_i\,, \tag{4.12}$$

where

$$\delta m_i = \frac{1}{m_i} F^t_{a_i, a_i}(\theta_1 - \theta_2 = i\pi)\,. \tag{4.13}$$

The decay processes of the higher particles, $A_k \to r_1 A_1 + r_2 A_2 + \cdots + r_l A_l$, where $r_i$ are integers which satisfy,

$$m_k \ge \sum_{i=1}^{l} r_i\, m_i\,, \qquad \sum_{i=1}^{l} r_i = n\,, \tag{4.14}$$

can be computed by Fermi's golden rule,

$$d\Gamma = (2\pi)^2\, \delta^2(P - p_1 - \cdots - p_n)\,|\,T_{fi}\,|^2\,\frac{1}{2E}\prod_{i=1}^{n}\frac{dp_i}{(2\pi)2E_i}\,. \tag{4.15}$$

Here $P$ denotes the 2-momentum of the decaying particle, whereas the amplitude $T_{fi}$ is given at the lowest order in $\eta$ by

$$T_{fi} \simeq \eta\,\langle A_k(P)\,|\,t(0)\,|\,A_1(p_1)\ldots A_1(p_s)A_2(p_{s+1})\ldots A_2(p_n)\rangle\,, \tag{4.16}$$

involving the matrix element of the density operator $t(x)$.

Given the overall $\mathbb{Z}_2$ symmetry of the action (4.6), higher mass particles are expected to decay in channels which respect this symmetry besides satisfying the usual kinematic constraints. For instance, the particle $A_5$ can only decay in the channel $A_5 \to A_1 A_1$, even though the decay process $A_5 \to A_1 A_2$ would be also permitted by kinematics. Similarly, other possible decay processes are

$$A_6 \to A_1 A_2\,;$$
$$A_7 \to A_1 A_1\,;$$
$$A_7 \to A_2 A_2\,;$$
$$A_7 \to A_1 A_3\,;$$
$$A_7 \to A_2 A_4\,;$$
$$A_7 \to A_1 A_1 A_2\,.$$

Let us now discuss in more detail the evolution of the mass of the lowest four particles $A_1, \ldots, A_4$ taking advantage of the actual values of the form factors of the operator $t(x)$ computed in [45] (in units of $m_1$):

$$\begin{aligned} F^t_{11}(i\pi) = -1.28\,; &\qquad F^t_{22}(i\pi) = -1.25\,; \\ F^t_{33}(i\pi) = -2.03\,; &\qquad F^t_{44}(i\pi) = -3.41\,. \end{aligned} \tag{4.17}$$

For small values of $\eta$, the evolution of the mass of the particles $A_1, A_2, A_3$ and $A_4$ is shown in Figure 4.2 where the the evolution of the continuum threshold $2m_1$ is also plotted with a dashed line. There exists a critical value $\eta_1^c \simeq 0.04$ over which the mass of the particle $A_4$ becomes higher than $2m_1$, which leads to decays $A_4 \to A_1 A_1$, and another critical value $\eta_2^c \simeq 0.08$ over which the mass of the particle $A_3$ surpasses $2m_1$. Hence there is an interval of values of the integrability breaking coupling constant $\lambda$ where the four lightest excitations survive with slightly shifted masses. This is an important observation in light of the computation of the dynamical structure factors of the theory presented later.

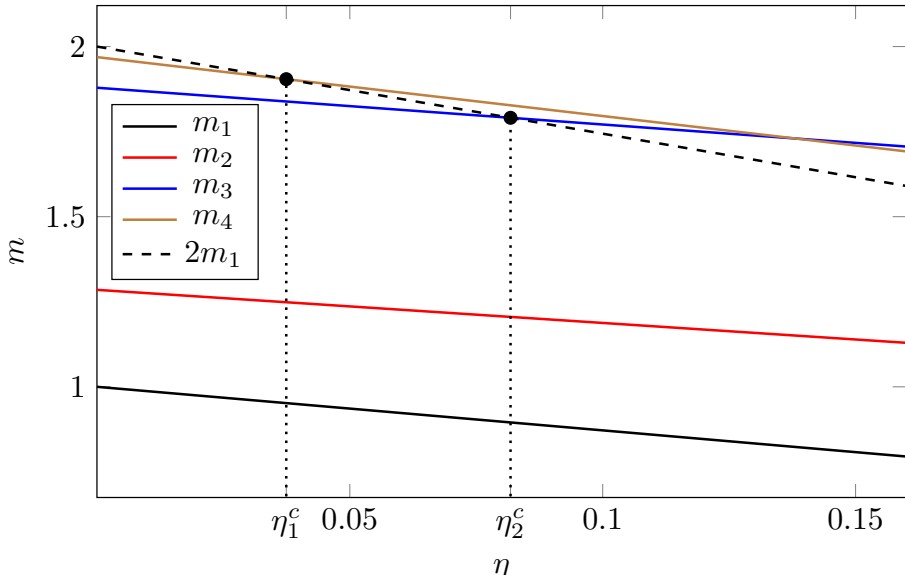

Figure 4.2: Evolution of the mass of particles $A_1, A_2, A_3$ and $A_4$ versus $\eta$. The dashed line corresponds to the threshold line to the continuum. The critical values $\eta_1^c$ and $\eta_2^c$ are those where the mass of the particles $A_4$ and $A_3$ crosses respectively the threshold line.

Of course, in the limit $\eta \to \infty$ only the lowest particle $A_1$ remains as a stable excitation, and the properties of system become more and more similar to those of the Ising universality class: $g > 0/g < 0$ leads to the high/low-temperature phase of the IM. Notably, as discussed in [45], an important aspect of the low-temperature phase of the general action (4.6) is the topological stability of the kink excitations. The disappearance of these excitations from the spectrum happens when, varying the parameter $\eta$, their mass becomes larger than the continuum threshold, i.e., for pure kinematical reasons rather than for topological instabilities. This stands in contrast to what happens in other non-integrable deformations (see, e.g. [54, 56]). For very large values of $\eta$, the natural counterpart of the $E_7$ theory for the TIM is the thermal perturbation of the IM described in Section 2: both theories are self-dual and have a $\mathbb{Z}_2$ symmetric action[9].

In conclusion, the presence of both couplings introduces two scales into the problem. Depending on the hierarchy between these scales, one either sees TIM at long distances (with small non-integrable corrections) or IM, with a crossover regime between the two. Experimentally (in the absence of symmetry breaking fields), these regimes and the transitions between them should be observable by tuning suitable control parameters, namely the ones related to temperature and chemical potential.

## 5 Form Factors of the Order/Disorder Operators in the TIM

Since the thermal deformation of the TIM corresponds to an integrable quantum field theory, the form factor expansion can exploited in order to compute the correlation functions of the model. For the operator $\epsilon(x)$, which also plays the role of the trace of the stress-energy tensor, its FF were computed in [47]. For the magnetisation operators, however, as already observed in [49], in the TIM there is an important novelty with respect to the similar IM, namely there

---

[9]Note that the $E_8$ theory present in the IM describes instead the dynamics of the Ising fixed point perturbed by the magnetisation operator for which $\mathbb{Z}_2$ is explicitly broken.

Table 5.1: Lowest energy states sorted according to the energy of their threshold and their $\mathbb{Z}_2$ parity.

| state | $\omega/m_1$ | parity | state | $\omega/m_1$ | parity |
|-------|--------------|--------|-------|--------------|--------|
| $A_2$ | 1.28558 | even | $A_1$ | 1.00000 | odd |
| $A_4$ | 1.96962 | even | $A_3$ | 1.87939 | odd |
| $A_1 A_1$ | $\geq 2.00000$ | even | $A_1 A_2$ | $\geq 2.28558$ | odd |
| $A_2 A_2$ | $\geq 2.57115$ | even | $A_6$ | $\geq 2.87939$ | odd |
| $A_1 A_3$ | $\geq 2.87939$ | even | $A_1 A_4$ | $\geq 2.96952$ | odd |
| $A_5$ | 2.53209 | even | $A_2 A_3$ | $\geq 3.16496$ | odd |
| $A_2 A_4$ | $\geq 3.25519$ | even | $A_1 A_5$ | $\geq 3.53209$ | odd |
| $A_7$ | 3.70167 | even | $A_3 A_4$ | $\geq 3.84901$ | odd |
| $A_3 A_3$ | $\geq 3.75877$ | even | | | |
| $A_2 A_5$ | $\geq 3.81766$ | even | | | |
| $A_1 A_6$ | $\geq 3.87939$ | even | | | |

are *two* odd spin operators, i.e. $\sigma(x)$ and $\sigma'(x)$, whose conformal dimensions differ less than 1. This circumstance has a drastic consequence on the resulting computation of the form factors of these operators (similarly for their duals). Namely, all the form factor equations that can be written down for the form factors of these operators, including those involving their asymptotic behaviour, are exactly the *same*! This implies that these operators cannot be distinguished from the form factor equations they satisfy[10] . Correspondingly, we expect to find a linear system of $n$ equations for the $n$ unknown parameters entering the form factors of these operators which *necessarily* must have rank $(n-1)$, i.e. there must be two free parameters to accommodate the two different magnetisation operators of the model. This is indeed what happens and, as discussed below, to fix uniquely the form factors of $\sigma(x)$ and $\sigma'(x)$ it is necessary to use their one-particle form factors $F_1$ and $F_3$ on the particle excitations $A_1$ and $A_3$. Interestingly enough, these two parameters can be fixed for both operators by exploiting the self-duality of the TIM and the cluster properties of the FF.

In the following, we consider the calculation of the form factors in the high-temperature phase of the TIM. Just as for the Ising model, the form factors in the low-temperature phase can be obtained by application of the Kramers–Wannier duality which swaps the roles of the order fields $\sigma$ and $\sigma'$ with the disorder fields $\mu$ and $\mu'$. Multi-particle states can be classified according to the their threshold energy and their $\mathbb{Z}_2$ parity, as shown in Table 5.1. This classification is convenient as their threshold energy specifies the importance of their contributions in the spectral expansion of correlation functions and dynamical structure factors, while their parity determines their occurrence via superselection rules implied by the $\mathbb{Z}_2$ spin reversal symmetry.

---

[10]It is worth to mention that the same situation happens for the magnetic deformation of the Ising Model, the one which breaks the $Z_2$ spin symmetry of the model and leads to the $E_8$ scattering theory of Zamolodchikov [31]: in this theory the magnetic operator $\sigma(x)$ and the thermal operator $\epsilon(x)$ are no longer distinguished by parity and they have matrix elements on the same multi-particle states of the theory, with the same asymptotic behaviour in the rapidities of all particles. This problem was successfully solved by Delfino and Simonetti, [33].

## 5.1 Order Operators

Anticipating that the FF of $\sigma$ and $\sigma'$ satisfy the same set of equations, in the following we denote a generic $\mathbb{Z}_2$ odd (order) operator by $\Phi$ and the corresponding dual (disorder) operator by $\widetilde{\Phi}$. The order operators have non-vanishing matrix elements between states with different $\mathbb{Z}_2$ parity. In particular this means that their vacuum expectation value (in the high temperature phase) is zero. According to the $\mathbb{Z}_2$ parity of the various particles, the non-vanishing one-particle form factors are

$$F_1^\Phi, F_3^\Phi, F_6^\Phi. \tag{5.1}$$

Given the presence of two order operators, we expect to find solutions of the form factor equations which depend on two free parameters, which we choose to be $F_1^\Phi$ and $F_3^\Phi$. This means that it should be possible to fix $F_6^\Phi$ in terms of the previous quantities $F_1^\Phi$ and $F_3^\Phi$.

Let's see how the form factors can be constructed step by step, referring the reader to Appendix A for the detailed expressions of the main quantities employed in the FF Ansatz, such as the minimal form factors $g_\alpha(\theta)$ and the polynomials $\mathcal{P}_\alpha(\theta)$ present in the denominator of their expressions. The $S$-matrix amplitudes $S_{ab}(\theta)$ which are needed for the computation of the form factors are summarised in Appendix B. Labelling the multi-particle states in terms of the increasing value of their rest energy, the first non vanishing two-particle FF of the order operators is

$$F_{12}^\Phi(\theta) = Q_{12}^\Phi(\theta)\frac{F_{12}^{min}(\theta)}{D_{12}(\theta)}, \tag{5.2}$$

where

$$F_{12}^{\min}(\theta) = g_{\frac{7}{18}}(\theta)g_{\frac{13}{18}}(\theta), \qquad D_{12}(\theta) = \mathcal{P}_{\frac{7}{18}}(\theta)\mathcal{P}_{\frac{13}{18}}(\theta). \tag{5.3}$$

$Q_{12}(\theta)$ is the polynomial which determines the operator and can be written as

$$Q_{12}(\theta) = \sum_{n=0}^{N_{12}} a_{12}^n \cosh^n(\theta). \tag{5.4}$$

The degree of this polynomial is fixed to be $N_{12} = 1$: for this result is obtained from eq. (A.17) using that $\lim_{\theta\to\infty} g_\alpha(\theta) \sim \exp(|\theta|/2)$ (see Appendix A) together with $\lim_{\theta\to\infty} \mathcal{P}_\alpha(\theta) \sim \exp(\theta)$. Hence, we need to fix two constants, $a_{12}^0$ and $a_{12}^1$, in order to determine the FF $F_{12}^\Phi(\theta)$. Using the bound state[11] form factor equation (A.4), these quantities can be expressed in terms of the constant one-particle form factors $F_1^\Phi$ and $F_3^\Phi$. The two linear equations for $a_{12}^0$ and $a_{12}^1$ are then[12]

$$0.113447\, a_{12}^0 - 0.0729222\, a_{12}^1 = F_1^\Phi\ ; \tag{5.5}$$
$$0.0328461\, a_{12}^0 + 0.011234\, a_{12}^1 = -F_3^\Phi\,.$$

Hence, if we knew the values of both $F_1^\Phi$ and $F_3^\Phi$, for $\Phi = \sigma$ or $\Phi = \sigma'$, we could determine the two coefficients $a_{12}^0$ and $a_{12}^1$ of this form factor for these operators.

The next two-particle FF to consider for the magnetisation operators is

$$F_{14}^\Phi(\theta) = \frac{Q_{14}^\Phi(\theta)}{D_{14}(\theta)}F_{14}^{\min}(\theta), \tag{5.6}$$

---

[11]In the following when implementing the bound state residue equations for the FF we assume that *all* the on-shell three-particle coupling $\Gamma_{ab}^c$ are positive, i.e. $\Gamma_{ab}^c > 0$.

[12]The coefficients of $a_{i,j}^k$ are computed by numerical integration of the minimal form factor functions (A.9). Although they can be performed to any desired precision, we only give here the first few non-zero digits to keep the equations simple.

where

$$F_{14}^{\min}(\theta) = g_{\frac{1}{6}}(\theta)\, g_{\frac{17}{18}}(\theta)\, g_{\frac{11}{18}}(\theta)\, g_{\frac{1}{2}}(\theta),$$
$$D_{14}(\theta) = \mathcal{P}_{\frac{1}{6}}(\theta)\, \mathcal{P}_{\frac{17}{18}}(\theta)\, \mathcal{P}_{\frac{11}{18}}(\theta)\, \mathcal{P}_{\frac{1}{2}}(\theta),$$

and

$$Q_{14}(\theta) = \sum_{n=0}^{N_{14}} a_{14}^{n} \cosh^{n}(\theta).$$

By enforcing the asymptotic behavior (5.6) of the FF, we can fix $N_{14} \leq 2$ and therefore there are three new constants to determine, i.e. $a_{14}^{0}, a_{14}^{1}, a_{14}^{2}$. There are three bound state residue equations, namely those relative to the simple poles coming from the particles $A_1, A_3$ and $A_6$: two of them which lead us back to $F_1^{\Phi}$, $F_3^{\Phi}$ while the third one brings in the new one-particle FF $F_6^{\Phi}$

$$0.0738796\, a_{14}^{0} - 0.0727572\, a_{14}^{1} + 0.071651\, a_{14}^{2} = F_1^{\Phi},$$
$$-0.00137278\, a_{14}^{0} + 0.000469519 a_{14}^{1} - 0.000160585\, a_{14}^{2} = F_3^{\Phi}, \qquad (5.7)$$
$$0.000110665\, a_{14}^{0} + 0.0000958386\, a_{14}^{1} + 0.0000829987\, a_{14}^{2} = -F_6^{\Phi}.$$

It is easy to see that, in absence of the values of $F_1^{\Phi}, F_3^{\Phi}$ and $F_6^{\Phi}$, the linear equations (5.5) and (5.7) written so far are not enough to find the five unknown constants $(a_{12}^{0}, a_{12}^{1}, a_{14}^{0}, a_{14}^{1}, a_{14}^{2})$. Hence, our strategy consists of considering more FF's, until there are enough linear equations in order to be able to fix all the necessary constants.

The next FF to consider, i.e. $F_{23}^{\Phi}(\theta)$, brings in three new unknown constants, $a_{23}^{0}, a_{23}^{1}, a_{23}^{2}$ but there are also three bound state residue equations relative to the particles $A_1, A_3$ and $A_6$. They yield the linear equations

$$0.0517869\, a_{23}^{0} - 0.0448488\, a_{23}^{1} + 0.0388402\, a_{23}^{2} = F_1^{\Phi},$$
$$-0.00638769\, a_{23}^{0} + 0.00218472\, a_{23}^{1} - 0.000747218\, a_{23}^{2} = F_3^{\Phi}, \qquad (5.8)$$
$$0.000693771\, a_{23}^{0} + 0.000445947\, a_{23}^{1} + 0.000286649\, a_{23}^{2} = -F_6^{\Phi}.$$

For the next two-particle FF given by $F_{15}^{\sigma}(\theta)$, there are three new unknown constants, $a_{15}^{0}, a_{15}^{1}, a_{15}^{2}$. The bound state axiom for the two poles related to the particles $A_3$ and $A_6$ imply the equations

$$0.0439601\, a_{15}^{0} - 0.0336754\, a_{15}^{1} + 0.0257969\, a_{15}^{2} = F_3^{\Phi}, \qquad (5.9)$$
$$0.00254047\, a_{15}^{0} + 0.000441149\, a_{15}^{1} + 0.0000766047\, a_{15}^{2} = F_6^{\Phi}.$$

Since the S-matrix amplitude $S_{15}(\theta)$ also has a double pole at $\theta = i\frac{\pi}{3}$, it can be exploited to give a corresponding residue equation according to (A.5) which yields the linear equations

$$\lim_{\theta \to i\frac{2\pi}{3}} \frac{F_{15}^{\Phi}(\theta)}{\Gamma_{12}^{1} \Gamma_{22}^{5}} = F_{12}^{\Phi}\left(i\frac{\pi}{6}\right), \qquad (5.10)$$

and

$$\lim_{\theta \to i\frac{\pi}{3}} \frac{F_{15}^{\Phi}(\theta)}{\Gamma_{23}^{1} \Gamma_{22}^{5}} = F_{23}^{\Phi}\left(i\frac{\pi}{18}\right). \qquad (5.11)$$

These two conditions give rise to the additional two equations

$$-0.0961442\, a_{15}^{0} + 0.0480721\, a_{15}^{1} - 0.024036\, a_{15}^{2} = 0.126345\, a_{12}^{0} + 0.109418\, a_{12}^{1}$$
$$-0.00651126\, a_{15}^{0} - 0.00325563\, a_{15}^{1} - 0.00162782\, a_{15}^{2} = 0.00189226\, a_{23}^{0}$$
$$+\, 0.00186351\, a_{23}^{1} + 0.0018352\, a_{23}^{2}. \qquad (5.12)$$

The FF $F_{34}^{\Phi}$ introduces four new constants $a_{34}^0, a_{34}^1, a_{34}^2, a_{34}^3$ and satisfies a bound state residue equation and four double pole residue equations.

Putting together all the equations collected so far, including the one-particle form factors $F_1^{\Phi}, F_3^{\Phi}$ and $F_6^{\Phi}$ involved in the computation, there are 18 unknown constants to fix. Altogether, for the above set of form factors there are 17 equations (11 from simple poles and 6 from double ones), but it turns out that only 16 of them are linearly independent. Therefore there exists a two-parameter family of solutions and $F_1^{\Phi}$ and $F_3^{\Phi}$ can be taken as the two parameters to label them. Notice, first of all, that $F_6$ is no longer a free parameter since it is fixed in terms of $F_1^{\Phi}$ and $F_3^{\Phi}$

$$F_6^{\Phi} = 0.115722 F_1^{\Phi} + 0.587743 F_3^{\Phi}. \tag{5.13}$$

Secondly, the coefficients entering the various FF are determined as

$$
\begin{aligned}
F_{12}: \quad & a_{12}^0 = 3.06131 F_1^{\Phi} - 19.8715 F_3^{\Phi}, \quad & a_{12}^1 = -8.95069 F_1^{\Phi} - 30.9146 F_3^{\Phi}, \\
F_{14}: \quad & a_{14}^0 = -160.899 F_1^{\Phi} - 1600.15 F_3^{\Phi}, \quad & a_{14}^1 = -626.504 F_1^{\Phi} - 3040.25 F_3^{\Phi}, \\
& a_{14}^2 = -456.311 F_1^{\Phi} - 1437.25 F_3^{\Phi}, \\
F_{15}: \quad & a_{15}^0 = 32.1365 F_1^{\Phi} + 177.579 F_3^{\Phi}, \quad & a_{15}^1 = 70.7301 F_1^{\Phi} + 289.789 F_3^{\Phi}, \\
& a_{15}^2 = 37.5681 F_1^{\Phi} + 114.447 F_3^{\Phi}, \\
F_{32}: \quad & a_{23}^0 = -38.6198 F_1^{\Phi} - 337.751 F_3^{\Phi}, \quad & a_{23}^1 = -142.958 F_1^{\Phi} - 621.037 F_3^{\Phi}, \\
& a_{23}^2 = -87.8337 F_1^{\Phi} - 266.777 F_3^{\Phi}, \\
F_{34}: \quad & a_{34}^0 = -493.626 F_1^{\Phi} - 2722.44 F_3^{\Phi}, \quad & a_{34}^1 = -1617.98 F_1^{\Phi} - 6682.94 F_3^{\Phi}, \\
& a_{34}^2 = -1495.64 F_1^{\Phi} - 5104.39 F_3^{\Phi}, \quad & a_{34}^3 = -399.244 F_1^{\Phi} - 1174.92 F_3^{\Phi}.
\end{aligned}
\tag{5.14}
$$

## 5.2 Disorder Operators

Let's now focus the attention on the FF's of the dual disorder operators $\mu(x)$ and $\mu'(x)$ (hereafter collectively denoted as $\widetilde{\Phi}$). In order to write down and find solution of the FF equations for the matrix elements of these operators, it is necessary to consider that

- these operators couple to the $\mathbb{Z}_2$ even multi-particle states.

- moreover, their vacuum expectation values $F_0^{\widetilde{\Phi}} = \langle 0 | \widetilde{\Phi} | 0 \rangle$ (in the conformal normalisation of both operators) are exactly known [26].

Taking into account the parity of the particles, the asymptotic behaviour of the FF, the semi-locality of the dual operators with respect to the kink states (but not with respect to the purely particle excitations), we have the following Ansatz for the corresponding lowest FF's of the dual operators

$$F_{11}^{\widetilde{\Phi}}(\theta) = \frac{1}{\cosh\dfrac{\theta}{2}} \frac{Q_{11}^{\widetilde{\Phi}}(\theta)}{D_{11}(\theta)} F_{11}^{\min}(\theta); \quad N_{11} = 1$$

$$F_{13}^{\widetilde{\Phi}}(\theta) = \cosh\frac{\theta}{2} \frac{Q_{13}^{\widetilde{\Phi}}(\theta)}{D_{13}(\theta)} F_{13}^{\min}(\theta); \quad N_{13} = 1$$

$$F_{22}^{\widetilde{\Phi}}(\theta) = \frac{Q_{22}^{\widetilde{\Phi}}(\theta)}{D_{22}(\theta)} F_{22}^{\min}(\theta); \qquad N_{22} = 1 \qquad (5.15)$$

$$F_{24}^{\widetilde{\Phi}}(\theta) = \frac{Q_{24}^{\widetilde{\Phi}}(\theta)}{D_{24}(\theta)} F_{24}^{\min}(\theta); \qquad N_{24} = 2$$

$$F_{33}^{\widetilde{\Phi}}(\theta) = \frac{1}{\cosh\dfrac{\theta}{2}} \frac{Q_{33}^{\widetilde{\Phi}}(\theta)}{D_{33}(\theta)} F_{33}^{\min}(\theta); \quad N_{33} = 3 \,.$$

Notice that we included an extra factor $1/\cosh\theta/2$ (which induces an annihilation pole present for equal particles at $\theta = i\pi$) in the expressions $F_{11}^{\widetilde{\Phi}}(\theta)$ and $F_{33}^{\widetilde{\Phi}}(\theta)$ in view of the kink nature of these excitations in the low-temperature phase and therefore the non-locality of both excitations $A_1$ and $A_3$ with respect to the disorder operators. For the same reason of non-locality, we also introduced an extra term $\cosh\theta/2$ in $F_{13}^{\widetilde{\Phi}}(\theta)$, which also guarantees that this FF (similarly to all the others) has a constant asymptotic behaviour at $|\theta| \to \infty$. Taking into account the known values of the vacuum expectation values $F_0^{\widetilde{\Phi}}$, in this case we have 19 unknowns (including in this counting, at this stage also the two VEV of the operators, even though their values are known by other means) and 20 equations: among them

- 2 equations comes from the kinematical pole of $F_{11}^{\widetilde{\Phi}}$ and $F_{33}^{\widetilde{\Phi}}$ since the particles $A_1$ and $A_3$ (being a kink in the low-temperature phase) are non-local with respect the disorder operators $\widetilde{\Phi}$.

- 12 equations come from the bound state pole residue equations.

- 6 equations come from the double pole residue equations.

However, only 18 of these linear equations are linearly independent, leading to a solution which can be parameterised in terms of $F_0^{\widetilde{\Phi}}$ and $F_2^{\widetilde{\Phi}}$ as follows: first of all, the one-particle FF's $F_4^{\widetilde{\Phi}}, F_5^{\widetilde{\Phi}}$ and $F_7^{\widetilde{\Phi}}$ are given in terms of $F_0^{\widetilde{\Phi}}$ and $F_2^{\widetilde{\Phi}}$ as follows

$$
\begin{aligned}
F_4^{\widetilde{\Phi}} &= 0.0979846 F_0^{\widetilde{\Phi}} - 0.71782 F_2^{\widetilde{\Phi}}, \\
F_5^{\widetilde{\Phi}} &= 0.529952 F_2^{\widetilde{\Phi}} - 0.100203 F_0^{\widetilde{\Phi}}, \\
F_7^{\widetilde{\Phi}} &= 0.037784 F_0^{\widetilde{\Phi}} - 0.157422 F_2^{\widetilde{\Phi}},
\end{aligned}
\qquad (5.16)
$$

while the remaining coefficients of the various FF's are

$$
\begin{aligned}
F_{11}^{\widetilde{\Phi}}: \quad & a_{11}^0 = 5.08848 F_2^{\widetilde{\Phi}} - 0.21014 F_0^{\widetilde{\Phi}}, \quad && a_{11}^1 = 5.08848 F_2^{\widetilde{\Phi}} - 1.21014 F_0^{\widetilde{\Phi}}, \\
F_{13}^{\widetilde{\Phi}}: \quad & a_{13}^0 = 89.6538 F_2^{\widetilde{\Phi}} - 13.8826 F_0^{\widetilde{\Phi}}, \quad && a_{13}^1 = 63.3814 F_2^{\widetilde{\Phi}} - 18.1224 F_0^{\widetilde{\Phi}}, \\
F_{22}^{\widetilde{\Phi}}: \quad & a_{22}^0 = 22.2545 F_2^{\widetilde{\Phi}} - 2.57115 F_0^{\widetilde{\Phi}}, \quad && a_{22}^1 = 17.9847 F_2^{\widetilde{\Phi}} - 5.1423 F_0^{\widetilde{\Phi}}, \\
F_{24}^{\widetilde{\Phi}}: \quad & a_{24}^0 = 162.262 F_2^{\widetilde{\Phi}} - 28.5588 F_0^{\widetilde{\Phi}}, \quad && a_{24}^1 = 251.212 F_2^{\widetilde{\Phi}} - 57.9849 F_0^{\widetilde{\Phi}}, \\
& a_{24}^2 = 90.1906 F_2^{\widetilde{\Phi}} - 27.0272 F_0^{\widetilde{\Phi}}, \\
F_{33}^{\widetilde{\Phi}}: \quad & a_{33}^0 = 438.962 F_2^{\widetilde{\Phi}} - 86.4572 F_0^{\widetilde{\Phi}}, \quad && a_{33}^1 = 1008.33 F_2^{\widetilde{\Phi}} - 233.757 F_0^{\widetilde{\Phi}}, \\
& a_{33}^2 = 730.328 F_2^{\widetilde{\Phi}} - 195.529 F_0^{\widetilde{\Phi}}, \quad && a_{33}^3 = 160.963 F_2^{\widetilde{\Phi}} - 49.2296 F_0^{\widetilde{\Phi}}.
\end{aligned}
\tag{5.17}
$$

## 5.3 Cluster Property

On the basis of the analysis of the previous subsections, we conclude that

- in the $\mathbb{Z}_2$ odd sector of the operators (the order operators), there are the 4 free parameters $(F_1^\sigma, F_3^\sigma)$ and $(F_1^{\sigma'}, F_3^{\sigma'})$ which determine the rest of the FF of the corresponding operators.

- in the $\mathbb{Z}_2$ even sector of the operators (the disorder operators), there are also 4 free parameters $(F_0^\mu, F_2^\mu)$ and $(F_0^{\mu'}, F_2^{\mu'})$ which determine the rest of the FF of the corresponding operators.

The interesting question we address now is whether there are additional equations which reduce the number of independent parameters that determine the FFs of the order and disorder fields. It turns out that this is indeed the case: namely, the *only* parameters to fix in order to specify all FF's of the order and disorder operators are nothing else but the VEV of the two disorder operators $\mu$ and $\mu'$, i.e. $F_0^\mu$ and $F_0^{\mu'}$.

To arrive at this conclusion, it is necessary to consider a set of *non-linear* equations which relate the different Form Factors. These are the cluster equations [25, 57, 58] which give rise to the following relations between the two-particle and one-particle FF's

$$
\lim_{\theta \to \infty} F_{ij}^\Phi(\theta) = \frac{\omega_{ij}}{F_0^\Phi}(F_i^\Phi F_j^\Phi),
\tag{5.18}
$$

where $\omega_{ij}$ is a phase which depends on the phase conventions for the one-particle states. Notice that in writing this equation we do not distinguish $\Phi$ and $\widetilde{\Phi}$, since this information can be simply recovered by the $\mathbb{Z}_2$ charge of the operators involved in this identity. In particular, the cluster equations lead to relations between form factors of the order and disorder operators [59].

Let us first employ the cluster equation (5.18) in the case of the form factor $F_{22}^{\widetilde{\Phi}}$ and see how this FF allows us to fix the ratio $F_2^{\widetilde{\Phi}}/F_0^{\widetilde{\Phi}}$. Indeed, using the expression of the coefficients appearing in $F_{22}^{\widetilde{\Phi}}$ (see eq.(5.17)) results in the quadratic equation

$$
0.851179 - 2.97692 \frac{F_2^{\widetilde{\Phi}}}{F_0^{\widetilde{\Phi}}} = \omega_{22} \left( \frac{F_2^{\widetilde{\Phi}}}{F_0^{\widetilde{\Phi}}} \right)^2.
\tag{5.19}
$$

Looking for a real solution of this equation, we can have $\omega_{22} = \pm 1$ and corresponding to the choice of the sign there are two pairs of solutions are

$$
\begin{aligned}
F_2^{\widetilde{\Phi}}/F_0^{\widetilde{\Phi}} &= (-3.23966, 0.262737) & \omega_{22} &= +1\,, \\
F_2^{\widetilde{\Phi}}/F_0^{\widetilde{\Phi}} &= (0.320413, 2.6565) & \omega_{22} &= -1\,.
\end{aligned}
\tag{5.20}
$$

We can now use another piece of information, namely the computation of the conformal dimension $\Delta$ of an operator (which in the TIM are *all* positive) in terms of the $\Delta$-theorem [58]. In view of the positivity of $\Delta$, this implies that the ratio $F_2^{\widetilde{\Phi}}/F_0^{\widetilde{\Phi}}$ should be positive, i.e. $F_2^{\widetilde{\Phi}}$ and $F_0^{\widetilde{\Phi}}$ must have the same sign. Hence, this positivity condition for $\Delta$ selects the second option in the equation (5.20) above as the proper one.

Employing now the known result for the vacuum expectation values of the different disorder operators $\mu$ and $\mu'$ [26]

$$
\begin{aligned}
F_0^{\mu} &= 1.59427\ldots |g|^{1/24} &= 1.44394\ldots m_1^{3/40}\,, \\
F_0^{\mu'} &= 2.45205\ldots |g|^{35/72}. &= 0.772185\ldots m_1^{7/8}\,,
\end{aligned}
\tag{5.21}
$$

where we used the mass-coupling relation (4.6), we can determine the corresponding one-particle FF's $F_2^{\mu}$ and $F_2^{\mu'}$. Substituting these two values into the $\Delta$-theorem, the two disorder operators can be finally identified with the assignments

$$
|F_2^{\mu}| = 0.462658\ldots\,,
\tag{5.22}
$$

$$
|F_2^{\mu'}| = 2.05131\ldots\,.
\tag{5.23}
$$

As discussed in Section 6, these values are confirmed by the numerical results from the truncated conformal space approach.

From the cluster equation for the form factors $F_{11}(\theta)$ and $F_{33}(\theta)$ it follows that

$$
-0.697009i \left(F_0^{\widetilde{\Phi}}\right)^2 + 2.93083i F_0^{\widetilde{\Phi}} F_2^{\widetilde{\Phi}} = \omega_{11} \left(F_1^{\Phi}\right)^2\,,
\tag{5.24}
$$

$$
-0.641012i \left(F_0^{\widetilde{\Phi}}\right)^2 + 2.09588i F_0^{\widetilde{\Phi}} F_2^{\widetilde{\Phi}} = \omega_{33} \left(F_3^{\Phi}\right)^2\,.
\tag{5.25}
$$

Assuming $F_1^{\Phi}$ and $F_3^{\Phi}$ to be real quantities and substituting $F_0^{\widetilde{\Phi}}$ and $F_2^{\widetilde{\Phi}}$ both for $\mu/\mu'$, gives that $\omega_{11} = \omega_{33} = i$, similarly to the Ising case. This leads to the predictions:

$$
\begin{aligned}
|F_1^{\sigma}| &= 0.710426\ldots\,, \\
|F_3^{\sigma}| &= 0.252315\ldots\,, \\
|F_1^{\sigma'}| &= 2.05592\ldots\,, \\
|F_3^{\sigma'}| &= 1.71395\ldots\,.
\end{aligned}
\tag{5.26}
$$

The remaining piece of information can be extracted from the cluster equation coming from $F_{12}(\theta)$ which has the form

$$
1.4135 F_0^{\widetilde{\Phi}} F_1^{\Phi} + 4.88205 F_0^{\widetilde{\Phi}} F_3^{\Phi} = \omega_{12} F_1^{\Phi} F_2^{\widetilde{\Phi}}\,.
\tag{5.27}
$$

Again, assuming that the one-particle form factors are real, this equation is only consistent with (5.21) and (5.26) if $\omega_{12} = -1$ and $F_3^{\Phi}$ has a sign opposite to that of $F_1^{\Phi}$.

So, as anticipated, we were able to fix all the one-particle form factors of the various order and disorder operators in terms of the VEV of the disorder operators, using the cluster properties of the two-particle form factors.

# 6 Comparison to TCSA

In this section we present the comparison between the theoretical values of the form factors and their numerical determination coming from the truncated conformal space approach (TCSA). The TCSA is a non-perturbative approach to perturbed conformal field theories, originally introduced in [27] to construct their spectra in finite volume, which for the tricritical Ising model was first performed in [41]. In recent years the scope of the method was extended to the computation of matrix elements [60, 61] and simulation of non-equilibrium dynamics [62]. For a recent review of TCSA and related Hamiltonian truncation methods the reader is referred to [63]. We give a brief overview of the numerical method in Appendix C; the specific implementation of the algorithm exploits the chiral factorisation of conformal field theory as described in [64].

## 6.1 Form Factors in Finite Volume

Since the TCSA is a finite volume approach, in order to compare its results to the form factor bootstrap solutions it is necessary to relate them firstly to the finite volume matrix elements. Since the $E_7$ model has a diagonal factorised scattering theory, here we recall the description of finite volume form factors for diagonal scattering, which was obtained in [60,61].

The multi-particle states in finite volume can be described by so-called Bethe–Yang quantisation relations. For an $N$-particle state let us define the following $N$ functions:

$$Q_i(\theta_1, \theta_2, \dots, \theta_N) = m_i L \sinh \theta_i - i \sum_{j \neq i} \log S_{ij}(\theta_i - \theta_j), \tag{6.1}$$

where $i, j = 1 \dots, N$, and $S_{ij}$ is the scattering matrix element between particles $i$ and $j$ (see Appendix B). The Jacobian matrix of the mapping defined by the functions $Q_i$ is given by

$$\mathcal{J}(\theta_1, \theta_2, \dots, \theta_N)_{i,j} = \frac{\partial Q_i(\theta_1, \theta_2, \dots, \theta_N)_i}{\partial \theta_j}, \tag{6.2}$$

and its determinant is denoted by

$$\rho(\theta_1, \theta_2, \dots, \theta_N) = \det \mathcal{J}(\theta_1, \theta_2, \dots, \theta_N). \tag{6.3}$$

A finite volume state $|\{I_1, I_2, \dots, I_N\}\rangle_L$ labelled by quantum numbers $I_i \in \mathbb{Z}$ corresponds to the following solution of the Bethe–Yang equations:

$$Q_i(\overline{\theta}_1, \overline{\theta}_2, \dots, \overline{\theta}_N) = 2\pi I_i. \tag{6.4}$$

The matrix element of a local operator between finite volume states is then given by [60]

$$\langle\{I_1', I_2', \dots, I_M'\}|\mathcal{O}(0,0)|\{I_1, I_2, \dots, I_N\}\rangle_L =$$
$$\frac{F^{\mathcal{O}}_{a_1' a_2' \dots a_M' a_1 a_2 \dots a_N}(\overline{\theta}_M' + i\pi, \dots, \overline{\theta}_1' + i\pi, \overline{\theta}_1, \dots, \overline{\theta}_N)}{\sqrt{\rho(\overline{\theta}_1', \dots, \overline{\theta}_M')\rho(\overline{\theta}_1, \dots, \overline{\theta}_N)}} + \text{disc.} + O(e^{-\mu L}), \tag{6.5}$$

where disconnected terms can appear when a (sub)set of particle types and their rapidities agree exactly in the two states [61]. The above formula is valid to all orders in $1/L$, up to corrections suppressed exponentially in large volume, which depend on the bound state structure of the theory [65], see also [66–68] for recent developments.

## 6.2 Leading and Sub-leading Magnetic Fields $\sigma$ and $\sigma'$

### 6.2.1 One-particle Form Factors

The form factor equations presented in Appendix A completely determine the one-particle form factors in terms of the vacuum expectation values of the disorder operators whose exact values are predicted in [69, 70]. These predictions can be directly checked against the TCSA data based on the finite volume one-particle form factor of a stationary particle [60]:

$$| \langle 0 | \mathcal{O} | \{0\} \rangle_{i,L} | = \frac{|F_i^{\mathcal{O}}|}{\sqrt{m_i L}} + O(e^{-\mu L}). \tag{6.6}$$

After identifying the one-particle energy levels, the value of the one-particle form factors can be refined further by an exponential fit of the remaining volume dependence, the results of which are compared to the bootstrap predictions in Table 6.1.

Table 6.1: One-particle form factors of the magnetisation operators, TCSA vs. form factor boostrap.. The digits in parentheses indicate the error in the last digit of the TCSA data, which is estimated from the stability of the exponential fit under the choice of the volume window. Note that the deviations in the subleading magnetisation data are an order of magnitude worse then for the leading one, and also that TCSA estimates of matrix elements involving higher particles are generally less accurate due to truncation errors, as discussed in Appendix C.2.

| Particle | $|F_i^{\sigma}|$ | $|F_i^{\sigma,TCSA}|$ | $|F_i^{\sigma'}|$ | $|F_i^{\sigma',TCSA}|$ |
|---|---|---|---|---|
| 1 | 0.71043 | 0.7103(2) | 2.05592 | 2.043(2) |
| 3 | 0.25232 | 0.252(1) | 1.71395 | 1.707(3) |
| 6 | 0.06608 | 0.0658(8) | 0.769448 | 0.768(4) |

### 6.2.2 Elementary Two-particle Form Factors $F_{12}, F_{14}, F_{23}$

In the zero momentum sector, the following relation holds between the rapidities of particles of type $i$ and $j$:

$$m_i \sinh \theta_1 + m_j \sinh \theta_2 = 0. \tag{6.7}$$

Let us recall the Bethe–Yang equations for a two-particle state:

$$\begin{aligned} m_i L \sinh \theta_1 - i \log S_{ij}(\theta_1 - \theta_2) &= 2\pi I, \\ m_j L \sinh \theta_2 - i \log S_{ij}(\theta_2 - \theta_1) &= -2\pi I, \end{aligned} \tag{6.8}$$

(note that either one of these equations can be replaced by the zero-momentum relation (6.7)). The finite volume prediction for the two-particle form factor then reads

$$| \langle 0 | \mathcal{O} | \{I, -I\} \rangle_{ij,L} | = \frac{|F_{ij}^{\mathcal{O}}(\bar{\theta}_1 - \bar{\theta}_2)|}{\sqrt{\rho_{ij}(\bar{\theta}_1, \bar{\theta}_2)}} + O(e^{-\mu L}). \tag{6.9}$$

Note that because of parity symmetry the states $|\{I, -I\}\rangle_{ij,L}$ and $|\{-I, I\}\rangle_{ij,L}$ are distinct for $m_i \neq m_j$. However, they are correspond to degenerate solutions of the Bethe-Yang equations, and also have two-particle finite volume form factors of identical magnitude. Naively, the TCSA could result in any linear combination of these degenerate states. However, in finite

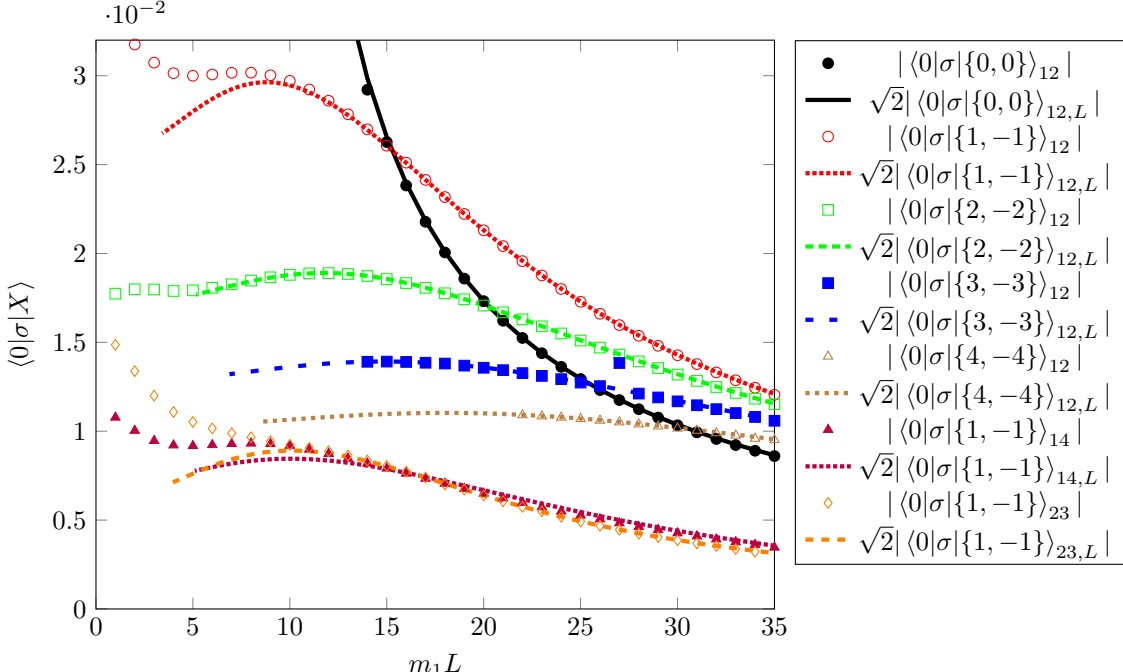

Figure 6.1: Matrix elements $\langle 0|\sigma|X\rangle$ (in units $m_1 = 1$) determined from TCSA (plotted with discrete markers), compared with finite volume form factors of two-particle states corresponding to different total momentum zero solutions to the Bethe–Yang equations (shown with lines). As discussed in Appendix C.2, the main source of error in this comparison are the exponential finite volume corrections to the matrix elements. Also note the outlier point in the matrix element $\langle 0|\sigma|\{-3,3\}\rangle$, which is due to the vicinity of a level crossing.

volume the degeneracy is lifted by exponential finite size effects, and consequently the two states occur in finite volume as parity eigenstates

$$\frac{1}{\sqrt{2}}\left(|\{I,-I\}\rangle_{ij,L} \pm |\{-I,I\}\rangle_{ij,L}\right), \tag{6.10}$$

which can be confirmed by directly evaluating the parity of the TCSA eigenvectors. As a result, one of the corresponding matrix elements vanishes and the other acquires a factor of $\sqrt{2}$ relative to (6.9) [60].

Figures 6.1 and 6.2 demonstrate the excellent agreement between the form factor predictions and the TCSA data. There is a single data point in both data sets at $m_1 L = 27$ with an exceptionally large deviation from the predicted line. Such effects are due to the vicinity of level crossing: eigenvectors corresponding to degenerate or nearly degenerate levels are very sensitive to any small perturbation, and thus even a small truncation error can have a disproportionately large effect [71].

Another way to present the comparison for the two-particle form factors and the corresponding TCSA matrix elements is shown in Figures 6.3 and 6.4, where they are plotted as a function of the rapidity difference of the two particles, which can be extracted from the momentum and energy expressions

$$m_i \sinh\theta_1 + m_j \sinh\theta_2 \;=\; 0, \tag{6.11}$$
$$m_i \cosh\theta_1 + m_j \cosh\theta_2 \;=\; E_{TCSA}, \tag{6.12}$$

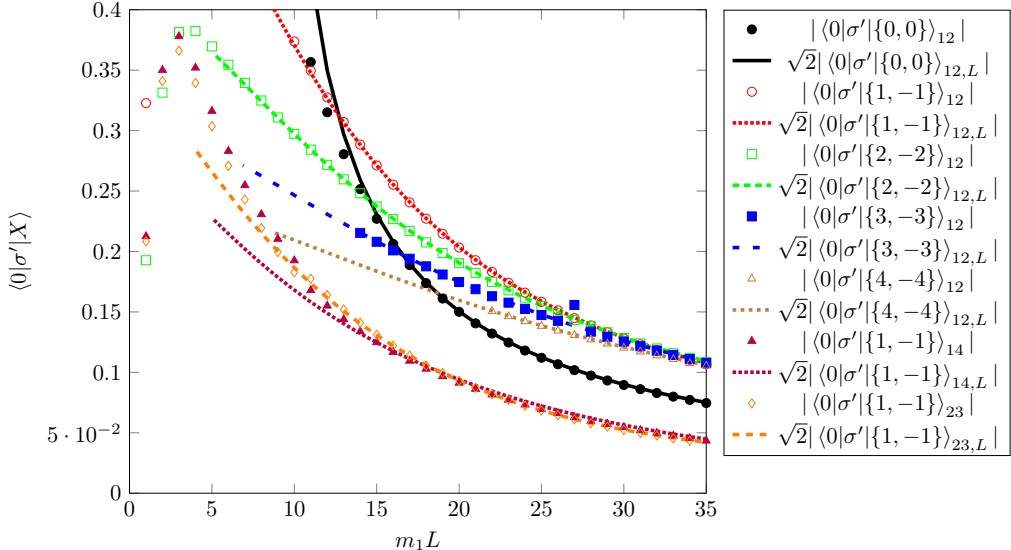

Figure 6.2: Matrix elements $\langle 0|\sigma'|X\rangle$ (in units $m_1 = 1$) determined from TCSA (plotted with discrete markers), compared with finite volume form factors of two-particle states corresponding to different total momentum zero solutions to the Bethe–Yang equations (shown with lines). As discussed in Appendix C.2, the main source of error in this comparison are the exponential finite volume corrections to the matrix elements. The outlier point in the matrix element $\langle 0|\sigma'|\{-3,3\}\rangle$ is due to the vicinity of the same level crossing as in Fig. 6.1.

from which one can extract $\theta_1, \theta_2$ and calculate the corresponding $\rho_{ij}$, and plot

$$|\sqrt{\rho_{ij}}\,\langle 0|\sigma|X\rangle_{ij,TCSA}|$$

as a function of the corresponding rapidity difference which, according to (6.9) must agree with $|F^{\mathcal{O}}_{ij}(\theta_1 - \theta_2)|$. Note that TCSA results extracted from two-particle states with the same particle content but different quantum numbers must scale to the same form factor prediction, which is indeed the case as demonstrated in Figures 6.3 and 6.4, apart from the previously observed exceptionally large deviation which was due to the vicinity of a level crossing at $m_1 L = 27$. The deviations at large rapidities correspond to data from small volumes, and result from the approximation of neglecting the exponential finite size corrections in (6.5).

For the case of $A_1 A_2$ two-particle states it was possible to identify four different ones, for which the data extracted from the TCSA scale to the same universal curve which is in an excellent agreement with the theoretical prediction, both for $\sigma$ and $\sigma'$. Note that all TCSA data deviate from the theoretical curves for small and large values of the rapidity difference (large and small volumes respectively), related to truncation errors and exponential finite size corrections, respectively. In the case of $A_1 A_4$ and $A_2 A_3$ two-particle states, only a single level could be identified for each, again showing excellent agreement with the form factor prediction.

### 6.2.3 Crossed Two-particle Form Factors $F_{12}, F_{14}, F_{15}, F_{23}, F_{34}$

Two-particle form factors can also be compared against matrix elements between two one-particle states using crossing symmetry. Since the TCSA calculation was carried out in the zero momentum sector, the one-particle states correspond to stationary particles with Bethe–Yang quantum number $I = 0$. Therefore these matrix elements are related to two-particle form

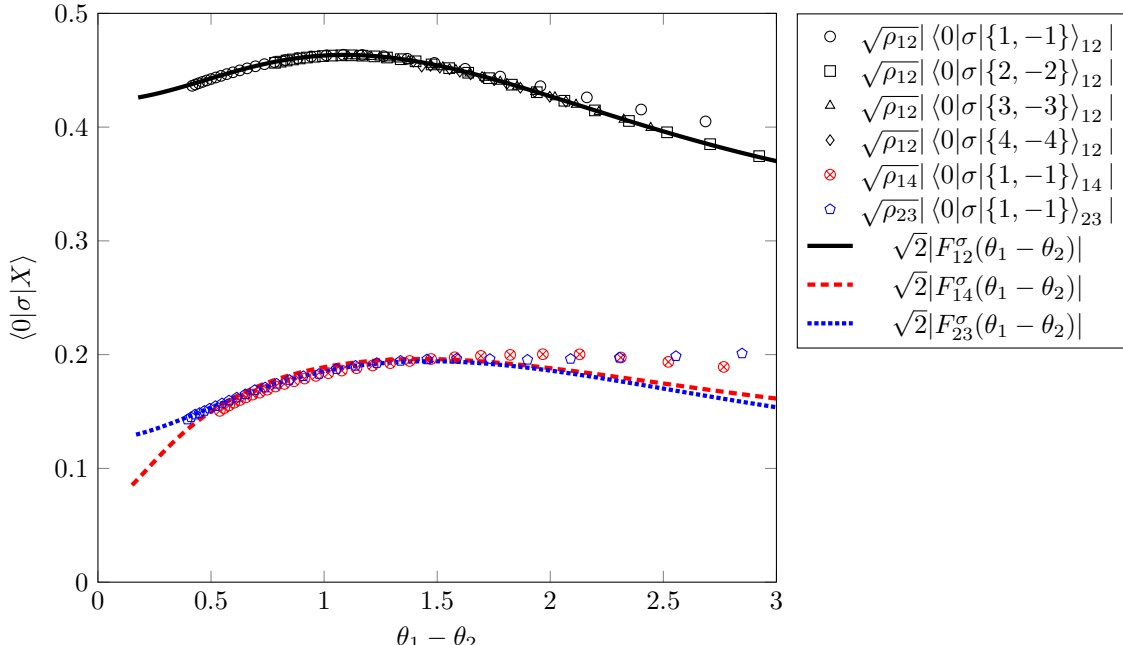

Figure 6.3: Rescaled vacuum-two-particle matrix elements of $\sigma$ (discrete markers) plotted against the rapidity difference and compared to the exact form factors obtained from the bootstrap (shown with lines). Form factors of 12 states of larger quantum numbers are better fit in large rapidities, due to smaller finite size effects. For twp-particle states 14 and 23 states we could only identify the lowest levels for each, since the higher ones are deeply in the continuum of 12 states, which makes their identification ambiguous.

factors evaluated at rapidity $i\pi$ (6.5):

$$_i\langle\{0\}|\mathcal{O}|\{0\}\rangle_j = \frac{F_{ij}^{\mathcal{O}}(i\pi)}{L\sqrt{m_i m_j}} + O(e^{-\mu L}).$$  (6.13)

From the TCSA data one can compute an estimate for these crossed two-particle form factors, taking into account the exponential corrections by extrapolating with an exponential fit in the volume[13]. The resulting extrapolated form factors are compared to the theoretical predictions in in Table 6.2.

## 6.3 Form Factors of the Disorder Operators $\mu$ and $\mu'$

The form factors of the disorder operators $\mu$ and $\mu'$ in the high-temperature phase can be determined from TCSA by computing the matrix elements of the order fields $\sigma$ and $\sigma'$ in the low-temperature phase. In this phase there are two degenerate ground states in finite volume, with one in the Neveu-Schwarz sector, denoted as $|0\rangle_{\text{NS}}$, and the other one ($|0\rangle_{\text{R}}$) in the Ramond sector. Due to tunneling effects in finite volume, they eventually correspond to $\mathbb{Z}_2$ even/odd linear combinations

$$|0\rangle_{\text{NS}} = \frac{1}{\sqrt{2}}\left(|+\rangle + |-\rangle\right), \qquad |0\rangle_{\text{R}} = \frac{1}{\sqrt{2}}\left(|+\rangle - |-\rangle\right),$$  (6.14)

---

[13]Extrapolation is especially important when some particle is a very weakly bound state which results in a small value for the exponent $\mu$ [65]. This is the case for particle 4 which is a very weakly bound state of two particles of species 1.

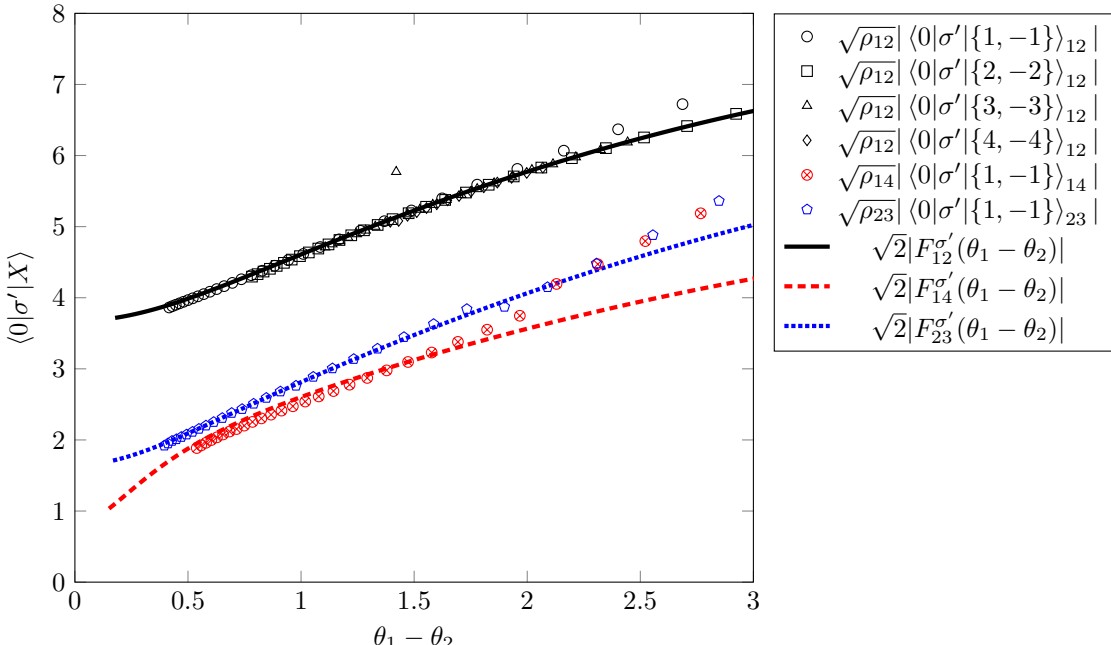

Figure 6.4: Rescaled vacuum-two-particle matrix elements of $\sigma'$ (discrete markers) plotted against the rapidity difference and compared to the exact form factors obtained from the bootstrap (shown with lines). Form factors of 12 states of larger quantum numbers are better fit in large rapidities, due to smaller finite size effects. For twp-particle states 14 and 23 states we could only identify the lowest levels for each, since the higher ones are deeply in the continuum of 12 states, which makes their identification ambiguous.

in terms of the states $|\pm\rangle$ which correspond to the states with definite values of magnetisation

$$
\begin{aligned}
\langle\pm|\sigma|\pm\rangle &= \pm F_0^\mu, \quad \langle\pm|\sigma'|\pm\rangle = \pm F_0^{\mu'}, \\
\langle\pm|\sigma|\mp\rangle &= 0 = \langle\pm|\sigma'|\pm\rangle,
\end{aligned}
\tag{6.15}
$$

(valid up to exponential finite size correction), where the $F_0$ are the exact vacuum expectation values determined in [69, 70]. As a result of (6.14), this translates to the order operators having vanishing expectation values in the NS and R ground states, while their off-diagonal matrix elements are equal to $F_0^\mu/F_0^{\mu'}$.

For the multi-particle states periodic boundary conditions imply that there are only states with even number of kinks (corresponding to the odd particles 1, 3 and 6 of the $E_7$ scattering theory). However, all types of multi-particle states appear over both vacua, i.e. both in the Neveu-Schwarz and Ramond sectors. Furthermore depending on the particle content and Bethe–Yang quantum numbers $I$ (cf. Eq. (6.8)), they appear either solo or in degenerate pairs. For example, there is a single $A_1A_1$ state in the Neveu–Schwarz sector for each even value of $I$, and a single state for each odd value of $I$ in the Ramond sector. In contrast, $A_2A_2$ states always have even $I$, and they come in two copies, one in both sectors.

Note that similarly to the ground states, the fields $\sigma$ and $\sigma'$ only have off-diagonal matrix elements between the two sectors. Consequently, for any given multi-particle form factor there are two choices for comparison to TCSA, choosing the vacuum from the Neveu-Schwarz and the multi-particle states from the Ramond sector, or vice versa.

Table 6.2: Crossed two-particle form factors compared to TCSA. The digits in parentheses indicate the error in the last digit of the TCSA data, which is estimated from the stability of the exponential fit under the choice of the volume window. Note that the deviations in the subleading magnetisation data are an order of magnitude worse then for the leading one, and also that TCSA estimates of matrix elements involving higher particles are generally less accurate due to truncation errors, as discussed in Appendix C.2. In particular, the apparent large deviation in the entries $F_{34}^{\sigma}(i\pi)$ and $F_{34}^{\sigma'}(i\pi)$ is due to both one-particle states being high in the spectrum, which makes the determination of the corresponding wave functions less accurate.

| form factor | $|F_{12}^{\sigma}(i\pi)|$ | $|F_{12}^{\sigma'}(i\pi)|$ | $|F_{14}^{\sigma}(i\pi)|$ | $|F_{14}^{\sigma'}(i\pi)|$ | $|F_{15}^{\sigma}(i\pi)|$ |
|---|---|---|---|---|---|
| Exact value | 5.74730 | 5.76839 | 5.88370 | 14.2234 | 1.29286 |
| TCSA (extrapolated) | 5.748(2) | 5.67(1) | 5.94(3) | 14.3(2) | 1.26(1) |

| form factor | $|F_{15}^{\sigma'}(i\pi)|$ | $|F_{23}^{\sigma}(i\pi)|$ | $|F_{23}^{\sigma'}(i\pi)|$ | $|F_{34}^{\sigma}(i\pi)|$ | $|F_{34}^{\sigma'}(i\pi)|$ |
|---|---|---|---|---|---|
| Exact value | 5.94185 | 7.55954 | 5.63568 | 12.0322 | 4.28979 |
| TCSA (extrapolated) | 5.74(3) | 7.563(2) | 5.48(3) | 11.73(6) | 3.946(3) |

### 6.3.1  One-particle Form Factors and Vacuum Expectation Values

The comparison between the one-particle form factors derived from the bootstrap and their numerical counter parts obtained from TCSA is presented in Table 6.3, which also includes the same for the exact vacuum expectation values ('zero-particle form factors') obtained in [69, 70]. As discussed above, TCSA matrix elements of one-particle states can be evaluated either taking the ground state from the Neveu-Schwarz and the one-particle state from the Ramond sector or vice versa.

### 6.3.2  Two-particle Form Factors: $F_{11}, F_{22}$ and $F_{13}$

The results for the two-particle form factors are presented on Figures 6.5 and 6.6, showing the same level of agreement as for the order operators. The deviations appearing for rapidities $\theta \gtrsim 1.5$ are again due to the neglected exponential finite size effects. As discussed above, the two-kink states $A_1A_1$ and $A_1A_3$ with even/odd quantum numbers live in the Neveu–Schwarz/Ramond sectors, respectively, while $A_2A_2$ states come in two copies, one in each sectors. In addition, the states $A_1A_3$ appear in even/odd parity pairs: as discussed in Subsection 6.2.2, one of them has vanishing matrix element, while the other one acquires an extra $\sqrt{2}$ factor .

### 6.3.3  $\Delta$-theorem

Another way to verify the form factor solution is to evaluate the sum rule of the $\Delta$-theorem (A.26). Here we perform this calculation in the high temperature phase. Since the expectation value of the order operator vanishes, this makes only sense for the disorder operators $\mu$ and $\mu'$. The contributions to the $\Delta$-theorem from the lowest lying states is presented in Table 6.4: their sum agrees well with the exact result from conformal field theory, the deviations are due to the higher states omitted from the sum. Note that, more relevant the operator is, better the convergence of the sum rule is; this is fully in line with the argument presented in [72] and from the expectations gained from examining the sum rules in other models, such as the integrable perturbations of the Ising field theory [32].

Table 6.3: Vacuum expectation values and one-particle form factors of the disorder operators, TCSA vs. form factor bootsrap. For the one-particle matrix elements obtained from TCSA, the top number correspond to the vacuum state in the Neveu-Schwarz and the particle state in the Ramond sector, while the bottom one is obtained with the opposite choice. For the VEV, there is no such distinction. The digits in parentheses indicate the error in the last digit of the TCSA data, which is estimated from the stability of the exponential fit under the choice of the volume window. Note that the deviations in the subleading magnetisation data are an order of magnitude worse then for the leading one, and also that TCSA estimates of matrix elements involving higher particles are generally less accurate due to truncation errors, as discussed in Appendix C.2. In particular, the apparent large deviation in the entries $F_{34}^{\sigma}(i\pi)$ and $F_{34}^{\sigma'}(i\pi)$ is due to both one-particle states being high in the spectrum, which makes the determination of the corresponding wave functions less accurate.

| Particle $i$ | $|F_i^{\mu}|$ | $|F_i^{\mu,TCSA}|$ | $|F_i^{\mu'}|$ | $|F_i^{\mu',TCSA}|$ |
|---|---|---|---|---|
| 0 (VEV) | 1.44394 | 1.4439(1) | 0.77219 | 0.764(2) |
| 2 | 0.46266 | 0.4626(2) | 2.05131 | 2.045(2) |
|  |  | 0.4622(1) |  | 2.042(1) |
| 4 | 0.19062 | 0.191(1) | 1.36564 | 1.37(1) |
|  |  | 0.184(3) |  | 1.34(1) |
| 5 | 0.10050 | 0.095(2) | 1.00972 | 0.95(2) |
|  |  | 0.097(1) |  | 0.96(1) |

In the low temperature phase, the $\Delta$-theorem can be evaluated for the order (instead of the disorder) fields; due to the Kramers-Wannier duality, this computation is eventually identical to that for the disorder operators in the high temperature phase with the same result as shown in Table 6.4.

Table 6.4: The upper half lists first few contributions to the $\Delta$-theorem, and the lower half compares their sum to the exact conformal weight.

|  | $F_2$ | $F_4$ | $F_5$ | $F_7$ | $F_{11}$ | $F_{22}$ | $F_{13}$ | $F_{24}$ |
|---|---|---|---|---|---|---|---|---|
| $\mu$ | 0.029637 | 0.002437 | 0.000456 | $8.2 \cdot 10^{-7}$ | 0.002731 | 0.000525 | 0.000634 | 0.000297 |
| $\mu'$ | 0.245714 | 0.033396 | 0.008574 | 0.000025 | 0.047420 | 0.013534 | 0.018690 | 0.015048 |

|  | sum | exact |
|---|---|---|
| $\mu$ | 0.0367 | 0.0375 |
| $\mu'$ | 0.3824 | 0.4375 |

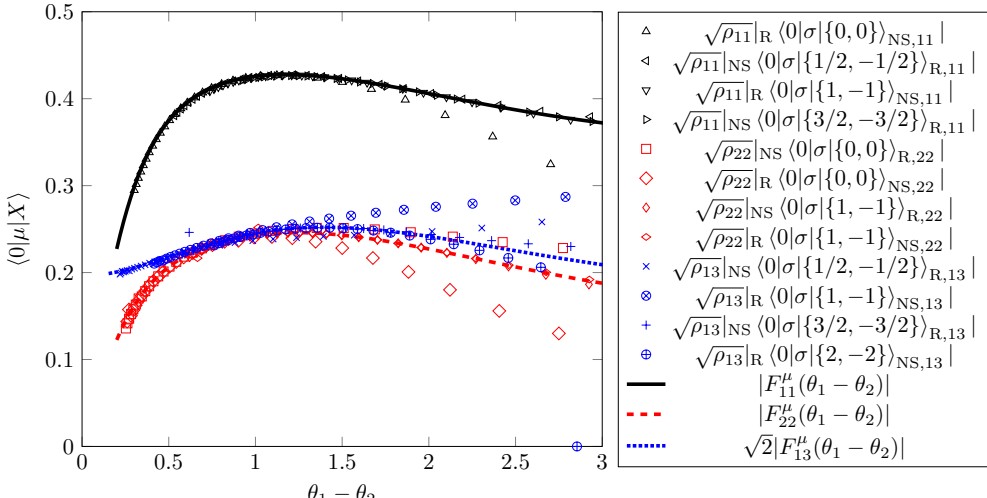

Figure 6.5: Rescaled vacuum-two-particle matrix elements of $\sigma$ coming from TCSA in the low temperature phase plotted against the rapidity difference using discrete markers, and compared to form factors of $\mu$ in the high temperature phase shown with lines. The R/NS index corresponds to the sector from which the two-particle state is extracted; the vacuum state is always the ground state of the other sector. As discussed in Appendix C.2, the main source of error in this comparison are the exponential finite volume corrections to the matrix elements. States with larger Bethe–Yang quantum numbers realise the same rapidity difference in the larger volume, leading to smaller finite size corrections in this comparison.

## 7 Dynamical Structure Factors of the Tricritical Ising Model

In any lattice realisation of the tricritical Ising model, such as the Blume–Capel model (1.2), it is necessary to specify the relation between the spin operator[14] $S(x, t)$ on the lattice and the relevant spin fields which are present in the low energy effective continuum $E_7$ quantum field theory. On a general ground, based on the spin $Z_2$ symmetry of the various fields involved, this relationship is of the form:

$$\mathcal{S}(x,t) = a_\sigma \sigma(x,t) + a_{\sigma'} \sigma'(x,t) + \cdots , \tag{7.1}$$

where the constants $a_{\sigma,\sigma'}$ are specific to the lattice model realisation. The dynamical spin response, $\mathcal{S}(\omega, q)$ that would be measured in a neutron scattering experiment can be expressed in terms of the imaginary part of a retarded spin-spin correlation function:

$$
\begin{aligned}
S(\omega,q) &= \int dx\, dt\, e^{i\omega t - iqx} \langle \mathcal{S}(x,t)\mathcal{S}(0,0)\rangle = \int dx\, dt\, e^{i\omega t - iqx} \sum_{i,j=\sigma,\sigma'} a_i a_j \langle \sigma_i(x,t)\sigma_j(0,0)\rangle \\
&\equiv \sum_{i,j=\sigma,\sigma'} a_i a_j \mathcal{S}^{ij}(\omega,q).
\end{aligned}
\tag{7.2}
$$

The details of the calculation of $\mathcal{S}^{ij}(\omega, q)$ in terms of the form factors of the various fields entering these quantities are presented in Appendix D. Let's make here just a few comments, focusing our attention on the high-temperature phase (as usual, the analysis of the low-temperature is done interchanging the role of order and disorder operators); notice that, contrary to the Ising model, in the TIM all operators have one- and two- particle contributions. The order

---

[14]This is the actual operator that couples to the neutrons in the scattering experiments.

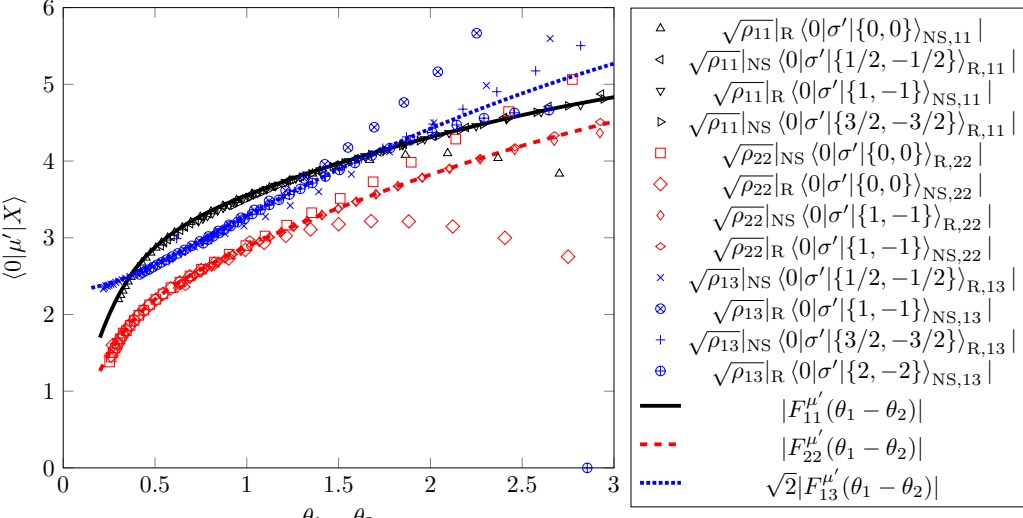

Figure 6.6: Rescaled vacuum-two-particle matrix elements of $\sigma'$ coming from TCSA in the low temperature phase plotted against the rapidity difference using discrete markers, and compared to form factors of $\mu'$ in the high temperature phase shown with lines. The R/NS index corresponds to the sector from which the two-particle state is extracted; the vacuum state is always the ground state of the other sector. As discussed in Appendix C.2, the main source of error in this comparison are the exponential finite volume corrections to the matrix elements. States with larger Bethe–Yang quantum numbers realise the same rapidity difference in the larger volume, leading to smaller finite size corrections in this comparison.

parameters have contributions from odd states while the disorder operators from the even particle states. These terms in the dynamical structure factors can be evaluated combining the results of the previous sections and Appendix D.

The weights of one-particle Dirac-$\delta$ contributions for the order and disorder operators are listed in Tables 7.1 and 7.2, respectively. The various two-particle contributions can also be evaluated using the formula (D.6) presented in Appendix D, and are plotted in Figures 7.1, 7.2 and 7.3, while their numerical values are collected in Table 7.3. The behaviour of the contribution $\mathcal{S}_2^{ij}$ just above the threshold $\omega = m_i + m_j$ is given by

$$S_2^{ij}(\omega, q = 0) = \mathcal{C}_{ij}\left[\omega - (m_i + m_j)\right]^{1/2 - \delta_{i,j}} + O\left(\left[\omega - (m_i + m_j)\right]^{3/2 - \delta_{i,j}}\right), \qquad (7.3)$$

where $\mathcal{C}_{ij}$ is a constant.

Comparing these results to the DSF in the Ising model discussed in Section 2.4, we can see significant differences which can be used to distinguish them in experimental signatures. First,

Table 7.1: One-particle weights in the contributions to the DSF $\mathcal{S}^{ij}$, $i, j = \sigma, \sigma'$.

| particle | $2\pi\dfrac{\|F_i^\sigma\|^2}{m_i}$ | $2\pi\dfrac{\|F_i^{\sigma'}\|^2}{m_i}$ | $2\pi\dfrac{F_i^\sigma F_i^{\sigma'*}}{m_i}$ |
|---|---|---|---|
| 1 | 3.17116 | 26.5578 | 9.1771 |
| 3 | 0.212838 | 9.82114 | 1.44579 |
| 6 | 0.0095296 | 1.29193 | 0.110957 |

Table 7.2: One-particle weights in the contributions to the DSF $\mathcal{S}^{ij}$, $i, j = \mu, \mu'$.

| particle | $2\pi \dfrac{|F_i^\mu|^2}{m_i}$ | $2\pi \dfrac{|F_i^{\mu'}|^2}{m_i}$ | $2\pi \dfrac{F_i^\mu F_i^{\mu'*}}{m_i}$ |
|---|---|---|---|
| 2 | 1.04617 | 20.5658 | 4.63846 |
| 4 | 0.115915 | 6.22406 | 0.849389 |
| 5 | 0.0250625 | 2.52991 | 0.25805 |
| 7 | 0.000566863 | 0.146462 | 0.00911174 |

the Ising DSF shows a signal from a single excitation, while in the TIM there is a complicated spectrum corresponding to different masses and their combinations, with their ratios predicted by the spectrum $E_7$ scattering theory (cf. Table 4.1).

The threshold behaviour is also significantly different. For the Ising model, the contributions coming from multi-particle continua vanishes as a power at the threshold. However, in the TIM there are two types of two-particle thresholds:

- For thresholds with identical particles $A_a A_a$, the behaviour is similar to the Ising case. This is due to the behaviour $S_{aa}(\theta = 0) = -1$ of the scattering amplitude, which due to (A.6) implies that the two-particle form factor vanishes at the threshold:

$$F_{aa}^{min}(\theta = 0) = 0. \tag{7.4}$$

  From (D.6) this leads to the two-particle contribution vanishing at the threshold.

- For thresholds with different particles $A_a A_b$, $a \neq b$, the two-particle contribution diverges at the threshold. In this case $S_{ab}(\theta = 0) = +1$ and $F_{ab}^{min}(\theta = 0)$ has a finite value. As a result, the zero of the denominator $|\sinh(\theta_1 - \theta_2)|$ in (D.6) leads to singular behaviour at the threshold.

As discussed in Subsection 4.2, in an experimental realisation of the spin system, the large scale dynamics depends on the relation between the temperature and vacancy couplings, which can be tuned using suitable experimental parameters. As a function of the parameters, the system is expected to exhibit a crossover from the simple DSF computed in Subsection 2.4 to the DSF characteristic of the TIM. When the parameters are tuned so that the dynamics corresponds to that of the $E_7$ model, the operator describing the response in an inelastic neutron scattering experiment is a combination of $\sigma$ and $\sigma'$, when the system is in the high temperature phase, or $\mu$ and $\mu'$ in the low temperature phase. These two phases are then clearly distinguishable by their different threshold structures. Note that while the specific amplitudes depend on the particular combination of operators, the locations of one-particle peaks and two-particle thresholds are fixed by the spectrum of the TIM. In moving from an $E_7$ DSF to that of an Ising DSF due to perturbations that break the $E_7$ integrability, we expect that at weak integrability breaking the various delta-function peaks and threshold singularities will shift (if they are at a frequency $\omega < 2m_1$) or broaden (if $\omega > 2m_1$). This reflects, respectively, the shifts in masses and decay processes due to weak integrability breaking discussed in the later part of Subsection 4.2.

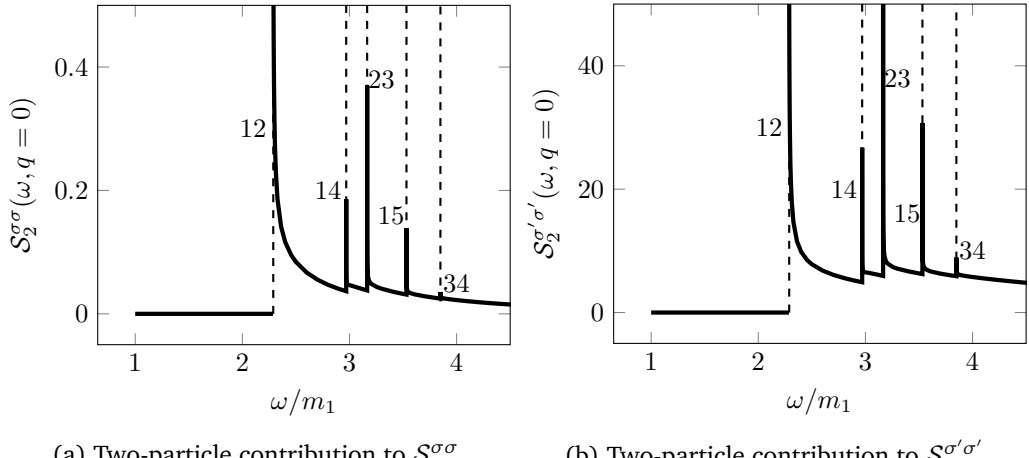

(a) Two-particle contribution to $\mathcal{S}^{\sigma\sigma}$

(b) Two-particle contribution to $\mathcal{S}^{\sigma'\sigma'}$

Figure 7.1: Two-particle contributions to the dynamical structure factor of the leading and subleading magnetisation operators. The values of the dynamical structure factors are shown in units of the massgap $m_1$ using the exact VEV (5.21).

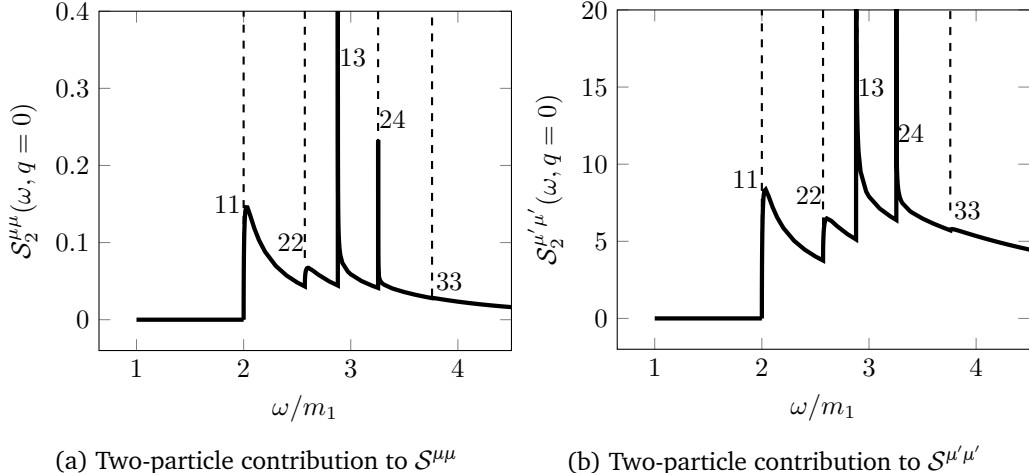

(a) Two-particle contribution to $\mathcal{S}^{\mu\mu}$

(b) Two-particle contribution to $\mathcal{S}^{\mu'\mu'}$

Figure 7.2: Two-particle contributions to the dynamical structure factor of the leading and subleading disorder operators. The values of the dynamical structure factors are shown in units of the massgap $m_1$ using the exact VEV (5.21).

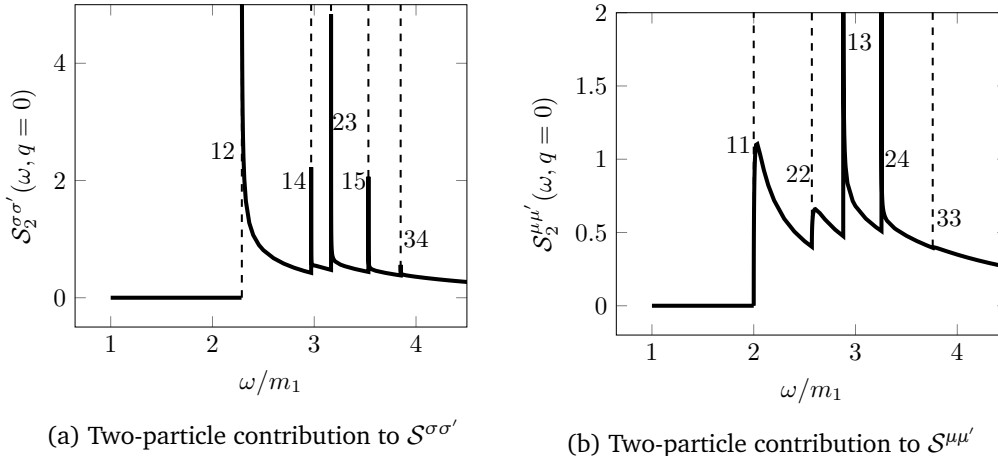

(a) Two-particle contribution to $\mathcal{S}^{\sigma\sigma'}$

(b) Two-particle contribution to $\mathcal{S}^{\mu\mu'}$

Figure 7.3: Two-particle contributions to the mixed dynamical structure factors of the leading and subleading magnetisation and disorder operators.

Table 7.3: Two-particle contributions to various dynamical structure factors calculated from the form-factor solution presented in the previous subsections.

| $\omega$ | $\mathcal{S}_2^{\sigma\sigma}$ | $\mathcal{S}_2^{\sigma'\sigma'}$ | $\mathcal{S}_2^{\mu\mu}$ | $\mathcal{S}_2^{\mu'\mu'}$ | $\mathcal{S}_2^{\sigma\sigma'}$ | $\mathcal{S}_2^{\mu\mu'}$ |
|---|---|---|---|---|---|---|
| 2.1 | 0 | 0 | 0.118391 | 7.22124 | 0 | 0.924625 |
| 2.4 | 0.116344 | 9.84403 | 0.057359 | 4.45951 | 1.07018 | 0.505759 |
| 2.7 | 0.055504 | 6.11336 | 0.058118 | 5.99721 | 0.58250 | 0.589421 |
| 3. | 0.046533 | 6.51685 | 0.058922 | 7.89907 | 0.55057 | 0.679920 |
| 3.3 | 0.040778 | 7.11038 | 0.046481 | 7.62455 | 0.53811 | 0.591966 |
| 3.6 | 0.031957 | 6.71276 | 0.032408 | 6.15006 | 0.46253 | 0.442530 |
| 3.9 | 0.024489 | 6.03023 | 0.025383 | 5.55370 | 0.38338 | 0.370885 |
| 4.2 | 0.018951 | 5.34815 | 0.020103 | 4.95863 | 0.31722 | 0.310682 |
| 4.5 | 0.015140 | 4.82431 | 0.016282 | 4.45800 | 0.26889 | 0.264021 |
| 4.8 | 0.012338 | 4.37690 | 0.013441 | 4.03219 | 0.23081 | 0.227167 |
| 5.1 | 0.010227 | 3.99007 | 0.011279 | 3.66613 | 0.20027 | 0.197559 |
| 5.4 | 0.008604 | 3.65296 | 0.009598 | 3.34868 | 0.17540 | 0.173419 |
| 5.7 | 0.007333 | 3.35720 | 0.008268 | 3.07131 | 0.15491 | 0.153482 |
| 6. | 0.006322 | 3.09619 | 0.007199 | 2.82736 | 0.13782 | 0.136829 |

## 8 Conclusions

In this paper we addressed the computation of the lowest form factors of the order and disorder operators in the tricritical Ising model. Beside the intrinsic interest in performing such a computation, there is an additional practical interest which consists of determining exactly the dynamical structure factors of the model, i.e. the set of quantities directly related to scattering experiments. These functions display a rich spectroscopy associated to the various energy thresholds ruled by the $E_7$ structure of the model and greatly differ from the behavior of similar quantities in the closest class of universality represented by the Ising model: therefore they are the ideal diagnosis to identify the class of universality of the tricritical Ising model. As commented in the text (see Section 4.2), interestingly enough the low part of the $E_7$ spectrum is sufficiently robust to persist to the breaking of integrability by means of the density operator and to display consequently the singularities of the dynamical structure factors associated to the lowest mass excitations.

The computation of the form factors was also very instructive from the genuine theoretical quantum field theory point of view: indeed, the tricritical Ising model provides a non-trivial example of counting the dimension of the operator space of relevant operators sharing the same symmetry but having different conformal properties. In the tricritical Ising model there are in fact two order operators $\sigma$ and $\sigma'$, accompanied by their dual disorder operators $\mu$ and $\mu'$. Based on the exact $S$-matrix in the high-low temperature phase of the model, which presents a rich structure of bound states and symmetries related to the exceptional algebra $E_7$, we showed that the resulting linear form factor equations have the right co-dimension to take into account the presence of two order operators and their duals. Moreover, employing the cluster equations for the form factors, we were also able to show that the only parameters which fix completely the operator content in the odd sector of the $\mathbb{Z}_2$ spin reversal symmetry are nothing else but the VEVs of the two disorder operators. The explicit expressions of the lowest FF of the order/disorder operators have been successfully checked versus their numerical determination obtained in terms of the truncated conformal space approach.

## Acknowledgments

We are grateful to Zhe Wang for the clarification of their experimental work. GM acknowledges the grant Prin 2017-FISI. The work of ML was supported by the National Research Development and Innovation Office of Hungary under the postdoctoral grant PD-19 No. 132118. GT and ML were partially supported by the National Research, Development and Innovation Office (NKFIH) through the OTKA Grant K 138606, and also within the Quantum Information National Laboratory of Hungary. This work was also partially supported by the CNR/MTA Italy-Hungary 2019-2021 Joint Project "Strongly interacting systems in confined geometries". R.M.K. was supported by the U.S. Department of Energy, Office of Basic Energy Sciences, under Contract No. DE-AC02-98CH10886.

## A Form Factor Equations

In this Appendix we briefly summarise the basic functional equations which rule the matrix elements of local fields in an integrable quantum field theory. The reader may consult [32,73, 74] for further details. First of all, the form factors (FF) of the operator $\Phi(x)$ are defined as

$$F^{\Phi}_{a_1,\dots,a_n}(\theta_1,\dots,\theta_n) = \langle 0|\Phi(0)|A_{a_1}(\theta_1),\dots,A_{a_n}(\theta_n)\rangle. \tag{A.1}$$

**Monodromy properties:** For a scalar operator $\Phi(x)$, relativistic invariance requires that its FF depend only on the rapidity differences $\theta_i - \theta_j$. Based on the elasticity of the scattering processes, the completeness of the asymptotic states and transformation under the crossing transformation, the FF satisfy the monodromy equations

$$\begin{aligned}
F^{\Phi}_{a_1,..,a_i,a_{i+1},..a_n}(\theta_1,..,\theta_i,\theta_{i+1},..,\theta_n) &= S_{a_i a_{i+1}}(\theta_i - \theta_{i+1}) F^{\Phi}_{a_1,..a_{i+1},a_i,..a_n}(\theta_1,..,\theta_{i+1},\theta_i,..,\theta_n), \\
F^{\Phi}_{a_1,a_2,...a_n}(\theta_1 + 2\pi i,\ldots,\theta_{n-1},\theta_n) &= F^{\Phi}_{a_2,a_3,...,a_n,a_1}(\theta_2,\ldots,\theta_n,\theta_1).
\end{aligned}$$

(A.2)

**Simple Poles and Recursive Equations:** The FF have pole singularities related to those of the $S$-matrix. Let's discuss firstly the simple poles, which are of two types. The first type are kinematical poles related to the annihilation processes of a pair of particle and anti-particle states. These singularities are located at $\theta_a - \theta_{\bar{a}} = i\pi$ and for the corresponding residue gives rise to a recursive equation between the $(n+2)$-particle FF and the $n$-particle FF

$$-i \lim_{\tilde{\theta}\to\theta}(\tilde{\theta}-\theta)F^{\Phi}_{a,\bar{a},a_1,...,a_n}(\tilde{\theta}+i\pi,\theta,\theta_1,\ldots,\theta_n)$$
$$= \left(1 - e^{2\pi i \omega_a}\prod_{j=1}^n S_{a,a_j}(\theta - \theta_j)\right)F^{\Phi}_{a,...,a_n}(\theta_1,\ldots,\theta_n),$$

(A.3)

where $\omega_a$ is the index of mutual semi-locality of the operator $\Phi$ with respect to the particle $A_a$.

A second type of simple poles is related to the presence of bound states appearing as simple poles in the $S$-matrix. If $\theta = iu^c_{ab}$ and $\Gamma^c_{ab}$ are the resonance angle and the three-particle coupling of the fusion $A_a \times A_b \to A_c$ respectively, then FF involving the particles $A_a$ and $A_b$ also has a pole at $\theta = iu^c_{ab}$ and its residue gives rise to a recursive equation between the $(n+2)$-particle FF and the $(n+1)$-particle FF

$$-i \lim_{\theta_{ab}\to iu^c_{ab}}(\theta_{ab} - iu^c_{ab})F^{\Phi}_{a,b,a_i,...,a_n}(\theta_a,\theta_b,\theta_1,\ldots,\theta_n) = \Gamma^c_{ab}F^{\Phi}_{c,a_i,...,a_n}(\theta_c,\theta_1,\ldots,\theta_n),$$

(A.4)

where $\theta_c = (\theta_a \bar{u}^a_{bc} + \theta_b \bar{u}^b_{ac})/u^c_{ab}$. In general, the FF may also have higher-order poles but, in order to address them, let's first discuss the general way of parameterising the FF and the special role played by the 2-particle FF.

A third kind of simple poles in the FF are related to double poles in the S-matrix which reflect multi-particle scattering processes. For a double pole in the S-matrix $S_{ab}$ located at an angle $\phi \in (0,\pi)$ which can be written $\phi = u^c_{ad} + u^e_{bd} - \pi$ for some $c$, $d$ and $e$, the FF has a simple pole at $\theta = i\phi$ [32]. For a two-particle FF, the corresponding residue is given by

$$-i \lim_{\theta_{ab}\to i\phi}(\theta_{ab} - i\phi)F^{\varphi}_{ab}(\theta_{ab}) = \Gamma^c_{ad}\Gamma^e_{bd}F^{\varphi}_{ce}(i\gamma),$$

(A.5)

where $\gamma - \pi - u^a_{cd} - u^b_{de}$.

In general, form factors can also have higher order poles, corresponding to third and higher order poles in the S-matrix. However, here we omit the equations satisfied by the residues at these poles, since they are not required in our calculation.

**2-particle FF:** Two-particle FF play a crucial role in the bootstrap approach since they provide the initial conditions for solving the recursive equations and they also encode all the basic properties that the matrix elements with higher number of particles inherit by factorisation property, such as the asymptotic behaviour and the analytic structure. So, let's see the main properties of the 2-particle FF $F^{\Phi}_{ab}(\theta)$. First of all, this is a meromorphic function of the rapidity difference defined in the strip $\mathcal{S} = Im\,\theta \in (0,\pi)$, with its monodromy dictated by the equations (A.2)

$$F^{\Phi}_{ab}(\theta) = S_{ab}(\theta)F^{\Phi}_{ab}(-\theta),$$

(A.6)

$$F_{ab}^{\Phi}(i\pi + \theta) = F_{ab}^{\Phi}(i\pi - \theta). \tag{A.7}$$

If $F_{ab}^{min}(\theta)$ denotes a solution of these equations, free of poles and zeros in the strip $\mathcal{S}$, the general solution $F_{ab}^{\Phi}(\theta)$ can be written as

$$F_{ab}^{\Phi}(\theta) = \frac{Q_{ab}^{\Phi}(\theta)}{D_{ab}(\theta)} F_{ab}^{min}(\theta), \tag{A.8}$$

where $D_{ab}(\theta)$ and $Q_{ab}^{\Phi}(\theta)$ are polynomials in $\cosh\theta$: the former is fixed by the singularity structure of $S_{ab}(\theta)$ while the latter carries the whole information about the operator $\Phi(x)$. For excitations which are non-local with respect to the operator $\Phi$, the form factor above will include an extra term $\cosh\theta/2$ (either in the numerator or in the denominator, according to the asymptotic behavior in $\theta$ of the form factor) which is even under $\theta \to -\theta$ but changes a sign under the transformation $\theta \to \theta + 2\pi i$ which probes the mutual non-locality of $\Phi$ wrt the excitations. See, for instance, the form factor expressions (5.15 relative to the disorder operators of the TIM wrt excitations which are kinks in the low-temperature phase.

**Minimal form factor:** The minimal form factors for the thermal deformation of the tricritical Ising model have the general expression

$$F_{ab}^{min}(\theta) = \left(-i\sinh\frac{\theta}{2}\right)^{\delta_{ab}} \prod_{\alpha\in\mathcal{A}_{ab}} (g_\alpha(\theta))^{p_\alpha}, \tag{A.9}$$

where

$$g_\alpha(\theta) = \exp\left\{2\int_0^\infty \frac{dt}{t} \frac{\cosh\left(\frac{\alpha}{18}-\frac{1}{2}\right)t}{\cosh\frac{t}{2}\sinh t}\sin^2\frac{(i\pi-\theta)t}{2\pi}\right\}. \tag{A.10}$$

For large values of the rapidity ($|\theta| \to \infty$), by saddle point evaluation it is easy to see that this function has the following asymptotical behavior

$$g_\alpha(\theta) \sim \mathcal{N}_\alpha \exp\left\{\frac{|\theta|}{2}-\frac{i\pi}{2}\right\}, \tag{A.11}$$

where

$$\mathcal{N}_\alpha(\theta) = \exp\left\{\int_0^\infty \frac{dt}{t}\left[\frac{\cosh\left(\frac{\alpha}{18}-\frac{1}{2}\right)t}{\cosh\frac{t}{2}\sinh t}-\frac{1}{t^2}\right]\right\}. \tag{A.12}$$

The function $g_\alpha(\theta)$ has an infinite number of poles outside the physical strip, these are shown explicitly in the infinite-product representation:

$$g_\alpha(\theta) = \prod_{k=0}^{\infty}\left[\frac{\left[1+\left(\dfrac{\widehat{\theta}/2\pi}{k+1-\dfrac{\alpha}{36}}\right)^2\right]\left[1+\left(\dfrac{\widehat{\theta}/2\pi}{k+\dfrac{1}{2}+\dfrac{\alpha}{36}}\right)^2\right]}{\left[1+\left(\dfrac{\widehat{\theta}/2\pi}{k+1+\dfrac{\alpha}{36}}\right)^2\right]\left[1+\left(\dfrac{\widehat{\theta}/2\pi}{k+1-\dfrac{3}{2}-\dfrac{\alpha}{36}}\right)^2\right]}\right]^{k+1},$$

where $\widehat{\theta} = i\pi - \theta$. The mixed representation:

$$
\begin{aligned}
g_\alpha(\theta) \;=\; & \prod_{k=0}^{N-1} \left[ \frac{\left[1 + \left(\dfrac{\widehat{\theta}/2\pi}{k+1-\dfrac{\alpha}{36}}\right)^2\right] \left[1 + \left(\dfrac{\widehat{\theta}/2\pi}{k+\dfrac{1}{2}+\dfrac{\alpha}{36}}\right)^2\right]}{\left[1 + \left(\dfrac{\widehat{\theta}/2\pi}{k+1+\dfrac{\alpha}{36}}\right)^2\right] \left[1 + \left(\dfrac{\widehat{\theta}/2\pi}{k+1-\dfrac{3}{2}-\dfrac{\alpha}{36}}\right)^2\right]} \right]^{k+1} \\
& \times \exp\left[2\int_0^\infty \frac{dt}{t} \frac{\cosh\left[t\left(\dfrac{1}{2}-\dfrac{\alpha}{18}\right)\right]}{\cosh\dfrac{t}{2}\sinh t}(N+1-Ne^{-2t})e^{-2Nt}\sin^2\frac{\widehat{\theta}t}{2\pi}\right],
\end{aligned}
$$

is particularly useful for numerical computations.

**Pole structure of the 2-particle FF:** The pole terms entering the general parameterisation (A.8) are in correspondence with the pole structure of the $S$-matrix and they can be expressed as

$$
D_{ab}(\theta) = \prod_{\alpha \in \mathcal{A}_{ab}} (\mathcal{P}_\alpha(\theta))^{i_\alpha} (\mathcal{P}_{1-\alpha}(\theta))^{j_\alpha}, \tag{A.13}
$$

where

$$
\begin{aligned}
i_\alpha = n+1, \quad & j_\alpha = n, \quad \text{if} \quad p_\alpha = 2n+1; \\
i_\alpha = n, \quad\;\; & j_\alpha = n, \quad \text{if} \quad p_\alpha = 2n,
\end{aligned} \tag{A.14}
$$

and we have introduced the notation

$$
\mathcal{P}_\alpha(\theta) \equiv \frac{\cos\pi\dfrac{\alpha}{18} - \cosh\theta}{2\cos^2\dfrac{\pi\alpha}{36}}. \tag{A.15}
$$

Notice that both $F_{ab}^{min}(\theta)$ and $D_{ab}(\theta)$ are normalised to 1 in $\theta = i\pi$.

**Bound on the asymptotic behaviour of the FF:** The polynomial $Q_{ab}^\Phi(\theta)$ is the term that contains the information about the operator $\Phi$. Such a quantity can be written as

$$
Q_{ab}^\Phi(\theta) \equiv \sum_{k=0}^{N_{ab}} a_{ab}^k \cosh^k\theta, \tag{A.16}
$$

and its degree $N_{ab}$ is determined by the upper bound on the asymptotic behaviour of the FF of the operator $\Phi(x)$: if $\Delta_\Phi$ is the conformal dimension of the scalar operator $\Phi(x)$ then its FF satisfy

$$
\lim_{|\theta_i|\to\infty} F_{a_1,\ldots,a_n}^\Phi(\theta_1,\ldots,\theta_n) \sim e^{y_\Phi|\theta_i|}, \tag{A.17}
$$

with

$$
y_\Phi \leq \Delta_\Phi. \tag{A.18}
$$

Since each polynomial $\mathcal{P}_\alpha(\theta)$ goes asymptotically as $e^\theta$ and the minimal form factor as shown in eq. (A.11), it is then easy to determine from the bound (A.17) the maximum degree $N_{ab}$ of

the polynomial $Q_{ab}(\theta)$. Notice that for the trace $\Theta(x)$ of the stress-energy tensor, the conservation law of this tensor implies that the polynomial $Q_{ab}(\theta)$ can be factorised as

$$Q_{ab}^{\Theta}(\theta) = \left( \cosh \theta + \frac{m_a^2 + m_b^2}{2 m_a m_b} \right)^{1-\delta_{ab}} P_{ab}(\theta), \tag{A.19}$$

where

$$P_{ab}(\theta) \equiv \sum_{k=0}^{N_{ab}'} a_{ab}^k \cosh^k \theta. \tag{A.20}$$

Moreover, for the diagonal elements $F_{aa}^{\Theta}$ we have the normalisation

$$F_{aa}^{\Theta}(i\pi) = \langle A_a(\theta_a) | \Theta(0) | A_a(\theta_a) \rangle = 2\pi m_a^2. \tag{A.21}$$

**Cluster property:** In a massive field theory, the form factors of relevant operators $\Phi_i$ are expected to satisfy[15] the asymptotic factorisation [25, 57, 58]

$$\lim_{\alpha \to \infty} F_{r+l}^{\Phi_a}(\theta_1 + \alpha, \ldots, \theta_r + \alpha, \theta_{r+1}, \ldots, \theta_{r+l})$$
$$= F_r^{\Phi_b}(\theta_1, \ldots, \theta_r) F_l^{\Phi_c}(\theta_{r+1}, \ldots, \theta_{r+l}), \tag{A.22}$$

where $\Phi_a, \Phi_b, \Phi_c$ label fields of the same conformal dimension. The origin of this identity simply comes from the fact that both functions in the right hand side satisfy the same form factor axioms and therefore are defining matrix elements of some operators $\Phi_b$ and $\Phi_c$ with an appropriate normalization.

For the case of order/disorder operators of the Ising model in the high-temperature phase, given the $\mathbb{Z}_2$ symmetry of these operators, the splitting of their form factors is simply determined by the even or odd number of particles: in case of the disorder operator $\mu$, splitting the set of particles into two sets of odd number of particles we have[16]

$$\lim_{\alpha \to \infty} F_{2r+2l+2}^{\mu}(\theta_1 + \alpha, \ldots, \theta_{2r+1} + \alpha, \theta_{2r+2}, \ldots, \theta_{2r+2l+2})$$
$$= \frac{1}{\langle \mu \rangle} F_{2r+1}^{\sigma}(\theta_1, \ldots, \theta_{2r+1}) F_{2l+1}^{\sigma}(\theta_{2r+2}, \ldots, \theta_{2r+2l+2}), \tag{A.23}$$

while for splitting into two sets of even number of particles we have

$$\lim_{\alpha \to \infty} F_{2r+2l}^{\mu}(\theta_1 + \alpha, \ldots, \theta_{2r} + \alpha, \theta_{2r+1}, \ldots, \theta_{2r+2l})$$
$$= \frac{1}{\langle \mu \rangle} F_{2r}^{\mu}(\theta_1, \ldots, \theta_{2r}) F_{2l}^{\mu}(\theta_{2r+1}, \ldots, \theta_{2r+2l}). \tag{A.24}$$

It is straightforward to generalise these relations to the tricritical Ising model based on the same symmetry property of the order/disorder operators and the particle excitations.

**Sum rules:** Here we recall two important sum rules for the correlation functions of off-critical models. Both of them involve the trace of the stress-energy tensor, the scalar field $\Theta(x)$. The first one is the c-theorem [75, 76], which enables us to extract the central charge of the ultraviolet CFT model in terms of the second moment of the off-critical two-point correlation function of the $\Theta(x)$ field

$$c = \frac{3}{2} \int_0^{\infty} dr \, r^3 \, \langle \Theta(r) \Theta(0) \rangle, \tag{A.25}$$

---

[15]Up to phases related to the normalization of the operators.

[16]In the following we have decided to normalize the FF in terms of the VEV of the disorder operator and this is the reason of $\langle \mu \rangle$ in the formulas below.

while the second one is the $\Delta$-theorem, which gives the ultraviolet scaling dimension of scaling fields [58]

$$\Delta_{\Phi}^{\text{uv}} = -\frac{1}{2\langle\Phi\rangle} \int_0^{\infty} dr \, r \, \langle\Theta(r)\Phi(0)\rangle. \tag{A.26}$$

Both sum rules can be efficiently evaluated in terms of the form factors of the various fields involved and the corresponding series thereof, relying on the rapidly convergent nature of the series [72].

# B Exact $S$-matrix of the High/Low Temperature Phases of the Tricritical Ising Model

In two dimensional space-time, the most convenient way to parameterise the relativistic dispersion relations of a massive particle $A_a$ (of mass $m_a$) is in terms of the rapidity variable, $E = p^{(0)} = m_a \cosh\theta_a$, $p = p^{(1)} = m_a \sinh\theta_a$. In a two-body scattering process involving the particles $A_a$ and $A_b$ there are two independent relativistic invariant quantities: these are the Mandelstam variables $s$ and $t$ given by

$$
\begin{aligned}
s &= (p_a + \mathbf{p}_b)^2 = m_a^2 + m_b^2 + 2m_a m_b \cosh(\theta_a - \theta_b), \\
t &= (\mathbf{p}_a - \mathbf{p}_b)^2 = m_a^2 + m_b^2 - 2m_a m_b \cosh(\theta_a - \theta_b).
\end{aligned}
\tag{B.1}
$$

Denoting the rapidity difference as $\theta \equiv \theta_a - \theta_b$, notice that the Mandelstam variable $t$ is simply obtained from $s$ in terms of the analytic continuation $\theta \to i\pi - \theta$, which is called *crossing transformation*. Given the definitions above, the exact and elastic two-body $S$-matrix amplitudes of the high and low temperature phases of the tricritical Ising model can be written as [22, 23]

$$S_{ab}(\theta) = \prod_{\alpha \in \mathcal{A}_{ab}} (f_\alpha(\theta))^{p_\alpha}, \tag{B.2}$$

where

$$f_\alpha(\theta) \equiv \frac{\tanh\frac{1}{2}\left(\theta + i\pi\frac{\alpha}{18}\right)}{\tanh\frac{1}{2}\left(\theta - i\pi\frac{\alpha}{18}\right)}. \tag{B.3}$$

For a given pair $A_a$ and $A_b$ of the asymptotic particles, the various integer numbers $\alpha = 1/, \ldots, 17$ give the location of the various poles of the S-matrix (in unit of $\pi/18$), while the integer numbers $p_\alpha$ give the multiplicity of each of these poles. Poles of odd order usually correspond to bound states while those of even order correspond to multi-particle scattering. The set of the $\alpha$'s of the various channels is given below, where we use the notation

$$\overset{\mathbf{a}\ \ p_\alpha}{(\alpha)}$$

to denote the location of the pole $\alpha$, of multiplicity $p_\alpha$, corresponding to the bound state $A_a$. The eventual minus sign in some of the amplitudes means that the corresponding product (B.2) must be multiplied by a $-$ sign.

In order to implement the form factor bootstrap Equations we need the values of the on-shell three-particle couplings[17] $\Gamma_{ab}^c = \Gamma_{abc}$, given by the residues of the $S$-matrix on the poles of the ground states. Their values can be determined up to a sign, and are reported in Table B.2. In our calculations we assume that they are positive, unless otherwise stated. Their eventual

---

[17]These quantities are perfectly symmetric in the three indices.

Table B.1: $S$-matrix amplitudes in the $E_7$ factorised scattering theory

| $a$ | $b$ | $S_{ab}$ | $a$ | $b$ | $S_{ab}$ |
|---|---|---|---|---|---|
| 1 | 1 | $-\overset{2}{(10)}\,\overset{4}{(2)}$ | 3 | 4 | $\overset{1}{(15)}\,(5)^2(7)^2(9)$ |
| 1 | 2 | $\overset{1}{(13)}\,\overset{3}{(7)}$ | 3 | 5 | $\overset{1}{(16)}\,\overset{6}{(10)}{}^3(4)^2(6)^2$ |
| 1 | 3 | $-\overset{2}{(14)}\,\overset{4}{(10)}\,\overset{5}{(6)}$ | 3 | 6 | $-\overset{2}{(16)}\,\overset{5}{(12)}{}^3\overset{7}{(8)}{}^3(4)^2$ |
| 1 | 4 | $\overset{1}{(17)}\,\overset{3}{(11)}\,\overset{6}{(3)}\,(9)$ | 3 | 7 | $\overset{3}{(17)}\,\overset{6}{(13)}{}^3(3)^2(7)^4(9)^2$ |
| 1 | 5 | $\overset{3}{(14)}\,\overset{6}{(8)}\,(6)^2$ | 4 | 4 | $\overset{4}{(12)}\,\overset{5}{(10)}{}^3\overset{7}{(4)}\,(2)^2$ |
| 1 | 6 | $-\overset{4}{(16)}\,\overset{5}{(12)}\,\overset{7}{(4)}\,(10)^2$ | 4 | 5 | $\overset{2}{(15)}\,\overset{4}{(13)}{}^3\overset{7}{(7)}{}^3(9)$ |
| 1 | 7 | $\overset{6}{(15)}\,(9)(5)^2(7)^2$ | 4 | 6 | $\overset{1}{(17)}\,\overset{6}{(11)}{}^3(3)^2(5)^2(9)^2$ |
| 2 | 2 | $\overset{2}{(12)}\,\overset{4}{(8)}\,\overset{5}{(2)}$ | 4 | 7 | $\overset{4}{(16)}\,\overset{5}{(14)}{}^3(6)^4(8)^4$ |
| 2 | 3 | $\overset{1}{(15)}\,\overset{3}{(11)}\,\overset{6}{(5)}\,(9)$ | 5 | 5 | $\overset{5}{(12)}{}^3(2)^2(4)^2(8)^4$ |
| 2 | 4 | $\overset{2}{(14)}\,\overset{5}{(8)}\,(6)^2$ | 5 | 6 | $\overset{1}{(16)}\,\overset{3}{(14)}{}^3(6)^4(8)^4$ |
| 2 | 5 | $\overset{2}{(17)}\,\overset{4}{(13)}\,\overset{7}{(3)}\,(7)^2(9)$ | 5 | 7 | $\overset{2}{(17)}\,\overset{4}{(15)}{}^3\overset{7}{(11)}{}^5(5)^4(9)^3$ |
| 2 | 6 | $\overset{3}{(15)}\,(7)^2(5)^2(9)$ | 6 | 6 | $-\overset{4}{(14)}{}^3\overset{7}{(10)}{}^5(12)^4(16)^2$ |
| 2 | 7 | $\overset{5}{(16)}\,\overset{7}{(10)}{}^3(4)^2(6)^2$ | 6 | 7 | $\overset{1}{(17)}\,\overset{3}{(15)}{}^3\overset{6}{(13)}{}^5(5)^6(9)^3$ |
| 3 | 3 | $-\overset{2}{(14)}\,\overset{7}{(2)}\,(8)^2(12)^2$ | 7 | 7 | $\overset{2}{(16)}{}^3\overset{5}{(14)}{}^5\overset{7}{(12)}{}^7(8)^8$ |

value depends on the phase convention for the multi-particle states. As shown in the main text, while most of them can be fixed to have a positive real value, the consistency of the form factor equations eventually determines some of them to have a negative sign.

# C  Truncated Conformal Space Approach

## C.1  Overview of the method

The truncated conformal space approach (TCSA) [27] consists in studying the numerical spectrum of the off-critical Hamiltonian on a infinite cylinder of circumference $L$ based on the truncated Hilbert space made of the conformal states

$$H = H_0 + V. \tag{C.1}$$

Table B.2: Three-particle couplings $\Gamma_{abc}$ coming from the simple poles of the $S$-matrix amplitudes.

| $abc$ | 112 | 114 | 123 | 134 | 135 | 146 |
|---|---|---|---|---|---|---|
| $|\Gamma_{abc}|$ | 4.83871 | 1.22581 | 7.34688 | 29.0008 | 19.1002 | 4.83871 |
| $abc$ | 156 | 167 | 222 | 224 | 225 | 233 |
| $|\Gamma_{abc}|$ | 114.477 | 29.0008 | 11.1552 | 19.1002 | 1.86121 | 75.3953 |
| $abc$ | 236 | 245 | 257 | 337 | 444 | 447 |
| $|\Gamma_{abc}|$ | 29.0008 | 114.477 | 19.1002 | 7.34688 | 686.12 | 114.477 |

In this expression $H_0$ is the Hamiltonian of the conformal fixed point on the cylinder while

$$V = g \int_0^L dv \, \varphi(w)$$

is the off-critical deformation, where $\varphi(x)$ is a relevant perturbation of conformal dimension $\Delta$. Notice that $w = u + iv$ is the coordinate along the cylinder with $u$ corresponding to time and $v \equiv v + L$ to the spatial direction. The Hamiltonian is defined in the interaction picture corresponding to the decomposition (C.1) at time $u = 0$. Using the conformal transformation $z = e^{\frac{2\pi}{L}w}$, the conformal theory on the cylinder is mapped onto the complex plane and therefore $H_0$ can be expressed in terms of the usual conformal generators $L_0, \overline{L}_0$ and the central charge $c$

$$H_0 = \frac{2\pi}{L}\left(L_0 + \overline{L}_0 - \frac{c}{12}\right), \tag{C.2}$$

while the field on the cylinder $\varphi$ (at time $u = 0$) is related to the field $\widetilde{\varphi}$ on the plane by the conformal transformation

$$\varphi = \left|\frac{2\pi}{L}\right|^{2\Delta} \widetilde{\varphi}. \tag{C.3}$$

The spectrum of $H$ depends on the dimensionless parameter $gR^{2-2\Delta}$ where the value of $g$ can be fixed in such a way that the mass gap is equal to 1. For the thermal deformation of the TIM $\Delta = 1/10$ and the relationship between the mass–gap $m_1$ and the coupling constant $g_2$ is given by [77]

$$m_1 = \mathcal{C}_2 g_2^{\frac{5}{9}}, \qquad \mathcal{C}_2 = \left(\frac{2\Gamma(\frac{2}{9})}{\Gamma(\frac{2}{3})\Gamma(\frac{5}{9})}\right)\left(\frac{4\pi^2\Gamma(\frac{2}{5})\Gamma(\frac{4}{5})^3}{\Gamma(\frac{1}{5})^3\Gamma(\frac{3}{5})}\right)^{5/18} = 3.7453728362\ldots. \tag{C.4}$$

The basis of the conformal Hilbert space is given by states $|m\rangle$ which are eigenstates of the generators $L_0, \overline{L}_0$ with eigenvalues $\Delta_m, \overline{\Delta}_m$. This basis is not necessarily chosen to be orthonormal, which can be accounted for by using the inner product matrix $b_{ml} = \langle m|l\rangle$. Then the matrix elements of the perturbed Hamiltonian can be written as

$$H_{mn} = \frac{2\pi}{L}\left[\left(\Delta_m + \overline{\Delta}_m - \frac{c}{12}\right)\delta_{mn} + 2\pi g\left(\frac{L}{2\pi}\right)^{2(1-\Delta)} b_{mk}^{-1}\langle k|\widetilde{\varphi}|n\rangle\right]. \tag{C.5}$$

Using the relation (C.4) the Hamiltonian can be rewritten in units of the mass gap $m_1$

$$h_{mn} = \frac{2\pi}{l}\left[\left(\Delta_m + \overline{\Delta}_m - \frac{c}{12}\right)\delta_{mn} + 2\pi\kappa\left(\frac{l}{2\pi}\right)^{2(1-\Delta)}b_{mk}^{-1}\langle k|\widetilde{\varphi}|n\rangle\right], \qquad (C.6)$$

with the dimensionless volume variable $l = m_1 L$ and coupling $\kappa = C_2^{-9/5}$. The matrix elements $\langle k|\varphi_i|n\rangle$ can be computed in terms of the structure constants of the OPE and the action of the conformal generators $L_n, \overline{L}_n$ on the states.

Note that due to translational invariance the Hamiltonian commutes with the momentum operator

$$P = \frac{2\pi}{L}\left(L_0 - \overline{L}_0\right). \qquad (C.7)$$

In our computations we only considered the sector with zero total momentum as that was sufficient for our purposes, and introduced a cut-off at chiral descendant level 20 which resulted in a truncated Hilbert space of dimension 623552. Once the Hamiltonian $H$ was diagonalised for different values of the volume $L$, we extracted the spectrum of the low energy eigenvalues and their associated eigenvectors as a function of the volume. Using the masses and scattering amplitudes predicted by the $E_7$ scattering theory, the vacuum and the one- and two-particle energy levels can be easily identified, allowing the numerical determination of vacuum expectation values as well as one–particle and two-particle form factors. For a detailed description of the relevant procedures the reader is referred to [60, 61].

## C.2   Sources of deviations between predicted matrix elements and TCSA results

The two sources of deviations between the numerical TCSA matrix elements and the form factor predictions are due to (1) neglecting exponential finite volume corrections in the analytic predictions (6.5) of finite volume matrix elements, and (2) truncation errors inherent in TCSA. The best agreement between form factor predictions and TCSA numerics is obtained in the so-called scaling regime, which is a range of volumes where exponential finite volume corrections and truncation errors are comparable, and their common magnitude determines the highest available precision.

The main source of deviations in our computations results from neglecting the exponential finite size corrections. In the two-particle form factors shown in Figures 6.3, 6.4, 6.5 and 6.6 it is these corrections that cause the deviations observed for larger rapidities. Note that two-particle states with larger Bethe quantum numbers have a given relative rapidity in larger volume. As a result, data extracted from higher energy levels fit the exact form factor predictions for a longer range in rapidity.

For one-particle form factors in Tables 6.1 and 6.3, and the crossed two-particle form factors in Table 6.2, we extrapolated the TCSA data using an exponential fit in the volume. The results are presented with the digits which were stable under the choice of the volume window used to the fit. The reason behind this procedure is that it is necessary to find a volume window where the volume is large enough (i.e. where a single exponential provides a good description of the finite size effects), but not too large so that truncation effects are small.

Turning now to truncation errors, they typically grow with the volume, and are also the larger the higher energy level is considered. In addition, renormalisation group arguments [78–81] show that their magnitude strongly depends on the conformal weight of the perturbing field. Truncation errors can be mitigated using leading order RG equations [80, 82], which lead to running couplings of operators depending on the perturbing operator and the OPE structure of the theory. The leading correction is the contribution of the identity operator, which only shifts the vacuum energy and has no effect on the eigenvectors. In our calculation we used truncation at chiral descendant level 20, where the corrections to the couplings $g$ and

$\lambda$ are 5 to 6 orders of magnitude smaller then the eventual physical couplings (set at the limit of infinite truncation). As a result, we omitted these corrections in our TCSA determination of form factors.

Despite truncation errors being generally small, one way they appear visibly is the presence of isolated outliers. These are related to the vicinity of level crossings, where the eigenvectors of nearly degenerate levels are extremely sensitive to any small perturbations, and even a small truncation error can have a disproportionately large effect [71].

Truncation errors are also enhanced when higher part of the energy spectrum is involved in a given quantity. For state vectors this implies that their higher components are less reliable, which is apparent in the numerical tables in Subsections 6.2 and 6.3. One particle form factors have rather small errors, since one of the vectors involved is a vacuum state which has dominantly small energy components. Crossed two-particle form factors are expected to have larger errors since higher energy (therefore less reliable) components of the vectors play significant role in the matrix element calculation, especially when both one-particle states are in the higher energy part of the spectrum (e.g. 3 and 4).

Finally we mention that operator matrix elements typically converge slower if the dimension of the operator is larger [83], hence we expect better agreement for the leading magnetisation then for the subleading one, which indeed turns out to be the case.

## D Form Factor Expansion for Dynamical Structure Factors

Considering a set of operators $\mathcal{O}_i$, their two-point functions $\langle \mathcal{O}_i(x, \tau) \mathcal{O}_j(0,0)\rangle$ can be evaluated using a spectral expansion obtained by inserting a resolution of the identity in terms of asymptotic multi-particle states:

$$\mathbb{I} = \sum_{n=1}^{\infty} \sum_{\{a_i\}} \prod_{k=1}^{\mathcal{N}} \left( \frac{1}{N_k^{\{a_i\}}!} \right) \int \frac{d\theta_1}{2\pi} \cdots \frac{d\theta_n}{2\pi} |a_1, \theta_1; \ldots; a_n, \theta_n\rangle \langle a_n, \theta_n; \ldots; a_1, \theta_1|, \qquad \text{(D.1)}$$

which sums over states involving an arbitrary number $n$ of particles with arbitrary rapidities $\theta_i$, $i = 1, \ldots, n$ and particle species labelled by the integers $1 \leq a_i \leq \mathcal{N}$ such that $a_1 \leq a_2 \leq \cdots \leq a_n$, while $N_k^{\{a_i\}}$ is the number of particles of type $k$ in the set $\{a_i\}$.

The Fourier transform of the correlator

$$\mathcal{S}^{ij}(\omega, q) = \int dx\, dt\, e^{i\omega t - iqx} \langle \mathcal{O}_i(x, t) \mathcal{O}_j(0,0)\rangle, \qquad \text{(D.2)}$$

is called the dynamical structure factor (DSF). Using (D.1) it can be written as a sum over $n$-particle contributions:

$$\mathcal{S}^{ij}(\omega, q) = \sum_{n=1}^{\infty} \mathcal{S}_n^{ij}(\omega, q), \qquad \text{(D.3)}$$

$$\mathcal{S}_n^{ij}(\omega, q) = \sum_{\{a_i\}} \prod_{k=1}^{\mathcal{N}} \left( \frac{1}{N_k^{\{a_i\}}!} \right) \int \frac{d\theta_1}{2\pi} \cdots \frac{d\theta_n}{2\pi} F_{a_1, a_2, \ldots, a_n}^{\mathcal{O}_i}(\theta_1, \theta_2, \ldots, \theta_n) F_{a_1, a_2, \ldots, a_n}^{\mathcal{O}_j*}(\theta_1, \theta_2, \ldots, \theta_n)$$

$$\times (2\pi)^2 \delta\left( \omega - \sum_{i=1}^{n} E_i \right) \delta\left( q - \sum_{i=1}^{n} P_i \right). \quad \text{(D.4)}$$

Although it is generally not possible to evaluate all terms in this infinite sum, it typically converges rapidly and most of the spectral weight comes from the first few contributions with

small number $n$ of particles. Furthermore, in a gapped theory truncating the sum leads to an exact result at low energies ($\omega$) due to the presence of energy thresholds. Here we consider the dynamical structure factor $\mathcal{S}_{ij}(\omega, q)$ for $q = 0$, and consider the contributions of the $n = 1, 2$ and 3-particle form factors. Assuming that the masses of the particles are ordered as $m_1 \leq m_2 \leq \ldots$, this gives the exact spectral function for energies $\omega < 4m_1$.

The single particle contributions to $\mathcal{S}^{ij}(\omega, q = 0)$ takes the simple form

$$\mathcal{S}_1^{ij}(\omega, q = 0) = \sum_a \frac{2\pi}{m_a} \delta(\omega - m_a) F_a^i F_a^{j*}, \tag{D.5}$$

where $a$ here indexes all the different particles that couple to $\mathcal{O}_i$ and $\mathcal{O}_j$, i.e. they give a coherent contribution of isolated delta-function peaks.

The two-particle contributions take the form

$$\mathcal{S}_2^{ij}(\omega, q = 0) = \sum_{a_1 \leq a_2} \left(\frac{1}{2}\right)^{\delta_{a_1, a_2}} \frac{\Theta(\omega - (m_{a_1} + m_{a_2}))}{m_{a_1} m_{a_2} |\sinh(\theta_1 - \theta_2)|} F_{a_1, a_2}^i(\theta_1 - \theta_2) F_{a_2, a_1}^{j*}(\theta_1 - \theta_2), \tag{D.6}$$

where

$$\theta_1 - \theta_2 = \text{arccosh}\left(\frac{\omega^2 - m_{a_i}^2 - m_{a_j}^2}{2 m_{a_i} m_{a_j}}\right), \tag{D.7}$$

from energy and momentum conservation. The two-particle contribution is an incoherent continuum, which for a given two-particle pair $(a_1, a_2)$ opens at the threshold for $\omega = m_{a_1} + m_{a_2}$. As $\omega$ approaches this threshold from above, the kinematical prefactor generally introduces a square-root van Hove singularity in the spectral function; however, this singularity can be washed out in particular cases when the two-particle form factor vanishes as $\theta_1 - \theta_2 \to 0$.

Finally, the three particle contribution can be written as [38]

$$\mathcal{S}_3^{ij} = \sum_{a_1 \leq a_2 \leq a_3} \left(\prod_k^{\mathcal{N}} \frac{1}{N_k^{\{a_1, a_2, a_3\}}!}\right) \int \frac{d\theta_3}{2\pi} \frac{F_{a_1, a_2, a_3}^{\mathcal{O}_i}(\theta_1, \theta_2, \theta_3) F_{a_1, a_2, a_3}^{\mathcal{O}_j*}(\theta_1, \theta_2, \theta_3)}{m_{a_1} m_{a_2} |\sinh(\theta_1 - \theta_2)|}, \tag{D.8}$$

where the rapidities $\theta_1, \theta_2, \theta_3$ satisfy the kinematic constraints

$$\begin{aligned} \omega &= m_1 \cosh\theta_1 + m_2 \cosh\theta_2 + m_3 \cosh\theta_3, \\ 0 &= m_1 \sinh\theta_1 + m_2 \sinh\theta_2 + m_3 \sinh\theta_3. \end{aligned} \tag{D.9}$$

For three particles of equal mass $m_1 = m_2 = m_3 = m$, the solution of the kinematic constraints can be written explicitly. The integration range of $\theta_3$ is restricted to

$$\cosh\theta_3 \leq \frac{\omega^2 - 3m^2}{2m\omega}, \tag{D.10}$$

where D.9 have two solutions related by swapping the sign of $\theta_{12}$, given by

$$\begin{aligned} \cosh\theta_{12} &= \frac{\omega^2 - 2m\omega\cosh\theta_3 - m^2}{2m^2} \text{ with the choice } \theta_{12} \geq 0, \\ \cosh\theta_1 &= \frac{2\sqrt{(3 + 4\cosh\theta_{12} + \cosh 2\theta_3)\cosh^4\frac{\theta_{12}}{2} - \sinh\theta_3\sinh\theta_{12}}}{2(1 + \cosh\theta_{12})}, \end{aligned} \tag{D.11}$$

with the sign of $\theta_1$ chosen so that $\sinh\theta_1 + \sinh(\theta_1 - \theta_{12}) = -\sinh\theta_3$ is satisfied.

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
