# Peer review of "Duality and Form Factors in the Thermally Deformed Two-Dimensional Tricritical Ising Model"

_SciPost Physics, doi:SciPost Phys. 12, 162 (2022)_

## Round 1 · Referee Report · Anonymous (Referee 1) · 2021-11-12

Report

The authors consider the problem of the determination of form factors in the thermal (E7) integrable deformation of the tricritical Ising model in two dimensions. This model possesses two magnetisation operators (leading and subleading) which are not distinguished by a symmetry and for this reason satisfy the same form factor equations. From this point of view the problem is similar to that solved by Delfino and Simonetti in Phys. Lett. B 383, 450 (1996) for the magnetic (E8) integrable deformation of the Ising model. In that case the magnetisation and energy operators satisfy the same form factor equations because the magnetic field leaves no symmetry. An additional feature of the E7 case is that high-low temperature duality provides another sector containing a pair of disorder operators (duals of the magnetisation operators). The authors then exploit the fact that the clustering property provides a relation between the form factors of magnetisation and disorder operators, as observed by Cardy and Delfino in Nucl. Phys. B 519, 551 (1998) for the Potts model. Having determined the form factors, the authors compare them with the numerical results obtained by the truncated conformal space approach and observe an excellent agreement. Equally successful is the check performed using the sum rule for the conformal weights (Delta theorem). The form factors are then used for the calculation of the dynamical structure factors. The paper also contains introductory sections in which the authors review properties of the tricritical Ising model such as duality, Landau-Ginzburg description, supersymmetry and renormalisation group flows. The paper provides a very interesting application of the form factor bootstrap and I recommend publication in SciPost after inclusion of the two references mentioned above.

---

## Round 1 · Referee Report · Gerard Watts (Referee 2) · 2021-12-28

Strengths

1- This paper gives the first calculation of the form factors of the magnetic fields and their duals in the tri-critical Ising model.

2- The paper presents the calculation of potentially experimentally observable signatures

3- The form-factor calculations are checked using the truncated conformal space approach, giving an independent check of the results

4- There is a discussion of the simpler Ising model results and of the space of RG flows of the tri-critical Ising model which help make the paper accessible.

Weaknesses

1- There are various technical issues with the paper (see report) which need addressing. (These are not critical)

2- The truncated conformal space results do not agree with the exact results to the number of digits reported, and some of the graphs are very hard to read, so that the degree of agreement is not clear.

3- The form factor equations are not solved in general, the results are only presented for some low-particle number states.

Report

The core of the paper is the demonstration that the form-factor equations for the leading and sub-leading magnetisation operators $\sigma$ and $\sigma'$ in the thermally-perturbed tri-critical Ising model, and their dual disorder operators $\mu$ and $\mu'$, can be solved in terms of the high-temperature vacuum expectation values $\langle\mu\rangle$, $\langle\mu'\rangle$. This is supplemented by a check on various properties of the form-factor solutions using the truncated conformal space approach and a demonstration that the results give potentially experimentally verifiable signatures.

Since the tri-critical Ising model is one of the few experimentally realisable models, this seems a very worthwhile result and very deserving of publication and entirely suitable for publication in SciPost Physics.

The paper is generally very well written and easy to follow.

Having said that, I think the paper needs some minor revisions - there are places where it is unclear, and there are also some minor errors and typos that I have listed in the "requested changes" section.

Apart from the detailed changes, there are a few areas where some more commentary would be very helpful.

The first is to which extent do the TCSA calculations confirm, agree with, or simply not disagree with the form-factor calculations? Some of the TCSA data appears in excellent agreement with the exact results for a large range of parameters - for example $\langle 0|\sigma|\{n,-n\}\rangle_{1,2}$ in figure 6.1, $\langle 0|\sigma'|\{n,-n\}\rangle_{1,2}$ for $n>0$ in figure 6.2, $\sqrt{\rho_{12}}|\langle 0|\sigma|\{n,-n\}\rangle|$ for $n>1$ in figure 6.3, $\sqrt{\rho_{12}}|\langle 0|\sigma|\{n,-n\}\rangle|$ for $n>1$ in figure 6.4, $\sqrt{\rho_{11}}|\langle 0|\sigma|\{n,-n\}\rangle|$ for $n>0$ in figure 6.6, $\sqrt{\rho_{11}}|\langle 0|\sigma'|\{n,-n\}\rangle|$ for $n>0$ in figure 6.6, but many other comparisons are far worse. I could not find a commentary or critical discussion of the differences between TCSA and exact results when the agreement was poor, and why some results were better than others, other than that some poor agreement is down to truncation effects or small volumes leading to large exponential corrections, and so do not really know whether the TCSA results actually give support to the details of the form-factor expressions or not. Some discussion would be very helpful.

The authors also go to some lengths to explain that at a finite cutoff $\Lambda$, $\lambda_\Lambda$ should be non-zero to keep the $E_7$ symmetry. I would have thought that the same argument applied to the TCSA scheme as well with its cutoff $N$ and that a (small) non-zero coupling $\lambda_N$ would also be needed. Perhaps the authors could comment on this, and whether the effect is important or not?

Secondly, the form-factor equations have only been solved for low-particle number states, and not even, as far as I could tell, for all two-particle states, and so I would like to see some comments on whether there are any potential obstructions to finding form factors for general particle states. This does not affect the relevance of the results here which stand on their own.

Finally, the construction of the full CFT of the TIM including non-local fields is not completely straightforward and that without care, even the OPEs (2.15) and (3.12) are hard to understand, the OPEs with branch cuts cannot be defined (they are ambiguous up to a sign) and a full construction of all the correlation functions in the CFT is problematic. This does not affect any of the results in the paper as the [missing] details of the CFT construction are irrelevant, I think, to what is done here, and the extra details would again not, I think, affect the TCSA calculations which only need the action of the (local) perturbing field, so it is far from a criticism of the paper, but I would have hoped to see at least a mention of the ambiguity in the OPEs.

(As an aside, I believe the only full discussions in which all structure constants can be calculated and consistency shown are in the work of Fröhlich et al which gives a construction from TFT data (in this approach, the fields $\psi(z)$, $\mu(z)$ etc, arise as defect creation fields at the ends of $Z_2$ defect lines, which is a "bosonic" theory in that there is no coupling to the spin-structure of the underlying space) and in the subsequent work of Runkel et al [see arXiv:1506.07547 for the Ising model] of a "fermionic" version which does couple to the spin structure. Whether these are in some way equivalent on the plane or on the cylinder, or indeed in general, has not yet been checked, I think.)

I would like to apologise to the authors for the time taken to submit this report - the pandemic has meant that my administrative load has been much higher than normal and it has taken until the Christmas vacation period for me to find time to write a considered response.

Requested changes

  • On page 9, equation (2.15) has a couple of issues which mean it is hard to understand what it conveys. Most obviously, there are factors of $1/\sqrt 2$ missing (see equation (2.14)), so is (2.15) only defined up to a constant?

-On page 9, equation (2.16) is technically wrong as

$$ \langle0|\sigma(\infty)= \lim_{z\to\infty}\langle0|\sigma(z) =0\;.$$
It would be better to write $\langle\sigma|$ instead of $\langle0|\sigma(\infty)$ where
$$ \langle\sigma| = \lim_{z\to\infty} |z|^{1/4} \langle0|\sigma(z)\;.$$
or define what $\sigma(\infty)$ is.

  • On page 11, equation (2.32) looks wrong. It has the wrong dimension - is this instead $F_0^2$? I could not find the value of $\langle\mu\rangle$ in reference [26], instead that paper gives $\langle\sigma\rangle$ in the low temperature limit as $\langle\sigma\rangle=2^{1/12} e^{-1/8} A^{3/2} m^{1/8}$ which is not the square-root of $F_0$ as given in (2.32). I do not know what is going on. Is $F_0$ here found by applying duality to the result of [26]? Is the missing factor of $2^{1/6}$ because of a typo in [26] or a typo here or a different choice of ground state? I think this needs some explanation.

  • On page 11, in equation (2.13), what does $\simeq$ mean? Does this mean that $1/|x|^{1/4}$ is the leading behaviour as $x\to 0$? Why is it not "="?

  • On page 17, the introduction of superfields (with no reference to superspace) seems very odd, especially as the superspace formalism is only applied to the NS sector, and there is no corresponding discussion of the R sector as "spin fields". The fields in this expansion also have the wrong normalisation - if $t = G_{-1/2}\bar G_{-1/2}\epsilon$ then $t(z) t(w) \sim (-1/25)|z-w|^{-12/5}$

  • On page 17, there is no need to introduce $(-1)^F$ to deduce there is a two-dimensional space of $R$ highest weight states. The fact that ${G_0,\bar G_0} = 0$ is sufficient.

  • On page 18, the structure constants are missing from (3.12). This could be very confusing. There are same issues about the choice of sign in the square-root branch cut. [This could be solved by defining fields $G(z)$ and $\mu(z)$ as defect-creating fields for the $Z_2$ defect and a convention for the placement of the corresponding defect lines as apply in the free-fermion case - not that this is needed, but some observation would help]

  • On page 21, the fact that $\Delta_\epsilon$ is small means that $y_\epsilon = 1-\Delta_\epsilon$ is large and perturbation theory is unreliable. I don't think this affects the qualitative nature of the results, but I think some comment would be helpful.

  • On page 24, I think it is relevant to the discussion of fine-tuning of parameters in expriments to recall that there are four relevant fields and so four parameters need fine-tuning. The magnetic field parameters would be set to zero, but since one of these is the most relevant field, it would need to very accurately set to zero not to trigger a flow to a different fixed point.

  • On page 26, in equation (5.5), are the coefficients of $a_{ij}^k$ known exactly or only numerically?

  • On page 33, table 6.1, there is no indication of the accuracy or reliability of the TCSA data. If all the digits can be relied upon, then this table shows disagreement. There are errors of various sorts here - truncation and exponential correction - and an estimate of the error would be helpful, rather than just reporting large numbers of decimal places which are, presumably, "wrong".

  • On page 5, it says "the different Verma modules of the conformal field theory". The Verma module is not actually a physically relevant space, since it includes all Virasoro descendants, and states which are identically zero. It would be better to find a re-formulation of this sentence which avoids the use of the words "Verma" and instead referred to "highest-weight modules" or something similar.

  • In figures 6.1-6.6, it appears that there is excellent agreement only for $F^i_{12}$. For $F^i_{14}$ and $F^i_{23}$, the fit is substantially worse, to my eyes. I wonder if the authors could comment on this? What would a "wrong" form-factor look like? I cannot tell from these figures how much of the behaviour is dictated simply by the two-particle S-matrix, and how much from the detail of the form-factor, and whether the figures really indicate confirmation of any form-factors apart from $F^i_{12}$.

For example, in table 6.2, the TCSA and exact results for $F^{\sigma'}_{34}(i\pi)$ differ by close to 10%. Is this a sign of agreement or disagreement?

  • On page 54, there is an ansatz for the form factor in (A.8) which disagrees with the choices in (5.15) for non-locality of the particles with respect to the fields. If appendix A is meant to explain the form-factor ansatz, then I think the choice of the location of the factors $\cosh(\theta/2)$ and $1/\cosh(\theta/2)$ needs to be mentioned - as it is, appendix A does not do the job of explaining (5.15).

  • On page 58, I think the conformal transformation should be $z = \exp(2\pi w/L)$ so that $z$ is invariant under $w \to w + i L$.

  • On page 11, it is said that "$\mu$ has a semi-local index equal to 1/2 with respect to the operator $\sigma$" but in the appendix this is referred to as a "mutual locality index". I think it would be better to use the same name in both places.

  • On page 9, the choice of representation (2.14) is not very symmetric, requiring $\bar\psi_0$ to be given by $\pm\sigma_2$. There is nothing wrong with this choice, but it might be helpful to give the representation of $\bar\psi_0$.

  • On page 3, it says "The TIM is the second conformal minimal model", when it is the second unitary conformal minimal model

  • On page 11, at the top, the authors say "we consider local operator fields" but then very soon say that $\mu$ is not local. I do wonder why the top sentence is there - it might be better to change it.

  • On page 29, the authors say they introduce an extra $\cosh(\theta/2)$ in $F^{\tilde\Phi}_{13}(\theta)$ "because of the kink nature", but earlier on page 25 they say they are dealing with the high temperature phase in which the particles are just regular particle, the kinks only appearing in the low temperature phase (according to page 19). Perhaps this could be clarified?

  • On page 8, the introduction of "P" and "A" seems unnecessary and indeed confusing. These are often used to describe the periodicity on the cylinder/torus and have exactly the opposite identification (P=R, A=NS). Since "P" and "A" are not used outside this subsection, as far as I can tell, maybe it would be better not to introduce them here? I also found it confusing to say that the branch cut for the free fermion starts from the origin - there must be branch cuts terminating at each insertion of a spin or disorder field, it is just that in the case of the plane with the only insertion at the origin and infinity then the branch cut must run from 0 to infinity. The description here will not confuse anybody who already knows what is going on, but I do wonder if it would help someone who did not understand the system and was reading it for the first time.

  • Some simple typos - I hope the authors do not mind me listing these, in case they missed them - page 3 "orginary" should be "ordinary"? page 8 "close contour" should be "closed contour" page 11 "corresponds the form" should be "corresponds to the form" page 20 "cooples"should be "couples" page 26 "set of equation" should be "set of equations" page 30 "consider set.. which relating.." could be "consider a set..which relate"? pages 32 and 56 "factor ised" should be "factorised" page 33 "and exponential" should be "an exponential" page 35 "curves small" could be "curves for small"? page 40 "to the for" could be "to that for"? page 41 "just few" should be "just a few" pages 54 and 56 "factor isation" should be "factorisation" page 55 "$G_\alpha(\theta)$" should be "$g_\alpha(\theta)$" page 61 "the the" should be "the"

---

## Round 2 · Referee Report · Anonymous · 2022-1-27

Report

The paper can be published.

---

## Round 2 · Referee Report · Gerard Watts · 2022-4-3

Report

I would like to thank the authors for the changes they have made which I think answer my previous queries very satisfactorily. I would also like to apologise for the time taken to write this report, for which I can only say that pressure of work in the past semester has been rather high.

I have, I should say, found a few different points which I think the authors may want to clarify, but which do not detract from the main results, which I think are now perfectly acceptable.

The first point is that of the "$Z_2$" symmetry in the TIM. On page 12, this is said to be $\sigma \to - \sigma, \mu \to -\mu$ at the conformal point, but on page 11 it is said to be $\sigma\to-\sigma, \mu\to\mu$ in the high-temperature phase. I find this very confusing. If it is the same symmetry, then the symmetry ought to act in the massless limit in the same way it does in the massive theory. This led me to wonder what the symmetry of the CFT is - and given the full set of fields (local and non-local) and the non-zero OPE coefficients, I think it looks like $Z_2\times Z_2$, with two generators, the first acting as $\{\sigma\to-\sigma,\mu\to\mu,\psi\to-\psi,\bar\psi\to-\bar\psi\}$ and the second as $\{\sigma\to\sigma,\mu\to-\mu,\psi\to-\psi,\bar\psi\to-\bar\psi\}$, with their product being $\{\sigma\to-\sigma,\mu\to-\mu,\psi\to\psi,\bar\psi\to\bar\psi\}$. Anyway, I think the comments on page 10 and page 11 appear contradictory and it would be good to clarify what is going on.

Secondly, on page 20, I think it is not the currents that have spins $\{1,5,7..\}$, but the charges. In the simplest case, $T_{\mu\nu}$ has spin 2 (or $T(z)$ has spin 2, $\bar T(\bar z)$ has spin -2) but the charges $P_\mu$ have spin 1 (or $P$ spin 1 and $\bar P$ spin -1)

Finally, in equation (A.22), which is $\langle\Phi\rangle$? In the references, in [25] the equation relates the form factors of $\Phi_a$, $\Phi_b$ and $\Phi_c$ but does not divide by any expectation value, whereas [57] and [58] only deal with the case of a single field $\Phi$. At the most basic level, I think the authors should consider clarifying what the general formula (A.22) means, as well as the explicit case (A.23) and (A.24).

There are also some typos but none of these at all get in the way of intelligibility.

---

## Round 2 · Author Response

We are grateful to the referees for their constructive criticism, and especially to Gerard Watts to his very careful reading of the paper, and suggesting so many points of improvement. We give our detailed answers in the list of changes.

---

## Round 2 · List of Changes

\documentclass[11pt]{article}
\usepackage{amssymb}
\usepackage{amsfonts}
\usepackage{color}
\usepackage{geometry}
\geometry{verbose,tmargin=2.5cm,bmargin=2.5cm,lmargin=2cm,rmargin=2cm}

\title{List of changes}

\begin{document}

\maketitle

\section{Answers to Referee 1}

We added the two references suggested by the referee.

\section{Answers to Gerard Watts}

\subsection{Comments in the Report section}
\begin{enumerate}
\item

The first is to which extent do the TCSA calculations confirm, agree with, or simply not disagree with the form-factor calculations? Some of the TCSA data appears in excellent agreement with the exact results for a large range of parameters - for example $\langle 0 | \sigma | \left\{n,-n\right\}\rangle_{1,2}$ in figure 6.1,
$\langle | \sigma' | \left\{n,-n\right\}\rangle_{1,2}$ for $n>0$
in figure 6.2, $\sqrt{\rho_{1,2}}|\langle 0 | \sigma | \left\{n,-n\right\}\rangle|$
for $n>1$
in figure 6.3,
$\sqrt{\rho_{1,1}}|\langle 0 | \sigma' | \left\{n,-n\right\}\rangle|$ for $n>1$
in figure 6.4,
$\sqrt{\rho_{1,1}}|\langle 0 | \sigma | \left\{n,-n\right\}\rangle|$
for $n>0$
in figure 6.6, $\sqrt{\rho_{1,1}}|\langle 0 | \sigma' | \left\{n,-n\right\}\rangle|$
for $n>0$
in figure 6.6, but many other comparisons are far worse. I could not find a commentary or critical discussion of the differences between TCSA and exact results when the agreement was poor, and why some results were better than others, other than that some poor agreement is down to truncation effects or small volumes leading to large exponential corrections, and so do not really know whether the TCSA results actually give support to the details of the form-factor expressions or not. Some discussion would be very helpful.

\textcolor{blue}{We added a discussion on different sources of errors when form factors to TCSA data, see Appendix C.2. We also updated the captions of the plots to point out the main features.}

\item
The authors also go to some lengths to explain that at a finite cutoff $\Lambda$, $\lambda_{\Lambda}$
should be non-zero to keep the $E_7$
symmetry. I would have thought that the same argument applied to the TCSA scheme as well with its cutoff $N$ and that a (small) non-zero coupling $\lambda_N$
would also be needed. Perhaps the authors could comment on this, and whether the effect is important or not?

\textcolor{blue}{In the new Appendix C.2 we discuss the issue of running couplings.}

\item
Secondly, the form-factor equations have only been solved for low-particle number states, and not even, as far as I could tell, for all two-particle states, and so I would like to see some comments on whether there are any potential obstructions to finding form factors for general particle states. This does not affect the relevance of the results here which stand on their own.

\textcolor{blue}{All the Form Factors can be obtained in principle starting from Form Factors on generic $n$-multiparticle state containing only particles of $A_1$, which must be constructed as solutions to the Form Factor equations. All other Form Factors by looking at the residues on the various channels. There is no particular obstruction to doing so, we decided to do differently because we are more interested in using the lowest energy Form Factors relevant for experimental comparison.}

\item
Finally, the construction of the full CFT of the TIM including non-local fields is not completely straightforward and that without care, even the OPEs (2.15) and (3.12) are hard to understand, the OPEs with branch cuts cannot be defined (they are ambiguous up to a sign) and a full construction of all the correlation functions in the CFT is problematic. This does not affect any of the results in the paper as the [missing] details of the CFT construction are irrelevant, I think, to what is done here, and the extra details would again not, I think, affect the TCSA calculations which only need the action of the (local) perturbing field, so it is far from a criticism of the paper, but I would have hoped to see at least a mention of the ambiguity in the OPEs.

\textcolor{blue}{We agree with this comment. In both equations we have substituted $\sim$ with $=$, which means adopting a specific convention for the spin fields, as it was done, for instance, in one of the Appendices of the Belavin-Polyakov-Zamolodchikov paper on CFT. We also inserted comments regarding the branch cuts (see also below). In any case, this issue is not particular relevant for the rest of our paper.
}
\end{enumerate}

\subsection{Comments in the Requested changes section}
\begin{enumerate}
\item
On page 9, equation (2.15) has a couple of issues which mean it is hard to understand what it conveys. Most obviously, there are factors of $1/\sqrt{2}$ missing (see equation (2.14)), so is (2.15) only defined up to a constant?

\textcolor{blue}{We have corrected the OPE with the factor $\frac{1}{\sqrt{2}}$, and also added a comment on sigma/mu being defect operators changing the boundary conditions for the fermion field.
}

\item On page 9, equation (2.16) is technically wrong as
\begin{equation}
\langle 0 |\sigma(\infty) = \lim_{z\rightarrow\infty}\langle 0|\sigma(z) = 0.
\end{equation}
It would be better to write $\sigma|$
instead of $\langle 0|\sigma(\infty)$
where
\begin{equation}
\langle \sigma | = \lim_{z\rightarrow\infty |z|^{1/4}}\langle 0| \sigma(z).
\end{equation}
or define what $\sigma(\infty)$ is.

\textcolor{blue}{Corrected.}

\item On page 11, equation (2.32) looks wrong. It has the wrong dimension - is this instead
$F^2_0$
? I could not find the value of $\langle \mu \rangle$ in reference [26], instead that paper gives $\langle \sigma \rangle$
in the low temperature limit as $\langle \sigma \rangle=2^{1/12}e^{-1/8}A^{3/2}m^{1/8}$ which is not the square-root of $F_0$ as given in (2.32). I do not know what is going on. Is $F_0$
here found by applying duality to the result of [26]? Is the missing factor of $2^{1/6}$
because of a typo in [26] or a typo here or a different choice of ground state? I think this needs some explanation.

\textcolor{blue}{Updated. We meant to write $\langle\sigma\rangle$ as it is in [26].}

\item On page 11, in equation (2.13), what does $\simeq$
mean? Does this mean that $1/|x|^{1/4}$ is the leading behaviour as $x\rightarrow0$
? Why is it not "$=$"?

\textcolor{blue}{Yes, it means that it is just the leading term and specifies the operator normalisation.}

\item On page 17, the introduction of superfields (with no reference to superspace) seems very odd, especially as the superspace formalism is only applied to the NS sector, and there is no corresponding discussion of the R sector as "spin fields". The fields in this expansion also have the wrong normalisation - if $t=G_{-1/2}\bar{G}_{-1/2}\epsilon$ then $t(z)t(w)\sim (-1/25) |z-w|^{-12/5}$

\textcolor{blue}{We only intended to use the superfields for the NS sector, where we think it provides the most transparent to summarize the sectors operator content. Indeed the normalisation was not treated carefully, and corrected the text accordingly. For the R sector, we rather concentrated on an exposition that emphasizes the structures underlying KW duality.}

\item On page 17, there is no need to introduce $(-1)^F$ to deduce there is a two-dimensional space of $R$
highest weight states. The fact that $\{G_0,\bar{G}_0\}=0$
is sufficient.

\textcolor{blue}{That is true, but we decided to keep the fermionic number operator as well for full analogy with the Ising model.}

\item On page 18, the structure constants are missing from (3.12). This could be very confusing. There are same issues about the choice of sign in the square-root branch cut. [This could be solved by defining fields $G(z)$ and $\mu(z)$ as defect-creating fields for the $Z_2$ defect and a convention for the placement of the corresponding defect lines as apply in the free-fermion case - not that this is needed, but some observation would help]

\textcolor{blue}{We have corrected the OPE taking care of the proper normalization of the zero-mode of the field $G$.}

\item On page 21, the fact that $\Delta_{\epsilon}$ is small means that $y_{\epsilon} =1-\Delta_{\epsilon}$ is large and perturbation theory is unreliable. I don't think this affects the qualitative nature of the results, but I think some comment would be helpful.

\textcolor{blue}{We agree, but we are not doing any perturbation theory in our paper. The first terms of the beta functions are correct as they are, independently of any further use of perturbation theory.}

\item On page 24, I think it is relevant to the discussion of fine-tuning of parameters in expriments to recall that there are four relevant fields and so four parameters need fine-tuning. The magnetic field parameters would be set to zero, but since one of these is the most relevant field, it would need to very accurately set to zero not to trigger a flow to a different fixed point.

\textcolor{blue}{We agree, of course. We have added an extra sentence concerning the $Z_2$ odd sector of the theory which simply requires to properly control the absence of any external magnetic field.}

\item On page 26, in equation (5.5), are the coefficients of $a_{i,j}^k$ known exactly or only numerically?

\textcolor{blue}{They are known exactly as integrals given in the Appendix. Now this is indicated in the text, see page 27.}

\item On page 33, table 6.1, there is no indication of the accuracy or reliability of the TCSA data. If all the digits can be relied upon, then this table shows disagreement. There are errors of various sorts here - truncation and exponential correction - and an estimate of the error would be helpful, rather than just reporting large numbers of decimal places which are, presumably, "wrong".

\textcolor{blue}{A more careful exponential fit was carried out using different volume windows for the fit, see also the discussion in Appendix C.2. Now we only show the digits which are stable during the fit, except the last one. We indicate the variation of the last digit using different windows.}

\item On page 5, it says "the different Verma modules of the conformal field theory". The Verma module is not actually a physically relevant space, since it includes all Virasoro descendants, and states which are identically zero. It would be better to find a re-formulation of this sentence which avoids the use of the words "Verma" and instead referred to "highest-weight modules" or something similar.

\textcolor{blue}{We have changed the terminology to "highest-weight modules" to avoid confusion, as suggested.}

\item In figures 6.1-6.6, it appears that there is excellent agreement only for $F^i_{12}$. For $F^i_{14}$ and $F^i_{23}$
, the fit is substantially worse, to my eyes. I wonder if the authors could comment on this? What would a "wrong" form-factor look like? I cannot tell from these figures how much of the behaviour is dictated simply by the two-particle S-matrix, and how much from the detail of the form-factor, and whether the figures really indicate confirmation of any form-factors apart from $F^i_{12}$.
For example, in table 6.2, the TCSA and exact results for $F^{\sigma'}_{34}(i\pi)$
differ by close to $10\%$. Is this a sign of agreement or disagreement?

\textcolor{blue}{Possible sources of discrepancies comparing to TCSA now are carefully discussed in Appendix C.2. Regarding $F_{14}$ and $F_{23}$, we were able to find only the the lowest Bethe--Yang state, therefore the finite size effects are more pronounced for larger rapidities. Due to the coupled nature of the form factor equations, the different quantities are eventually very much related and so the agreement is very tight, which may be obscured somewhat by the fact that for some rapidity values the finite size effects are large due to the given data being extracted from relatively small volume. We also commented on the issue of $F^{\sigma'}_{34}(i\pi)$ in the caption of Table 6.2.}

\item On page 54, there is an ansatz for the form factor in (A.8) which disagrees with the choices in (5.15) for non-locality of the particles with respect to the fields. If appendix A is meant to explain the form-factor ansatz, then I think the choice of the location of the factors $\cosh(\theta/2)$ and $1/\cosh(\theta/2)$ needs to be mentioned - as it is, appendix A does not do the job of explaining (5.15).

\textcolor{blue}{We have slightly extended the discussion in the Appendix to give reason of the appeareance of this term $\cosh\theta/2$ (either in the numerator or denominator of the form factors) according to the non-locality of the operator wrt the excitations.}

\item On page 58, I think the conformal transformation should be $z=\exp(2\pi w/L)$ so that $z$ is invariant under $w\rightarrow w + iL$.

\textcolor{blue}{We have corrected it, thanks!}

\item On page 11, it is said that "$\mu$ has a semi-local index equal to $1/2$ with respect to the operator $\sigma$
" but in the appendix this is referred to as a "mutual locality index". I think it would be better to use the same name in both places.

\textcolor{blue}{We have made the terminology uniform.}

\item On page 9, the choice of representation (2.14) is not very symmetric, requiring $\bar{\psi}_0$ to be given by $\pm\sigma_2$
. There is nothing wrong with this choice, but it might be helpful to give the representation of $\bar{\psi}_0$.

\textcolor{blue}{We have not introduced any representation for $\bar\psi$, simply because we do not need it and we did not want to further complicate the text.}

\item On page 3, it says "The TIM is the second conformal minimal model", when it is the second *unitary* conformal minimal model

\textcolor{blue}{We have included "unitary".}

\item On page 11, at the top, the authors say "we consider local operator fields" but then very soon say that $\mu$ is not local. I do wonder why the top sentence is there - it might be better to change it.

\textcolor{blue}{We have changed it.}

\item On page 29, the authors say they introduce an extra $\cosh(\theta/2)$ in $F^{\tilde{\Phi}}_{13}(\theta)$ "because of the kink nature", but earlier on page 25 they say they are dealing with the high temperature phase in which the particles are just regular particle, the kinks only appearing in the low temperature phase (according to page 19). Perhaps this could be clarified?

\textcolor{blue}{What really matters is the non-locality of the operator wrt the excitations. The simplest way of determining this fact is to consider the low-temperature phase, where the excitations which are non-local wrt the order operator $\sigma$ are the kinks. Going to high-temperature phase, this converts to the non-locality of the {\em now} particles BUT with respect to the disorder operator $\mu$. We have added a few extra words in the text in this respect.}

\item On page 8, the introduction of "P" and "A" seems unnecessary and indeed confusing. These are often used to describe the periodicity on the cylinder/torus and have exactly the opposite identification (P=R, A=NS). Since "P" and "A" are not used outside this subsection, as far as I can tell, maybe it would be better not to introduce them here? I also found it confusing to say that the branch cut for the free fermion starts from the origin - there must be branch cuts terminating at each insertion of a spin or disorder field, it is just that in the case of the plane with the only insertion at the origin and infinity then the branch cut must run from 0 to infinity. The description here will not confuse anybody who already knows what is going on, but I do wonder if it would help someone who did not understand the system and was reading it for the first time.

\textcolor{blue}{We agree and therefore we have simplified the text accordingly.}

\item Some simple typos - I hope the authors do not mind me listing these, in case they missed them -
\begin{enumerate}
\item page 3 "orginary" should be "ordinary"? \textcolor{blue}{Corrected.}
\item page 8 "close contour" should be "closed contour" \textcolor{blue}{Corrected.}
\item page 11 "corresponds the form" should be "corresponds to the form" \textcolor{blue}{Corrected.}
\item page 20 "cooples"should be "couples" \textcolor{blue}{Corrected.}
\item page 26 "set of equation" should be "set of equations" \textcolor{blue}{Corrected.}
\item page 30 "consider set.. which relating.." could be "consider a set..which relate"? \textcolor{blue}{Corrected.}
\item pages 32 and 56 "factor ised" should be "factorised" \textcolor{blue}{Corrected.}
\item page 33 "and exponential" should be "an exponential" \textcolor{blue}{Corrected.}
\item page 35 "curves small" could be "curves for small"? \textcolor{blue}{Corrected.}
\item page 40 "to the for" could be "to that for"? \textcolor{blue}{Corrected.}
\item page 41 "just few" should be "just a few" \textcolor{blue}{Corrected.}
\item pages 54 and 56 "factor isation" should be "factorisation" \textcolor{blue}{Corrected.}
\item page 55 "$G_{\alpha}$" should be "$g_{\alpha}$" \textcolor{blue}{Corrected.}
\item page 61 "the the" should be "the" \textcolor{blue}{Corrected.}
\end{enumerate}

\end{enumerate}

\end{document}

Resubmission 2109.09767v3 on 11 April 2022

---

## Round 3 · Author Response

We thank the referee for his thorough review of our work. We have clarified the points raised in his review (c.f. the list of changes).

---

## Round 3 · List of Changes

• We have added a discusson of $\mathbb{Z}_2$ symmetries on page 10.

  • Indeed, it is the spin of the charges, not of the currents that is referred to on page 20. We corrected the statement accordingly.

  • We added the required clarification in Appendix A.

---

## Editorial Decision

published